# Improved Bayesian Regret Bounds for Thompson Sampling in Reinforcement Learning

**Ahmadreza Moradipari**[* †]     **Mohammad Pedramfar**[* ‡]     **Modjtaba Shokrian Zini**[* §]

**Vaneet Aggarwal** [¶]

## Abstract

In this paper, we prove the first Bayesian regret bounds for Thompson Sampling in reinforcement learning in a multitude of settings. We simplify the learning problem using a discrete set of surrogate environments, and present a refined analysis of the information ratio using posterior consistency. This leads to an upper bound of order $\widetilde{O}(H\sqrt{d_{l_1}T})$ in the time inhomogeneous reinforcement learning problem where $H$ is the episode length and $d_{l_1}$ is the Kolmogorov $l_1-$dimension of the space of environments. We then find concrete bounds of $d_{l_1}$ in a variety of settings, such as tabular, linear and finite mixtures, and discuss how how our results are either the first of their kind or improve the state-of-the-art.

## 1   Introduction

Reinforcement Learning (RL) is a sequential decision-making problem in which an agent interacts with an unknown environment typically modeled as a Markov Decision Process (MDP) [38, 8]. The goal of the agent is to maximize its expected cumulative reward. This problem has a variety of applications, including robotics, game playing, resource management, and medical treatments. The key challenge in RL is to balance the so-called exploration-exploitation trade-off efficiently: exploring unseen state-action pairs to gain more knowledge about the unknown environment or exploiting the current knowledge to maximize the expected cumulative reward. Two efficient approaches have been developed to control this trade-off: *optimism in the face of uncertainty* (OFU) and *Thompson Sampling* (TS) (or Posterior Sampling (PS)). OFU constructs a confidence set of statistically plausible MDPs that includes the true MDP with high probability and plays an optimistic policy according to the MDP with maximum gain from this set [5, 40]. TS samples a statistically plausible MDP from a posterior distribution and plays the optimistic policy of the sampled MDP [29, 31]. In this work, we focus on the latter, and by combining an information theoretical approach first introduced by [36] with analysis based on posterior consistency tools, we prove state-of-the-art Bayesian regret bounds in a variety of settings.

In this paper, we start by defining the Bayesian RL problem, where transition and reward functions are Bayesian and time inhomogeneous. The Bayesian RL problem we consider is more comprehensive than in previous works, as we allow for both Bayesian transition and Bayesian rewards, and do not make any assumption on their individual prior. To simplify the learning problem, we utilize the notion of surrogate environments, which is a discretization of the environments space, and its learning task and TS regret is a proxy to that of the main problem. The construction of the surrogate environments was first introduced by [18] with an incorrect proof, which is fixed in our work by

---

[*] Authors have equal contribution.

[†] Toyota Motor North America, InfoTech Labs, Mountain View, CA, USA, `ahmadreza.moradipari@toyota.com`

[‡] Purdue University, West Lafayette, IN, USA, `mpedramf@purdue.edu`

[§] `modjtaba.shokrianzini@gmail.com`

[¶] Purdue University, West Lafayette, IN, USA, `vaneet@purdue.edu`

This work was supported in part by the National Science Foundation under grant CCF-2149588 and Cisco, Inc.

37th Conference on Neural Information Processing Systems (NeurIPS 2023).

defining the surrogate environments through an optimization. Of main importance is the size of this new environment space. The Bayesian regret decomposes to the product of two terms, one being the cumulative mutual information of the environment and history traversed by the policy. By the well-known entropy estimation of the mutual information, this significant factor in the regret is connected to the $l_1-$dimensions ($d_{l_1}$) of the transition and reward functions space, which can be more succinctly interpreted as the $l_1-$dimension $d_{l_1}$ of the environment space. The latter is in turn estimated by the size of the space of surrogate environments.

The information ratio, representing a trade-off of exploration/exploitation, is the other significant term in the decomposition of the TS Bayesian regret. In an improvement to [18], our novel analysis of this ratio based on posterior consistency tools, shows that this trade-off is bounded by $H^{3/2}$, where $H$ is the episode length. This bound is general and independent of the dimension of transition/reward function space at each step, which is is a key factor behind the advantage of our regret bound, such as the $\sqrt{SA}$ advantage in the tabular case compared to [18], or the lack of any restriction on the prior (e.g., Dirichlet prior) compared to [31]. Following a further refined approach, we finally estimate the TS Bayesian regret to be $\widetilde{O}(\lambda\sqrt{d_{l_1}T})$ for large enough $T$ in the time inhomogeneous setting. Here, a new term 'value diameter' $\lambda$, which is the average difference of the optimal value functions at different states, is used in bounding the information ratio, where instead of $H^{3/2}$, we have the smaller term $\lambda H^{1/2}$. Bounding the information ratio with $\lambda$ is a conceptual contribution of our work, which shows that the ratio is bounded by a *value-dependent* term, which is in nature different from $H$ but always $\leq H + 1$. Further, there exists another bound for $\lambda$; in environments where states are reachable from one another in $D$ steps, we have $\lambda \leq D + 1$. In 'well-connected' MDPs, one could have $D \ll H$, implying an improvement over the $H^{3/2}$ information ratio bound.

Our generic bound is abstract in terms of $d_{l_1}$, so we estimate it in more explicit terms for useful applications. [18] have bounded $d_{l_1}$ in the tabular and linear case without formalizing this notion, and while for tabular MDPs, $d_{l_1}$ was bounded by $SAH$, for linear MDPs with feature space dimension $d_f$, we investigate their claim of the bound $d_f H$. Detailed in Appendix G, we show a counterexample to their analysis, and we manage to find a correct estimate in this setting. We also introduce finite mixtures MDPs and are the first to prove a TS Bayesian regret of order $\widetilde{O}(\lambda\sqrt{HmT})$, where $m$ is the number of mixtures.

Lastly, we note that our regret bound of order $\widetilde{O}(\lambda\sqrt{d_{l_1}T})$ is the first in the general nonlinear time inhomogeneous Bayesian RL setting for TS, and generalizing [31, Conj. 1], we conjecture it to be optimal if $\lambda$ can be replaced by $\widetilde{O}(\sqrt{H})$.

**Related work.** Since the introduction of information ratio by [35, 36], a new line of research has emerged to provide tighter regret bounds for TS. The general approach involves factoring the Bayesian regret into two components: an information ratio that captures the trade-off between optimal action selection and information gain, and a cumulative information gain term that depends on the target environment and the history of previous observations. Then, both components are bounded separately using information theoretic tools.

In the bandit setting, this analysis has been used to bound Bayesian regret for TS [14, 9], as well as that of a new algorithm called information-directed sampling (IDS) [35, 26, 23, 19, 20]. This analysis has also been used in partial monitoring [25, 24] and RL with a specific Dirichlet prior and additional assumptions [28, 27] or when the true environment is too complicated to learn [4]. More recently, [18] studied the Bayesian regret of TS in RL without any prior assumptions for tabular MDP. This is the closest work to our paper and we discuss our generalization in detail in Section 5.

The Bayesian tabular MDP case has also been studied with the additional Dirichlet prior assumption in [31], where they achieve a regret bound matching ours. In an independent approach, the first non-linear Bayesian RL model was considered by [16] with a regret bound of $dH^{3/2}T^{1/2}$ where $d$ is a notion of dimension of their model, but their results were limited to Gaussian process settings with linear kernels. Finally, [11] considered general non-linear Bayesian RL models and introduced an algorithm that obtains $dH^{1+\alpha/2}T^{1-\alpha/2}$ where $\alpha$ is a tuning parameter and $d$ is the dimension of $\mathcal{S} \times \mathcal{A} \times \mathcal{S}$.

It is worth noting that there is another line of work that incorporates confidence regions into TS to achieve Bayesian regret bounds that can match the best possible frequentist regret bounds by

UCB in both bandit settings [35] and RL [31, 30, 32, 12]. However, this technique often results in a sub-optimal Bayesian regret, as the best bound known for UCB itself is not optimal.

Table 1: Bayesian regret bounds for TS (i.e. PSRL)

| Reference | Tabular | Linear | General | Comments |
|---|---|---|---|---|
| [29] | $\sqrt{H^3 S^2 A L}$ | - | - | - |
| [30] | - | - | $L^* \sqrt{d_K d_E H L}$ | Uses Eluder dimension
Lipschitz assumption |
| [31] | $\sqrt{H^3 S A L}$ | - | - | Dirichlet prior |
| [28] | $\sqrt{H^3 S A L}$ | - | - | Assumptions on prior |
| [12] | $L^* \sqrt{H^3 S^2 A^2 L}$ | - | $L^* \gamma \sqrt{H L}$ | Assumptions on regularity & noise
Lipschitz assumption |
| [18] | $\sqrt{H^4 S^2 A^2 L}$ | - | - | - |
| This paper | $\lambda \sqrt{H^2 S A L}$ | $\lambda \sqrt{d_{l_1} H L}$ | $\lambda \sqrt{d_{l_1} H L}$ | Assumptions 1 & 2
Holds in the limit $L \to \infty$ |

As discussed in Section 4.3 of [16], the Lipschitz term $L^*$, which is used in the grayed papers in the table, may grow exponentially in episode length. Note that [18] claims a regret bound for the linear setting. However, as discussed in Appendix G.1, their proof is incorrect.

While our work's emphasis is on theoretical guarantees for TS, we discuss here the experiments using this algorithm. Previous works on PSRL [35, 26, 23, 20, 31] come with extensive experiments on TS (and/or its variants), and discussions on computational efficiency of PSRL. In particular, experiments in [31] support the assertion that "PSRL dramatically outperforms existing algorithms based on OFU". In addition, PSRL with oracle access has been shown to be the most performant, esp. when compared to recent OFU based UCBVI/UCBVI-B, or even variants of PSRL such as Optimistic PSRL [39, Fig. 1.3]. However, an important limitation in experiments is the need for oracle access to an optimal policy, and that can not be always satisfied efficiently. Nevertheless, clever engineering can make TS work even in large scale Deep RL. Indeed, for general RL settings, the recent work [37] shows how to implement TS in Deep RL on the Atari benchmark and concludes that "Posterior Sampling Deep RL (PSDRL) significantly outperforms previous state-of-the-art randomized value function approaches, its natural model-free counterparts, while being competitive with a state-of-the-art (model-based) reinforcement learning method in both sample efficiency and computational efficiency". In summary, experiments in the literature provide enough support for the empirical performance of TS.

## 2 Preliminaries

### 2.1 Finite-horizon MDP

We follow the literature's conventions in our notation and terminology to avoid confusion when comparing results. The environment is a tuple $\mathcal{E} = (\mathcal{S}, \mu_{\mathcal{S}}, \mathcal{A}, \mu_{\mathcal{A}}, H, \{P_h\}_{h=1}^H, \{r_h\}_{h=1}^H)$, where $\mathcal{S}$ is the topological measurable state space, $\mathcal{A}$ is the topological measurable action space, $\mu_{\mathcal{S}}$ and $\mu_{\mathcal{A}}$ are base probability measures on $\mathcal{S}$ and $\mathcal{A}$ respectively, $H$ is the episode length, $P_h : \mathcal{S} \times \mathcal{A} \to \Delta_{\mathcal{S}, \mu_{\mathcal{S}}}$ is the transition probability kernel, and $r_h : \mathcal{S} \times \mathcal{A} \to \Delta_{[0,1], \text{Lebesgue}}$ is the reward function, where we fix the convention $r(s, a) := \mathbb{E}_x[r(x|s, a)] = \int_0^1 x r(x|s, a) \, \mathrm{d}x$ as we mostly deal with its mean value. Notice that $\Delta_{X, \mu}$ is the set of probability distributions over $X$ that are absolutely continuous with respect to $\mu$. We will use $\Delta_X$ when the base measure is clear from the context. We assume $\mathcal{S}$, $\mathcal{A}$ are known and deterministic while the transition probability kernel and reward are unknown and random. Throughout the paper, the implicit dependence of $P_h$ and $r_h$ on $\mathcal{E}$ should be clear from the context.

Let $\Theta_h^P$ be the topological function space of $P_h$ and $\Theta^P = \Theta_1^P \times \cdots \times \Theta_H^P$ be the full function space. The space $\Theta_h^P$ is assumed to be separable and equipped with prior probability measure $\rho_h^P$ yielding the product prior probability measure $\rho^P = \rho_1^P \otimes \cdots \otimes \rho_H^P$ for $\Theta^P$. The exact same definition with similar notations $\Theta_h^R, \rho_h^R, \rho^R, \Theta^R$ applies for the reward function. Notice the explicit assumption of time inhomogeneity in these definitions, with all 'layers' $h$ being independent. The two sets define the set of all environments parametrized by $\Theta = \Theta_1 \times \cdots \times \Theta_H$ where $\Theta_h = \Theta_h^P \times \Theta_h^R$. Note that the prior is assumed to be known to the learner. This setting implies that an environment $\mathcal{E}$ sampled according to the prior $\rho = \rho^P \otimes \rho^R$ is essentially determined by its transition and reward functions pair $\{(P_h, r_h)\}_{h=1}^H$. We simplify the notation to view $\Theta$ as the set of all environments, i.e., saying

$\mathcal{E} \in \Theta$ should be viewed as $\{(P_h, r_h)\}_{h=1}^H \in \Theta$. The space of all possible real-valued functions $\{(P_h, r_h)\}_{h=1}^H$ has a natural vector space structure. Therefore it is meaningful to discuss the notion of the convex combination of environments. We assume that $\Theta$ is a convex subspace of the space of all possible environments. This assumption is not restrictive, since we may replace any environment space with its convex hull. Note that we do not assume that the support of the prior is convex.

*Remark* 1. The case of joint prior may be of interest, but to our knowledge all prior works also take $\rho^P, \rho^R$ to be independent.

**Agent, policy and history.** An agent starts at an initial state $s_1^\ell$, which is fixed for all episodes $\ell$. It observes a state $s_h^\ell$ at layer $h$ episode $\ell$, takes action $a_h^\ell$, and receives reward $r_h^\ell$. The environment changes to the next random state $s_{h+1}^\ell$ with probability $P_h(s_{h+1}^\ell | s_h^\ell, a_h^\ell)$. The agent stops acting at $s_{H+1}$ and the environment is reset to its initial state.

We define $\mathcal{H}_{\ell,h}$ as the history $(s_1^\ell, a_1^\ell, r_1^\ell, \dots, s_h^\ell, a_h^\ell, r_h^\ell)$. Denote by $\mathcal{D}_\ell = (\mathcal{H}_{1,H}, \dots, \mathcal{H}_{\ell-1,H})$ the history up to episode $\ell$, where $\mathcal{D}_1 := \emptyset$. Finally, let $\Omega_h = \prod_{i=1}^h (\mathcal{S} \times \mathcal{A} \times [0,1])$ be the set of all possible histories up to layer $h$.

A policy $\pi$ is represented by stochastic maps $(\pi_1, \dots, \pi_H)$ where each $\pi_h : \Omega_{h-1} \times \mathcal{S} \to \Delta_{\mathcal{A}, \mu_\mathcal{A}}$. Let $\Pi_S$ denote the entire stationary policy class, stationary meaning a dependence only on the current state and layer and let $\Pi \subseteq \Pi_S$.

**Value and state occupancy functions.** Define the value function $V_{h,\pi}^\mathcal{E}$ as the value of the policy $\pi$ interacting with $\mathcal{E}$ at layer $h$:

$$V_{h,\pi}^\mathcal{E}(s) := \mathbb{E}_\pi^\mathcal{E} \left[ \sum_{h'=h}^H r_{h'}(s_{h'}, a_{h'}) \Big| s_h = s \right], \tag{1}$$

where $\mathbb{E}_\pi^\mathcal{E}$ denotes the expectation over the trajectory under policy, transition, and reward functions $\pi, P_h, r_h$. The value function at step $H+1$ is set to null, $V_{H+1,\pi}^\mathcal{E}(\cdot) := 0$. We assume there is a measurable function $\pi_\mathcal{E}^* : \Theta \to \Pi$ such that $V_{h,\pi_\mathcal{E}^*}^\mathcal{E}(s) = \max_{\pi \in \Pi} V_{h,\pi}^\mathcal{E}(s), \forall s \in \mathcal{S}, h \in [H]$. The optimal policy $\pi^*$ is a function of $\mathcal{E}$, making it a random variable in the Bayesian setting. Lastly, let the *state-action occupancy probability measure* be $\mathbb{P}_\pi^\mathcal{E}(s_h = s, a_h = a)$, also known as the state occupancy measure under policy $\pi$ and environment $\mathcal{E}$. It follows from the definitions that this measure is absolutely continuous with respect to $\mu_{\mathcal{S} \times \mathcal{A}} := \mu_\mathcal{S} \times \mu_\mathcal{A}$. Let $d_{h,\pi}^\mathcal{E}(s, a)$ denote the Radon–Nikodym derivative so that we have $d_{h,\pi}^\mathcal{E}(s, a) \, \mathrm{d}\mu_{\mathcal{S} \times \mathcal{A}} = \mathrm{d}\mathbb{P}_\pi^\mathcal{E}(s_h = s, a_h = a)$. We will assume throughout the paper that this density $d_{h,\pi}^\mathcal{E}(s, a)$ is measurable and upper bounded for all $\pi, \mathcal{E}, s, a, h$. The upper bound is a reasonable assumption, and it happens trivially in the tabular case ($d_{h,\pi}^\mathcal{E}(s, a) \leq SA$). This also happens, e.g., when one assumes that the maps $(\mathcal{E}, s, a, s', h) \mapsto P_h^\mathcal{E}(s'|s, a)$ and $(\pi, s, a, h) \mapsto \pi_h(a|s)$ are continuous and $\Theta, \mathcal{S}, \mathcal{A}$ and the set of all optimal policies (as a subset of $\Pi$) are compact.

## 2.2 Bayesian regret

We formulate the expected regret over $L$ episodes and $T = LH$ total steps in an environment $\mathcal{E}$ as

$$\mathfrak{R}_L(\mathcal{E}, \pi) = \mathbb{E} \left[ \sum_{\ell=1}^L \left( V_{1,\pi_\mathcal{E}^*}^\mathcal{E}(s_1^\ell) - V_{1,\pi^\ell}^\mathcal{E}(s_1^\ell) \right) \right], \tag{2}$$

where the expectation is over the randomness of $\pi = \{\pi^\ell\}_\ell$. The Bayesian regret is $\mathfrak{BR}_L(\pi) = \mathbb{E}[\mathfrak{R}_L(\mathcal{E}, \pi)]$. For Thompson Sampling (TS), the algorithm selects the optimal policy of a given sample $\mathcal{E}_\ell$ picked from the posterior $\mathcal{E}_\ell \sim \mathbb{P}(\mathcal{E} \in \cdot | \mathcal{D}_\ell)$:

$$\pi_{\mathrm{TS}}^\ell = \mathrm{argmax}_{\pi \in \Pi} V_{1,\pi}^{\mathcal{E}_\ell}(s_1^\ell). \tag{3}$$

Importantly, the law of TS aligns with the posterior, i.e., $\mathbb{P}(\mathcal{E}|\mathcal{D}_\ell) = \mathbb{P}(\pi_{\mathrm{TS}}^\ell = \pi_\mathcal{E}^* | \mathcal{D}_\ell)$.

*Remark* 2. Note that $\mathbb{P}(\pi_{\mathrm{TS}}^\ell = \pi_\mathcal{E}^* | \mathcal{D}_\ell)$ is a probability for a specific measure on the space of optimal policies. To ensure that $\int_{\Pi^*} \mathbb{P}(\pi^*|\mathcal{D}_\ell) \mathrm{d}\rho_{\Pi^*} = 1$, we need an appropriate measure $\rho_{\Pi^*}$ on $\Pi^*$. Given the law of TS, the natural choice for this measure is the push-forward of the prior measure $\rho$ under the map $star : \Theta \to \Pi^*$, where $star(\mathcal{E}) = \pi_\mathcal{E}^*$.

## 2.3 Notations

For Bayesian RL, conditional expressions involving a given history $\mathcal{D}_\ell$ are widely used. We adopt the notation in [18] to refer to such conditionals; let $\mathbb{P}_\ell(\cdot) := \mathbb{P}(\cdot|\mathcal{D}_\ell), \mathbb{E}_\ell[\cdot] := \mathbb{E}[\cdot|\mathcal{D}_\ell]$. We can rewrite the Bayesian regret as

$$\mathfrak{BR}_L(\pi) = \sum_{\ell=1}^{L} \mathbb{E} \left[ \mathbb{E}_\ell \left[ V_{1,\pi_\mathcal{E}^*}^\mathcal{E}(s_1^\ell) - V_{1,\pi}^\mathcal{E}(s_1^\ell) \right] \right] \tag{4}$$

and define the conditional mutual information $\mathbb{I}_\ell(X;Y) := D_{\mathrm{KL}}(\mathbb{P}((X,Y) \in \cdot|\mathcal{D}_\ell)||\mathbb{P}(X \in \cdot|\mathcal{D}_\ell) \otimes \mathbb{P}(Y \in \cdot|\mathcal{D}_\ell))$. For a random variable $\chi$ and random policy $\pi$, the following will be involved in the information ratio:

$$\mathbb{I}_\ell^\pi(\chi; \mathcal{H}_{\ell,h}) := \mathbb{I}_\ell(\chi; \mathcal{H}_{\ell,h}|\pi) = \mathbb{E}_\pi[D_{\mathrm{KL}}(\mathbb{P}_\ell((\chi, \mathcal{H}_{\ell,h}) \in \cdot|\pi)||\mathbb{P}_\ell(\chi \in \cdot|\pi) \otimes \mathbb{P}_\ell(\mathcal{H}_{\ell,h} \in \cdot|\pi))], \tag{5}$$

Note that $\mathbb{E}[\mathbb{I}_\ell(X;Y)] = \mathbb{I}(X;Y|\mathcal{D}_\ell)$. To clarify, $\mathbb{P}_\ell(\mathcal{H}_{\ell,h} \in \cdot|\pi)$ is the probability of $\mathcal{H}_{\ell,h}$ being generated under $\pi$ within some environment. Given that the histories under consideration are generated by the TS algorithm, they are always generated in the true environment $\mathcal{E}$ under an optimal policy $\pi_{\mathcal{E}'}^*$. For $\pi = \pi_{\mathrm{TS}}^\ell$, this can be computed as $\mathbb{P}_\ell(\mathcal{H}_{\ell,h}|\pi) = \int_\mathcal{E} P(\mathcal{H}_{\ell,h}|\pi, \mathcal{E}) \, \mathrm{d}\mathbb{P}_\ell(\mathcal{E})$, where $P(\mathcal{H}_{\ell,h}|\pi, \mathcal{E})$ is an expression in terms of transition and reward functions of $\mathcal{E}$ and $\pi$.

Finally, we define $\bar{\mathcal{E}}_\ell$ as the mean MDP where $P_h^{\bar{\mathcal{E}}_\ell}(\cdot|s,a) = \mathbb{E}_\ell[P_h^\mathcal{E}(\cdot|s,a)]$ is the mean of posterior measure, and similarly for $r_h^{\bar{\mathcal{E}}_\ell}(\cdot|s,a) = \mathbb{E}_\ell[r_h^\mathcal{E}(\cdot|s,a)]$. We note that under the independence assumption across layers, the same is given for the state-occupancy density $d_{h,\pi}^{\bar{\mathcal{E}}_\ell} = \mathbb{E}_\ell[d_{h,\pi}^\mathcal{E}]$.

## 3 Bayesian RL problems

**Definition 1.** A Bayesian RL in this paper refers to the time-inhomogeneous finite-horizon MDP with independent priors on transition and reward functions, as described in Section 2.1.

The Bayesian RL *problem* is the task of finding an algorithm $\pi$ with optimal Bayesian regret as defined in Eq. (4). Below we list the variations of this problem. A setting considered by most related works such as [31, 16] is the following:

**Definition 2.** The **time (reward) homogeneous** Bayesian RL refers to the Bayesian RL setting where the prior $\rho^P$ ($\rho^R$) is over the space $\Theta^P$ ($\Theta^R$) containing the single transition (reward) function $P$ ($r$) defining $\mathcal{E}$, i.e., all layers have the same transition (reward) functions.

**Definition 3.** The **tabular** Bayesian RL is a Bayesian RL where $\mathcal{S}, \mathcal{A}$ are finite sets.

**Definition 4** (Linear MDP [41, 22]). Let $\phi^P : \mathcal{S} \times \mathcal{A} \to \mathbb{R}^{d_f^P}, \phi^R : \mathcal{S} \times \mathcal{A} \to \mathbb{R}^{d_f^R}$ be feature maps with bounded norm $\|\phi^P(s,a)\|_2, \|\phi^R(s,a)\|_2 \leq 1$. The **linear** Bayesian RL is a Bayesian RL where for any $\mathcal{E} = \{(P_h^\mathcal{E}, r_h^\mathcal{E})\}_{h=1}^H \in \Theta$, there exists vector-valued maps $\psi_h^{P,\mathcal{E}}(s), \psi_h^{R,\mathcal{E}}(s)$ with bounded $l_2-$norm such that for any $(s,a) \in \mathcal{S} \times \mathcal{A}$,

$$P_h^\mathcal{E}(\cdot|s,a) = \langle \phi^P(s,a), \psi_h^{P,\mathcal{E}}(\cdot) \rangle, \quad r_h^\mathcal{E}(\cdot|s,a) = \langle \phi^R(s,a), \psi_h^{R,\mathcal{E}}(\cdot) \rangle \tag{6}$$

A restricted version of the finite mixtures called linear mixture was first considered in [6] in the frequentist setting. Here, we consider the general setting.

**Definition 5.** The **finite mixtures** Bayesian RL is a Bayesian RL where for any $h \in [H]$ there exists fixed conditional distributions $\{Z_{h,i}^P : \mathcal{S} \times \mathcal{A} \to \Delta_\mathcal{S}\}_{i=1}^{m_h^P}$ and $\{Z_{h,i}^R : \mathcal{S} \times \mathcal{A} \to \Delta_{[0,1]}\}_{i=1}^{m_h^R}$, such that for any environment $\mathcal{E}$ given by $\{(P_h^\mathcal{E}, r_h^\mathcal{E})\}_{h=1}^H$, there exists parametrized probability distributions $a_h^{P,\mathcal{E}} : \mathcal{S} \times \mathcal{A} \to \Delta_{m_h^P}, a_h^{R,\mathcal{E}} : \mathcal{S} \times \mathcal{A} \to \Delta_{m_h^R}$ such that

$$P_h^\mathcal{E}(\cdot|s,a) = \sum_{i=1}^{m_h^P} a_{h,i}^{P,\mathcal{E}}(s,a) Z_{h,i}^P(\cdot|s,a), \quad r_h^\mathcal{E}(\cdot|s,a) = \sum_{i=1}^{m_h^R} a_{h,i}^{R,\mathcal{E}}(s,a) Z_{h,i}^R(\cdot|s,a) \tag{7}$$

# 4  Surrogate learning

Next, we define the discretized surrogate learning problem, and bound the size of the surrogate environments space, a significant term in the regret. To do so, we need to first define the Kolmogorov dimension of a set of parametrized distributions, esp. working out the case of $l_1-$distance. In the definitions below, we implicitly assume any required minimal measurability assumptions on the involved sets.

**Definition 6.** Given a set $\mathcal{F}$ of $\mathcal{O}-$parametrized distributions $P : \mathcal{O} \to \Delta(\mathcal{S})$ over a set $\mathcal{S}$ where both $\mathcal{O}, \mathcal{S}$ are measurable. Let $\mathcal{M}(\cdot, \cdot) : \mathcal{F} \times \mathcal{F} \to \mathbb{R}^{\geq 0}$ be a *distance*, i.e., $\mathcal{M}(P, Q) \geq 0 \overset{=}{\leftrightarrow} P = Q$. Then its right $\varepsilon-$covering number is the size $K_{\mathcal{M}}(\varepsilon)$ of the smallest set $\mathcal{C}_{\mathcal{M}}(\varepsilon) = \{P_1, \ldots, P_{K_{\mathcal{M}}(\varepsilon)}\} \subset \mathcal{F}$ such that

$$\forall P \in \mathcal{F}, \; \exists P_j \in \mathcal{C}_{\mathcal{M}}(\varepsilon) : \; \mathcal{M}(P, P_j) \leq \varepsilon. \tag{8}$$

The potential asymmetry of $\mathcal{M}$ (e.g., KL-divergence) requires the notion of left/right covering number. The right covering number will be the default, so covering number will always refer to that.

**Definition 7.** Let $d_{\mathcal{M}}(\varepsilon) = \log(K_{\mathcal{M}}(\varepsilon))$. Define the Kolmogorov $\mathcal{M}-$dimension $d_{\mathcal{M}}$ of $\mathcal{F}$ as

$$d_{\mathcal{M}} = \limsup_{\varepsilon \to 0} \frac{d_{\mathcal{M}}(\varepsilon)}{\log(\frac{1}{\varepsilon})}. \tag{9}$$

For $l_1(P, Q) := \sup_{o \in \mathcal{O}} ||P(\cdot|o) - Q(\cdot|o)||_1$, applying Definition 6 to the sets $\Theta_h^P, \Theta_h^R$ with $\mathcal{O} = \mathcal{S} \times \mathcal{A}$, and denote the respective covering numbers by $L_h^P(\varepsilon), L_h^R(\varepsilon)$ corresponding to covering sets $\mathcal{C}_h^P(\varepsilon), \mathcal{C}_h^R(\varepsilon)$. Similarly applying Eq. (9) and denote the corresponding $l_1-$dimensions by $d_{l_1,h}^P(\varepsilon), d_{l_1,h}^R(\varepsilon), d_{l_1,h}^P, d_{l_1,h}^R$ and $d_{l_1}^P := \sum_h d_{l_1,h}^P, d_{l_1}^R := \sum_h d_{l_1,h}^R$. The sums $d_{l_1,h} := d_{l_1,h}^P + d_{l_1,h}^R, d_{l_1} := d_{l_1}^P + d_{l_1}^R$ can be interpreted as the $l_1-$dimension of $\Theta_h$ and $\Theta$, i.e., the environment space.

*Remark* 3. We can also apply this framework to the KL-divergence, by $\mathcal{M}_{\mathrm{KL}}(P, Q) := \sup_{o \in \mathcal{O}} D_{\mathrm{KL}}(P(\cdot|o)||Q(\cdot||o))$. This was implicitly used by [18] to prove their regret bound in the tabular case. Note that Pinsker's lemma (Lemma 9) implies that the KL-divergence is larger than the squared total variance, and the latter is trivially larger than the $l_1$ distance. Therefore, $l_1-$dimension is smaller than $d_{\mathcal{M}_{\mathrm{KL}}}$, allowing for tighter regret bounds.

We now revisit the definition of $\varepsilon-$value partitions and show their existence is guaranteed by finite $l_1-$covering numbers. These partitions are the origins of surrogate environments.

**Definition 8.** Given $\varepsilon > 0$, an $\varepsilon-$value partition for a Bayesian RL problem is a partition $\{\Theta_k\}_{k=1}^K$ over $\Theta$ such that for any $k \in [K]$ and $\mathcal{E}, \mathcal{E}' \in \Theta_k$,

$$V_{1,\pi_{\mathcal{E}}^*}^{\mathcal{E}}(s_1^\ell) - V_{1,\pi_{\mathcal{E}}^*}^{\mathcal{E}'}(s_1^\ell) \leq \varepsilon. \tag{10}$$

A *layered* $\varepsilon-$value partition is one where the transition functions are independent over layers after conditioning on $k$. Throughout this paper, we will only consider layered $\varepsilon-$value partition. We define $K_{\mathrm{surr}}(\varepsilon)$ as the minimum $K$ for which there exists a layered $\varepsilon-$value partition.

Inspired by Eq. (9), we define the surrogate dimension as $d_{\mathrm{surr}} = \limsup_{\varepsilon \to 0} \frac{K_{\mathrm{surr}}(\varepsilon)}{\log(1/\varepsilon)}$.

**Lemma 1.** *Given a Bayesian RL, we have $K_{\mathrm{surr}}(\varepsilon) \leq \prod_h L_h^P(\varepsilon/(2H)^2) \times L_h^R(\varepsilon/(4H))$. This implies $d_{\mathrm{surr}} \leq d_{l_1}$.*

The above is proved in Appendix B. It is hard to find $d_{\mathrm{surr}}$, but one can estimate $d_{l_1}$, and according to the above, this acts as a proxy for $K_{\mathrm{surr}}$. This is useful as the regret relates to $K_{\mathrm{surr}}$. But to show this, we need to construct *surrogate environments* inside each partition, and show that learning those is almost equivalent to the original problem. Let $\zeta$ be a discrete random variable taking values in $\{1, \cdots, K_{\mathrm{surr}}(\varepsilon)\}$ that indicates the partition $\mathcal{E}$ lies in, such that $\zeta = k$ if and only if $\mathcal{E} \in \Theta_k$.

**Lemma 2.** *For any $\varepsilon-$value partition and any $\ell \in [L]$, there are random environments $\tilde{\mathcal{E}}_\ell^* \in \Theta$ with their laws only depending on $\zeta, \mathcal{D}_\ell$, such that*

$$\mathbb{E}_\ell \left[ V_{1,\pi_{\mathcal{E}}^*}^{\mathcal{E}}(s_1^\ell) - V_{1,\pi_{TS}^\ell}^{\mathcal{E}}(s_1^\ell) \right] - \mathbb{E}_\ell \left[ V_{1,\pi_{\mathcal{E}}^*}^{\tilde{\mathcal{E}}_\ell^*}(s_1^\ell) - V_{1,\pi_{TS}^\ell}^{\tilde{\mathcal{E}}_\ell^*}(s_1^\ell) \right] \leq \varepsilon. \tag{11}$$

The expectation in both equations is over $\mathcal{E}$ and $\pi_{TS}^\ell \in \{\pi_{\mathcal{E}'}^*\}_{\mathcal{E}' \in \Theta}$, with both sampled independently $\sim \mathbb{P}_\ell(\cdot)$, and the $K$ different values of $\tilde{\mathcal{E}}_\ell^*$. The second expectation over $(\tilde{\mathcal{E}}_\ell^*, \mathcal{E})$ is over pairs that are in the same partition, i.e., $\tilde{\mathcal{E}}_\ell^*, \mathcal{E}$ are independent only after conditioning on $\zeta$.

We note that the proof in [18, App. B.1] contains the use of a lemma that does not apply to construct the law of the environment $\tilde{\mathcal{E}}_\ell^*$. More details is provided in Appendix C, where we find $\tilde{\mathcal{E}}_\ell^*$ by minimizing an expected value of $\pi_{TS}^\ell$.

# 5  Bayesian regret bounds for Thompson Sampling

## 5.1  General Bayesian regret bound

We start by introducing the notion of value diameter.

**Definition 9.** Given the environment $\mathcal{E}$, its value diameter is defined as

$$\lambda_\mathcal{E} := \max_{1 \leq h \leq H} (\sup_s V_{h,\pi_\mathcal{E}^*}^\mathcal{E}(s) - \inf_s V_{h,\pi_\mathcal{E}^*}^\mathcal{E}(s)) + \max_{1 \leq h \leq H, s \in \mathcal{S}, a \in \mathcal{A}} (r_h^{\sup}(s,a) - r_h^{\inf}(s,a)),$$

where $r_h^{\sup}(s,a)$ (and $r_h^{\inf}(s,a)$) is the supremum (and infimum) of the set of rewards that are attainable under the distribution $r_h(s,a)$ with non-zero probability. As a special case, if rewards are deterministic, then we have $r_h^{\sup}(s,a) = r_h^{\inf}(s,a)$ for all $s, a$. The (average) value diameter over $\Theta$ is denoted by $\lambda := \mathbb{E}_{\mathcal{E} \sim \rho}[\lambda_\mathcal{E}^2]^{1/2}$.

As the value function is between 0 and $H$, we have $\lambda_\mathcal{E} \leq H + 1$ implying $\lambda \leq H + 1$. Note that value diameter is closely related to the notion of diameter commonly defined in finite RL problems. Strictly speaking, for a time-homogeneous RL, it is straightforward to see that the value diameter is bounded from above by one plus the diameter [33].

We now discuss the assumptions surrounding our results. The main technical assumption of this paper is the existence of consistent estimators, which as we will see in Appendix K, is closely related to the notion of posterior consistency:

**Assumption 1.** *There exists a strongly consistent estimator of the true environment given the history.*

Roughly speaking, we assume that with unlimited observations under TS, it is possible to find the true environment. For this assumption to fail, we need to have two environments that produce the same distribution over histories under TS and are therefore indistinguishable from the point of view of TS. The precise description of this assumption is detailed in Appendix K.

Another necessary technical assumption is that almost all optimal policies visit almost all state action pairs in their respective environment.

**Assumption 2.** *For almost every environment $\mathcal{E} \in \Theta$ and almost every $(s,a) \in \mathcal{S} \times \mathcal{A}$ and every $h \in [H]$, we have*

$$d_{h,\pi_\mathcal{E}^*}^\mathcal{E}(s,a) \neq 0.$$

Recall that, for any environment $\mathcal{E} \in \Theta$, the policy $\pi_\mathcal{E}^*$ is the optimal policy of $\mathcal{E}$ within the policy class $\Pi$. Therefore, one example of how the above assumption holds is when $\Pi$ is the set of $\varepsilon$-greedy algorithms and transition functions of environments assign non-zero probability to every state. Under these assumptions, we discuss our main result and its corollaries.

**Theorem 3.** *Given a Bayesian RL problem, for all $\varepsilon > 0$, we have*

$$\mathfrak{BR}_L(\pi_{TS}) \leq 2\lambda \sqrt{\log(K_{\mathrm{surr}}(\varepsilon))T} + L\varepsilon + T_0 \tag{12}$$

*where $T_0$ does not depend on $T$. This can be further upper bounded by*

$$\mathfrak{BR}_L(\pi_{TS}) \leq \widetilde{O}(\lambda\sqrt{d_{l_1}T}). \tag{13}$$

*for large enough $T$. Given a homogeneous $l_1$ dimension $d_{\mathrm{hom}} = d_{l_1,h}, \forall h$, this simplifies to*

$$\mathfrak{BR}_L(\pi_{TS}) \leq \widetilde{O}(\lambda\sqrt{H d_{\mathrm{hom}}T}). \tag{14}$$

*Remark* 4. For all regret bounds, we will replace $\lambda \leq H + 1$ to compare our result. For the case of homogeneous dimensions, we obtain $\widetilde{O}(H^{3/2}\sqrt{d_{\mathrm{hom}}T})$. Crucially, our main result shows a new conceptual understanding of the information ratio by bounding it by two terms of different nature: $H$ and $\lambda$, where the latter can be bounded by either the largest diameter of the environments or $H$.

*Remark* 5. Despite not impacting the asymptotics, the impact of $T_0$ can be large depending on the structure of the RL problem, and could be dominant even for large $T$s in practice.

*Remark* 6. Considering time as a part of the state observation, one could apply this regret analysis to particular time-homogeneous settings. However, this mapping of time-inhomogeneous RLs to homogeneous ones is not surjective, hence the result above does not readily extend to time-homogeneous settings.

While [16] were the first to consider a nonlinear Bayesian RL model, their bound is limited to the Gaussian process (with linear kernel) setting, while ours in the nonlinear time inhomogeneous setting makes no assumptions on the prior and is the first such bound. Our novel analysis allow us to upper bound the information ratio by $\lambda\sqrt{H}$ instead of, for example $H^{3/2}\sqrt{SA}$ ([18]) in the tabular case, improving the regret bound by a square root relevant to the dimension $d$ of the problem.

The detailed proof is given in Appendix D. Following [18], the regret (4) is rewritten using Lemma 2 to reduce the problem into its surrogate, and we use the well-known information-ratio trick by multiplying and dividing by the mutual information. We follow that with a Cauchy-Schwarz, summarized below

$$\mathfrak{BR}_L(\pi_{\mathrm{TS}}) \leq \mathbb{E}\left[\sum_{\ell=1}^{L} \frac{\mathbb{E}_\ell\left[V_{1,\pi_\mathcal{E}^*}^{\tilde{\mathcal{E}}_\ell^*}(s_1^\ell) - V_{1,\pi_{\mathrm{TS}}^\ell}^{\tilde{\mathcal{E}}_\ell^*}(s_1^\ell)\right]}{\sqrt{\mathbb{I}_\ell^{\pi_{\mathrm{TS}}^\ell}(\tilde{\mathcal{E}}_\ell^*; \mathcal{H}_{\ell,H})}} \sqrt{\mathbb{I}_\ell^{\pi_{\mathrm{TS}}^\ell}(\tilde{\mathcal{E}}_\ell^*; \mathcal{H}_{\ell,H})}\right] + L\varepsilon \quad (15)$$

$$\leq \sqrt{\mathbb{E}\left[\sum_{\ell=1}^{L} \frac{\left(\mathbb{E}_\ell\left[V_{1,\pi_\mathcal{E}^*}^{\tilde{\mathcal{E}}_\ell^*}(s_1^\ell) - V_{1,\pi_{\mathrm{TS}}^\ell}^{\tilde{\mathcal{E}}_\ell^*}(s_1^\ell)\right]\right)^2}{\mathbb{I}_\ell^{\pi_{\mathrm{TS}}^\ell}(\tilde{\mathcal{E}}_\ell^*; \mathcal{H}_{\ell,H})}\right] \mathbb{E}\left[\sum_{\ell=1}^{L} \mathbb{I}_\ell^{\pi_{\mathrm{TS}}^\ell}(\tilde{\mathcal{E}}_\ell^*; \mathcal{H}_{\ell,H})\right]} + L\varepsilon \quad (16)$$

Note the cost $\varepsilon$ at each episode (Lemma 2) in the first inequality, yielding the overall error $L\varepsilon$. Then, we can bound the mutual information appearing in the regret term by $\mathbb{E}\left[\sum_{\ell=1}^{L} \mathbb{I}_\ell^{\pi_{\mathrm{TS}}^\ell}(\tilde{\mathcal{E}}_\ell^*; \mathcal{H}_{\ell,H})\right] = I_\ell^{\pi_{\mathrm{TS}}^\ell}(\tilde{\mathcal{E}}_\ell^*; \mathcal{D}_\ell) \leq I_\ell^{\pi_{\mathrm{TS}}^\ell}(\zeta; \mathcal{D}_\ell) \leq H(\zeta) \leq \log(K_{\mathrm{surr}}(\varepsilon))$, where we used the mutual information chain rule, followed by data processing inequality to substitute $\tilde{\mathcal{E}}_\ell^* \to \zeta$, and finally used the trivial bound by the entropy. But the main novelty of our approach lies in our control of the first term

$$\Gamma_\ell(\pi_{\mathrm{TS}}^\ell) := \frac{\left(\mathbb{E}_\ell\left[V_{1,\pi_\mathcal{E}^*}^{\tilde{\mathcal{E}}_\ell^*}(s_1^\ell) - V_{1,\pi_{\mathrm{TS}}^\ell}^{\tilde{\mathcal{E}}_\ell^*}(s_1^\ell)\right]\right)^2}{\mathbb{I}_\ell^{\pi_{\mathrm{TS}}^\ell}(\tilde{\mathcal{E}}_\ell^*; \mathcal{H}_{\ell,H})} \quad (17)$$

called the information ratio. In our analysis, we have the following bound on its expectation.

$$\mathbb{E}[\Gamma_\ell(\pi_{\mathrm{TS}}^\ell) \mid \mathcal{E}_0] \leq \mathbb{E}\left[\sum_h \int \frac{\mathbb{E}_\ell\left[(\lambda_\mathcal{E} d_{h,\pi^*}^{\bar{\mathcal{E}}_\ell}(s,a))^2\right]}{\mathbb{E}_\ell\left[d_{h,\pi^*}^{\bar{\mathcal{E}}_\ell}(s,a)\right]} \mu_{\mathcal{S}\times\mathcal{A}} \mid \mathcal{E}_0\right],$$

where the average is taken over all histories $\mathcal{D}_\ell$ that are generated from running TS on the true environment $\mathcal{E}_0$, and we have introduced the smaller term $\lambda_\mathcal{E}$ instead of $H$ in [18]. While [18] essentially bound the above only in the tabular setting with $SAH^3$, we manage to generally bound the above with a more precise bound using Doob's consistency theorem. Assumption 1 allows us to use Doob's consistency theorem to conclude that for almost every environment $\mathcal{E}_0$, almost every infinite sequence of histories $(\mathcal{D}_\ell)_{\ell=1}^\infty$ sampled from $\mathcal{E}_0$, and every integrable function $f$, the posterior mean $\mathbb{E}_\ell[f(\mathcal{E})] = \mathbb{E}[f(\mathcal{E}) \mid \mathcal{D}_\ell]$ converges to $f(\mathcal{E}_0)$. In particular, we conclude that $\mathbb{E}[\Gamma_\ell(\pi_{\mathrm{TS}}^\ell) \mid \mathcal{E}_0]$ tends to $\lambda_{\mathcal{E}_0}^2 H$ in the limit, allowing us to claim that for large enough $\ell$, the expected information ratio $\mathbb{E}[\Gamma_\ell(\pi_{\mathrm{TS}}^\ell)]$ is uniformly bounded by $2\mathbb{E}[\lambda_\mathcal{E}^2]H = 2\lambda^2 H$. As there are $L$ many such ratios, the two bounds together yield $2\sqrt{\lambda^2 HL} \cdot \sqrt{\log(K_{\mathrm{surr}}(\varepsilon))} + L\varepsilon$. This bound is true for large enough $\ell$, giving the additional additive term $T_0$ in the theorem. Since this term is additive, applying Lemma 1 to bound $\log(K_{\mathrm{surr}}(\varepsilon))$, we have successfully shown the asymptotic behavior of the regret, independent of the prior, is of order $\widetilde{O}(H\sqrt{d_{l_1}T})$.

## 5.2 Applications

In each application below, the challenge is to bound $d_{l_1}$ using the specifics of the model, and except for the case of tabular Bayesian RL, such analysis has not been carried out rigorously. We formalize the corollaries and show they are state-of-the-art compared to the literature.

**Tabular RL.** The result below follows from Theorem 3; the main contribution comes from our new information ratio bound, followed by the estimate $\widetilde{O}((\frac{1}{\varepsilon})^{SAH})$ of $K_{\text{surr}}(\varepsilon)$ ([18]).

**Corollary 4.** *Given a tabular Bayesian RL problem, for large enough $T$,*

$$\mathfrak{BR}_L(\pi_{TS}) \leq \widetilde{O}(\lambda\sqrt{HSAT}), \tag{18}$$

*where the polylogarithmic terms are explicitly in terms of $H, S, A, L$.*

We observe that our result matches [31] when their result in the time homogeneous setting (Definition 2) is extended to time inhomogeneous. However, in that paper, the authors assume a Dirichlet based prior which we do not.

**Linear RL.** A previous state-of-the-art $\widetilde{O}(d_f H^{3/2}\sqrt{T})$ was claimed by [18] to hold for linear Bayesian RLs with deterministic reward. We note:

- As in the previous cases, their proof in bounding their information ratio includes a factor of $d_f$, which ours avoids.

- We show that the proof bounding $K_{\text{surr}}(\varepsilon)$ in [18, App. B.4] is incorrect, starting with a wrong application of Cauchy-Schwarz and a wrong mutual information in their definition of information ratio. We provide counterexamples for the estimates found therein to substantiate our claim (see Appendix G.1).

To state our own corollary in this case, we need to define a few notions. Let $d_{l_1}^f = d_{l_1}^{P,f} + d_{l_1}^{R,f}$ be the sum of the $l_1$−dimensions of the feature map space $\{\psi_h^{P,\mathcal{E}}\}_{\mathcal{E}\in\Theta}, \{\psi_h^{R,\mathcal{E}}\}_{\mathcal{E}\in\Theta}$ where the $l_1$−distance between feature maps is defined as $l_1(\psi_h^{\mathcal{E}}, \psi_h^{\mathcal{E}'}) = \int_s \|\psi_h^{\mathcal{E}} - \psi_h^{\mathcal{E}'}\|_1 \mu_{\mathcal{S}}$. Our corollary also provides a concrete bound in the case of *mixture* linear Bayesian RL where the feature maps are themselves a sum of finitely many **fixed** feature maps. This means for all $\mathcal{E} \in \Theta$, we have

$$\psi_h^{P,\mathcal{E}} = \sum_{i=1}^{m_h^P} a_{h,i}^{P,\mathcal{E}} \Psi_{h,i}^P(s), \quad \psi_h^{R,\mathcal{E}} = \sum_{i=1}^{m_h^R} a_{h,i}^{R,\mathcal{E}} \Psi_{h,i}^R(s) \tag{19}$$

where $\{\Psi_{h,i}^P(s)\}_{i=1}^{m_h^P}, \{\Psi_{h,i}^R(s)\}_{i=1}^{m_h^R}$ are finitely many fixed feature maps and $\forall \mathcal{E}, h : \sum_i |a_{h,i}^{P,\mathcal{E}}|^2, \sum_i |a_{h,i}^{R,\mathcal{E}}|^2 \leq C_a$ for some constant $C_a > 0$. Let $M = M^P + M^R = \sum_h m_h^P + \sum_h m_h^R$.

**Corollary 5.** *For a linear Bayesian RL, for large enough $T$,*

$$\mathfrak{BR}_L(\pi_{TS}) \leq \widetilde{O}(\lambda\sqrt{d_{l_1}^f T}). \tag{20}$$

*Given a linear Bayesian RL with finitely many states and total feature space dimension $d_f = d_f^P + d_f^R$, we have $d_{l_1} \leq 2d_f HS$, yielding for large enough $T$,*

$$\mathfrak{BR}_L(\pi_{TS}) \leq \widetilde{O}(\lambda\sqrt{Hd_f ST}). \tag{21}$$

*Given a mixture linear Bayesian RL, for large enough $T$,*

$$\mathfrak{BR}_L(\pi_{TS}) \leq \widetilde{O}(\lambda\sqrt{MT}), \tag{22}$$

The proof is given in Appendix G. The fact that $d_{l_1}$ appears instead of $d_f$ in the general bound is not counter-intuitive, as we should expect the complexity of the feature map space $\{\psi_h^{P,\mathcal{E}}(s)\}_{\mathcal{E}\in\Theta,h\in[H]}, \{\psi_h^{R,\mathcal{E}}(s)\}_{\mathcal{E}\in\Theta,h\in[H]}$ to play a role in the regret, especially as this space can be very complex, and model very different environments that can not be grouped in the same $\varepsilon$−value partition.

Therefore, opposite to the claim made by [18], this complexity can not be captured by simply $d_f$ except maybe in degenerate cases, such as when $\mathcal{S}$ is finite, which is our second statement. More generally, if each feature map $\psi_h^{P,\mathcal{E}}(s), \psi_h^{R,\mathcal{E}}(s)$ can be characterized with a vector of uniformly bounded norm $\boldsymbol{a}_h^{P,\mathcal{E}} \in \mathbb{R}^{m_h^P}, \boldsymbol{a}_h^{R,\mathcal{E}} \in \mathbb{R}^{m_h^R}$, then we can bound the regret in terms of $m_h^P, m_h^R$'s, as is done in Eq. (22) (the finite state case corresponds to $m_h^P = d_f^P S, m_h^R = d_f^R S$).

**Finite mixtures RL.** To state our finite mixtures model result, we need to set the following notations. Let $d_{l_1}^m = d_{l_1}^{m,P} + d_{l_1}^{m,R} = \sum_h d_{l_1,h}^{m,P} + \sum_h d_{l_1,h}^{m,R}$ correspond to the total $l_1-$dimension of the space of mixtures coefficient maps $\{\boldsymbol{a}_h^{P,\mathcal{E}}(s,a)\}_{\mathcal{E}\in\Theta}, \{\boldsymbol{a}_h^{R,\mathcal{E}}(s,a)\}_{\mathcal{E}\in\Theta}$ with $l_1-$ distance defined as $l_1(\boldsymbol{a}_h^{\mathcal{E}}, \boldsymbol{a}_h^{\mathcal{E}'}) = \sup_{s,a} \|\boldsymbol{a}_h^{\mathcal{E}}(s,a) - \boldsymbol{a}_h^{\mathcal{E}'}(s,a)\|_1$. Define also the restricted finite mixtures model where $\boldsymbol{a}_h^{P,\mathcal{E}}, \boldsymbol{a}_h^{R,\mathcal{E}}$ are vectors in $\mathbb{R}^{m_h^P}, \mathbb{R}^{m_h^R}$ independent of $(s,a)$ and let $M = M^P + M^R = \sum_h m_h^P + \sum_h m_h^R$.

**Corollary 6.** *Given a finite mixtures Bayesian RL problem, for large enough $T$,*

$$\mathfrak{BR}_L(\pi_{TS}) \leq \widetilde{O}(\lambda\sqrt{d_{l_1}^m T}). \tag{23}$$

*Assuming the restricted finite mixtures model, for large enough $T$,*

$$\mathfrak{BR}_L(\pi_{TS}) \leq \widetilde{O}\left(\lambda\sqrt{MT}\right). \tag{24}$$

*which, given a uniform dimension $m = m_h^P = m_h^R$, yields $\widetilde{O}(\lambda\sqrt{HmT})$.*

We prove the above in Appendix H, deriving it from our generic bound, after relating the $l_1-$dimension $d_{l_1}$ of the environment space to that of the mixtures coefficients. To the best of our knowledge, this is the first bound for finite mixtures Bayesian RL problems. We note that in a previous work ([6]), a restricted version of finite mixtures, like in Eq. (24), was considered in the frequentist setting.

We finish this section by proposing the following conjecture, in line with [31, Conj. 1].

**Conjecture 7.** *For the Bayesian RL, the following is true and optimal for **all** $T$:*

$$\mathfrak{BR}_L(\pi_{TS}) \leq O\left(\inf_{\varepsilon>0}(\sqrt{H\log(K_{\text{surr}}(\varepsilon))T} + L\varepsilon)\right). \tag{25}$$

*where the constant factor is independent of the prior. This means there exists a Bayesian RL problem such that $\mathfrak{BR}_L(\pi_{TS}) = \widetilde{\Omega}(\sqrt{Hd_{\text{surr}}T})$. All polylogarithmic terms are in terms of $H, d_{\text{surr}}, T$.*

Note that the above coincides with the lower bound for the (model-based) time inhomogeneous frequentist setting; see e.g., [21] for the proven lower bound for the tabular case. This is also $\sqrt{H}$ higher (this factor being baked in $d_{\text{surr}}$) than that of the time homogeneous frequentist setting, which is expected, according to [21, App. D]. Note that in this conjecture, the $\lambda$ in our bound is replaced by $\sqrt{H}$, and the conjecture is not for $T$ large enough, but for all $T$. Supporting this conjecture requires experiments where TS can be exactly implemented assuming access to an oracle which provides the optimal policy for a query environment. Simulations have been performed for the similar [31, Conj. 1] in the time homogeneous case. Our conjecture is similar but with the additional expected factor of $\sqrt{H}$ due to time inhomogeneity, thus their simulation also supports the above.

## 6 Conclusions

In this paper, we have addressed the Bayesian Reinforcement Learning (RL) problem in the context of time inhomogeneous transition and reward functions. By considering both Bayesian transition and Bayesian rewards without prior assumptions, we have extended the scope of previous works, making our formulation more comprehensive. To simplify the learning problem, we have introduced surrogate environments, which discretize the environment space. We have established a connection between the size of this new environment space and the $l_1$-dimensions of the transition and reward functions space, providing insights into the $l_1$-dimension of the environment space denoted by $d_{l_1}$. We have employed posterior consistency tools to analyze the information ratio, which captures the trade-off between exploration and exploitation. We conjecture that (at least a weakened version of) our posterior consistency assumption should hold in general, which is left for future work. Our analysis has resulted in a refined approach to estimate the Bayesian regret in Thompson Sampling (TS), yielding a regret bound of $\widetilde{O}(\lambda\sqrt{d_{l_1}T})$ for large enough time steps $T$. The result is specialized to linear, tabular, and finite mixtures MDPs.

**Limitations:** While the paper provides asymptotic generic regret bound for TS in a generalized setup which improve the state of the art results, finding lower bounds, esp. one dependent on $\lambda$, are left open. In addition, the issue of prior misspecificity is not discussed and left for future studies.

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
