# A    Related works

In the related works section of the main text, we mostly focused on Bayesian regret. Here we include a brief paragraph on bounds on frequentist regret.

For the frequentist setting, various algorithms with provable regret guarantees have been proposed for model-free tabular MDPs. These include UCBVI [7], optimistic Q-learning [21], RLSVI [34, 42], and UCB-Advantage [43]. These algorithms were further generalized to linear or linear mixture MDPs, such as LSVI-UCB [22], OPPO [10], and UCRL-VTR [6, 44]. Slightly more related to our work, model-based frequentist bounds have also been shown for a variant of posterior sampling (PS) in the tabular setting [3]. For the specific variant of *optimistic* PSRL, the optimal bound in the tabular setting with a Dirichlet prior was shown in [39]. To our knowledge, a frequentist bound for PS is still an open problem for general RLs. Minimax regret bounds have also been studied for variants of TS, as in [13]. Most recently, [2] presented VO$Q$L, an algorithm that achieves the optimal bound of $\widetilde{O}(d\sqrt{HT})$ in the general model-free nonlinear setting, where $d$ represents the generalized Eluder dimension of the value function space. Note that the notion of dimension used in our regret bounds is different, and unrelated, to the Eluder dimension used in model-free estimations. For frequentist model-based, the optimal bound was achieved in the tabular setting by [7]. As another research direction, [15] utilized kernel-Hilbert spaces to estimate the value of infinite horizon Markov reward process (MRP) for RL problem, and [1] pave the way for scalability challenges in kernel models.

# B    Proof of Lemma 1

To avoid conflict with the environment space notation $\Theta = \Theta_1 \times \cdots \times \Theta_H$, we adopt the notation $\Theta_k^\varepsilon$ to refer to $\varepsilon-$value partitions.

**Proof of covering number estimate.**
Let $\{B_{h,i}^P(\varepsilon/(2H)^2)\}_{i=1}^{L_h^P(\varepsilon/(2H)^2)}, \{B_{h,i}^R(\varepsilon/(4H))\}_{i=1}^{L_h^R(\varepsilon/(4H))}$ be the $\varepsilon-$balls giving an $\varepsilon-$covering for $\Theta_h^P, \Theta_h^R$. Then define the $\varepsilon-$value partition $\cup_{k=1}^K \Theta_k^\varepsilon = \Theta$ where $K = \prod_h L_h^P(\varepsilon/(2H)^2) \times L_h^R(\varepsilon/(4H))$ as follows. Each $k \in [K]$ can be enumerated as a $2H-$tuple $(i_1, j_1, \ldots, i_H, j_H)$ where $i_h \in [L_h^P(\varepsilon/(2H)^2)], j_h \in [L_h^R(\varepsilon/(4H))]$. Define $\Theta_k^\varepsilon = \{\mathcal{E}|P_h^\mathcal{E} \in B_{h,i_h}^P(\varepsilon/(2H)^2), r_h^\mathcal{E} \in B_{h,j_h}^R(\varepsilon/(4H))\}$. It is straightforward to check that $\cup_k \Theta_k^\varepsilon = \Theta$. Any environment appearing redundantly can be removed from all but one of the $\Theta_k^\varepsilon$'s it lives in, so that we have a true partition of $\Theta_k^\varepsilon$.

Next, we will need to use the following lemma.

Proving that our partition is an $\varepsilon-$value partition requires us to show that for any $\mathcal{E}, \mathcal{E}' \in \Theta_k^\varepsilon$ : $V_{1,\pi_\mathcal{E}^*}^\mathcal{E}(s_1^\ell) - V_{1,\pi_\mathcal{E}^*}^{\mathcal{E}'}(s_1^\ell) \le \varepsilon$. We have

$$V_{1,\pi_\mathcal{E}^*}^\mathcal{E}(s_1^\ell) - V_{1,\pi_\mathcal{E}^*}^{\mathcal{E}'}(s_1^\ell) = \sum_{h=1}^H \mathbb{E}_{\pi_\mathcal{E}^*}^{\mathcal{E}'} \left[ \mathbb{E}_{s' \sim P_h^\mathcal{E}(\cdot|s_h,a_h)}[V_{h+1,\pi_\mathcal{E}^*}^\mathcal{E}(s')] - \mathbb{E}_{s' \sim P_h^{\mathcal{E}'}(\cdot|s_h,a_h)}[V_{h+1,\pi_\mathcal{E}^*}^\mathcal{E}(s')] \right]$$

$$+ \sum_{h=1}^H \mathbb{E}_{\pi_\mathcal{E}^*}^{\mathcal{E}'}[r_h^\mathcal{E}(s_h,a_h) - r_h^{\mathcal{E}'}(s_h,a_h)], \tag{26}$$

Rewrite the first term and bound it as follows:

$$\sum_{h=1}^H \mathbb{E}_{\pi_\mathcal{E}^*}^{\mathcal{E}'} \left[ \int_\mathcal{S} P_h^\mathcal{E}(s'|s_h^\ell,a_h^\ell)V_{h+1,\pi_\mathcal{E}^*}^\mathcal{E}(s') - \int_\mathcal{S} P_h^{\mathcal{E}'}(s'|s_h^\ell,a_h^\ell)V_{h+1,\pi_\mathcal{E}^*}^\mathcal{E}(s') \right]$$

$$\le \sum_{h=1}^H \mathbb{E}_{\pi_\mathcal{E}^*}^{\mathcal{E}'} \left[ \left( \int_\mathcal{S} \left| P_h^\mathcal{E}(s'|s_h^\ell,a_h^\ell) - P_h^{\mathcal{E}'}(s'|s_h^\ell,a_h^\ell) \right| V_{h+1,\pi_\mathcal{E}^*}^\mathcal{E}(s') \right) \right]. \tag{27}$$

where integrals are with respect to the measure on $\mathcal{S}$. Then, we can bound probability transitions terms by

$$
\sum_{h=1}^{H} \mathbb{E}_{\pi_{\mathcal{E}}^*}^{\mathcal{E}'} \left[ \left( \int_{\mathcal{S}} \left| P_h^{\mathcal{E}}(s'|s_h^{\ell}, a_h^{\ell}) - P_h^{\mathcal{E}'}(s'|s_h^{\ell}, a_h^{\ell}) \right| V_{h+1, \pi_{\mathcal{E}}^*}^{\mathcal{E}}(s') \right) \right]
$$

$$
\leq H \sum_{h=1}^{H} \mathbb{E}_{\pi_{\mathcal{E}}^*}^{\mathcal{E}'} \left[ \int_{\mathcal{S}} \left| P_h^{\mathcal{E}}(s'|s_h^{\ell}, a_h^{\ell}) - P_h^{\mathcal{E}'}(s'|s_h^{\ell}, a_h^{\ell}) \right| \right]
$$

$$
\leq H \sum_{h=1}^{H} \sup_{s,a} \left( \int_{\mathcal{S}} \left| P_h^{\mathcal{E}}(s'|s, a) - P_h^{\mathcal{E}'}(s'|s, a) \right| \right) \tag{28}
$$

$$
= H \sum_{h=1}^{H} l_1(P_h^{\mathcal{E}}, P_h^{\mathcal{E}'}) \leq H \left( 2 \frac{\varepsilon}{4H^2} \cdot H \right)
$$

and similarly, reward terms by

$$
\sum_{h=1}^{H} E_{\pi_{\mathcal{E}}^*}^{\mathcal{E}'} [r_h^{\mathcal{E}}(s_h^{\ell}, a_h^{\ell}) - r_h^{\mathcal{E}'}(s_h^{\ell}, a_h^{\ell})] = \sum_{h=1}^{H} E_{\pi_{\mathcal{E}}^*}^{\mathcal{E}'} \left[ \int_0^1 x \left( r_h^{\mathcal{E}}(x|s_h^{\ell}, a_h^{\ell}) - r_h^{\mathcal{E}'}(x|s_h^{\ell}, a_h^{\ell}) \right) dx \right]
$$

$$
\leq \sum_{h=1}^{H} E_{\pi_{\mathcal{E}}^*}^{\mathcal{E}'} \left[ \int_0^1 \left| x \left( r_h^{\mathcal{E}}(x|s_h^{\ell}, a_h^{\ell}) - r_h^{\mathcal{E}'}(x|s_h^{\ell}, a_h^{\ell}) \right) \right| dx \right]
$$

$$
\leq \sum_{h=1}^{H} \sup_{s,a} \int_0^1 \left| x \left( r_h^{\mathcal{E}}(x|s, a) - r_h^{\mathcal{E}'}(x|s, a) \right) \right| dx \tag{29}
$$

$$
= \sum_{h=1}^{H} l_1(r_h^{\mathcal{E}}, r_h^{\mathcal{E}'}) \leq H \left( 2 \frac{\varepsilon}{4H} \right)
$$

See also Remark 7 for the reward term bound. In the first inequality, we used that $V_{h+1, \pi_{\mathcal{E}}^*}^{\mathcal{E}}(s')$ is always bounded by $H$, and in both cases we used the fact that the transition and reward functions of $\mathcal{E}, \mathcal{E}'$ live inside the same balls, with their $l_1$ distance being at most twice the radii $\varepsilon/(2H)^2$ and $\varepsilon/(4H)$, respectively. Adding up the above two estimates equals $\varepsilon$, as desired. This shows that our $\Theta_k^{\varepsilon}$ partition is an $\varepsilon-$value partition, hence $K_{\mathrm{surr}}(\varepsilon) \leq K = \prod_h L_h^P(\varepsilon/(2H)^2) \times L_h^R(\varepsilon/(4H))$.

*Remark* 7. Notice that the $l_1-$distance of two reward functions is over their probability distributions, and is larger than their expected norm difference, i.e.,

$$
l_1(r, r') = \sup_{s,a \in \mathcal{S} \times \mathcal{A}} ||r(\cdot|s, a) - r'(\cdot|s, a)||_1 = \sup_{s,a \in \mathcal{S} \times \mathcal{A}} \int_0^1 |r(x|s, a) - r'(x|s, a)|
$$

$$
\geq \sup_{s,a \in \mathcal{S} \times \mathcal{A}} \int_0^1 x |r(x|s, a) - r'(x|s, a)| = \sup_{s,a \in \mathcal{S} \times \mathcal{A}} \mathbb{E}[|r(s, a) - r'(s, a)|]. \tag{30}
$$

**Proof of $d_{\mathrm{surr}} \leq d_{l_1}$.**
By taking the log, dividing by $\log(1/\varepsilon)$, and taking the $\limsup$ of both sides of this inequality, we can infer the second statement of the lemma:

$$
d_{\mathrm{surr}} = \limsup_{\varepsilon \to 0} \frac{\log(K_{\mathrm{surr}}(\varepsilon))}{\log(1/\varepsilon)}
$$

$$
\leq \sum_{h=1}^{H} \limsup_{\varepsilon \to 0} \frac{\log(L_h^P(\varepsilon/(2H)^2))}{\log(\frac{1}{\varepsilon/(2H)^2})} \cdot \frac{1}{1 + \log((2H)^2)/\log(\varepsilon/(2H)^2)} +
$$

$$
\sum_{h=1}^{H} \limsup_{\varepsilon \to 0} \frac{\log(L_h^R(\varepsilon/(4H)))}{\log(\frac{1}{\varepsilon/(2H)^2})} \cdot \frac{1}{1 + \log(4H)/\log(\varepsilon/(4H))} \tag{31}
$$

$$
= \sum_h d_{l_1, h}^P + \sum_h d_{l_1, h}^R = d_{l_1}.
$$

*Fact* 1. We separate the statement proved in Eqs. (28) and (29) as fact, useful for future use: for all $\mathcal{E}, \mathcal{E}' \in \Theta$,

$$V_{1,\pi_{\mathcal{E}}^*}^{\mathcal{E}}(s_1^\ell) - V_{1,\pi_{\mathcal{E}}^*}^{\mathcal{E}'}(s_1^\ell) \leq H \sum_{h=1}^{H} l_1(P_h^{\mathcal{E}}, P_h^{\mathcal{E}'}) + \sum_{h=1}^{H} l_1(r_h^{\mathcal{E}}, r_h^{\mathcal{E}'}) \,. \tag{32}$$

## C  Proof of Lemma 2

While we follow the proof of the same lemma in [18, App. B.1], we will need to correct some mistakes. Let us restate the equation of the statement:

**Lemma.** *For any $\varepsilon-$value partition and any $\ell \in [L]$, there are random environments $\tilde{\mathcal{E}}_\ell^* \in \Theta$ with their laws only depending on $\zeta, \mathcal{D}_\ell$, such that*

$$\mathbb{E}_\ell \left[ V_{1,\pi_{\mathcal{E}}^*}^{\mathcal{E}}(s_1^\ell) - V_{1,\pi_{TS}^\ell}^{\mathcal{E}}(s_1^\ell) \right] - \mathbb{E}_\ell \left[ V_{1,\pi_{\mathcal{E}}^*}^{\tilde{\mathcal{E}}_\ell^*}(s_1^\ell) - V_{1,\pi_{TS}^\ell}^{\tilde{\mathcal{E}}_\ell^*}(s_1^\ell) \right] \leq \varepsilon \,. \tag{33}$$

*Proof.* Assume a partition $\Theta_k^\varepsilon$ satisfying Definition 8 with error $\varepsilon$ exists. Let $\mathcal{E}_\ell \sim P(\cdot|\mathcal{D}_\ell)$. We want to decompose $\mathbb{E}_\ell \left[ V_{1,\pi_{TS}^\ell}^{\mathcal{E}_\ell}(s_1^\ell) \big| \mathcal{E}_\ell \in \Theta_k^\varepsilon \right]$, where the expectation is over all $\mathcal{E}_\ell \in \Theta_k^\varepsilon$ and all $\pi_{TS}^\ell = \pi_{\mathcal{E}_\ell'}^*$, where $\mathcal{E}_\ell' \in \Theta$, with $\mathcal{E}_\ell'$ independent of $\mathcal{E}_\ell$. We decompose this by writing the expectation only over the former.

$$\begin{aligned} \mathbb{E}_\ell \left[ V_{1,\pi_{TS}^\ell}^{\mathcal{E}_\ell}(s_1^\ell) \big| \mathcal{E}_\ell \in \Theta_k^\varepsilon \right] &= \int_{\mathcal{E}_0 \in \Theta_k^\varepsilon} \mathbb{P}\left( \mathcal{E}_\ell = \mathcal{E}_0 | \mathcal{E}_\ell \in \Theta_k^\varepsilon \right) \mathbb{E}_\ell \left[ V_{1,\pi_{TS}^\ell}^{\mathcal{E}_0}(s_1^\ell) \big| \mathcal{E}_\ell \in \Theta_k^\varepsilon \right] \mathrm{d}\rho(\mathcal{E}_0) \\ &= \int_{\mathcal{E}_0 \in \Theta_k^\varepsilon} \mathbb{P}\left( \mathcal{E}_\ell = \mathcal{E}_0 | \mathcal{E}_\ell \in \Theta_k^\varepsilon \right) \mathbb{E}_\ell \left[ V_{1,\pi_{TS}^\ell}^{\mathcal{E}_0}(s_1^\ell) \right] \mathrm{d}\rho(\mathcal{E}_0) \,, \end{aligned} \tag{34}$$

where the last equation is due to the independence between $\mathcal{E}_\ell$ and $\mathcal{E}_\ell'$.

We would like to find some $\tilde{\mathcal{E}}_{k,\ell}^*$ such that its corresponding expected value is smaller than the integral above, i.e.

$$\mathbb{E}_\ell \left[ V_{1,\pi_{TS}^\ell}^{\tilde{\mathcal{E}}_{k,\ell}^*}(s_1^\ell) \right] \leq \int_{\mathcal{E}_0 \in \Theta_k^\varepsilon} \mathbb{P}\left( \mathcal{E}_\ell = \mathcal{E}_0 | \mathcal{E}_\ell \in \Theta_k^\varepsilon \right) \mathbb{E}_\ell \left[ V_{1,\pi_{TS}^\ell}^{\mathcal{E}_0}(s_1^\ell) \right] \mathrm{d}\rho(\mathcal{E}_0) \tag{35}$$

We set values of this random variable as $\tilde{\mathcal{E}}_{k,\ell}^* = \mathbb{E}_\ell[\mathcal{E}|\mathcal{E} \in \Theta_k^\varepsilon], \forall k \in [K_{\mathrm{surr}}(\varepsilon)]$. In other words, the posterior mean over $\Theta_k^\varepsilon$. Now, we can define a new random environment $\tilde{\mathcal{E}}_\ell^*$ which prior can be easily computed as $\mathbb{P}_\ell \left( \tilde{\mathcal{E}}_\ell^* = \tilde{\mathcal{E}}_{k,\ell}^* \right) = \mathbb{P}_\ell(\mathcal{E} \in \Theta_k^\varepsilon)$ and $\zeta(\tilde{\mathcal{E}}_{k,\ell}^*) = k$. Note the law of $\tilde{\mathcal{E}}_\ell^*$ only depends on $\zeta \in [K]$ and $\mathcal{D}_\ell$, and conditional on $\zeta$, $\tilde{\mathcal{E}}_\ell^*$ is independent of $\mathcal{E}$, as desired. While this definition of $\tilde{\mathcal{E}}_\ell^*$ may not be in $\Theta$, that is fine as we do not use this condition in our proof (see also Remark 8).

Notice that as a result, the overall posterior mean of $\bar{\mathcal{E}}_\ell^* := \mathbb{E}_\ell[\tilde{\mathcal{E}}_\ell^*]$ coincides with $\bar{\mathcal{E}}_\ell$. Also note that by this definition of $\tilde{\mathcal{E}}_{k,\ell}^*$, and the independence over layers even after conditioning on $\Theta_k^\varepsilon$, the mean $\mathbb{E}_\ell[V_{1,\pi_{TS}^\ell}^{\tilde{\mathcal{E}}_{k,\ell}^*}(s_1^\ell)]$ is in fact equal to the mean on the right hand side of Eq. (35), thus our construction is sharp in that it satisfies this inequality with an equality.

Now, we are ready to make the connection with Eq. (11), showing that solving the surrogate environment problem is 'almost the same' as solving the original problem, up to some $\varepsilon$. Integrating over the different values of $k$ with prior $\mathbb{P}_\ell(\mathcal{E} \in \Theta_k^\varepsilon)$,

$$\mathbb{E}_\ell \left[ V_{1,\pi_{TS}^\ell}^{\tilde{\mathcal{E}}_\ell^*}(s_1^\ell) \right] \leq \mathbb{E}_\ell \left[ V_{1,\pi_{TS}^\ell}^{\mathcal{E}}(s_1^\ell) \right] \implies \mathbb{E}_\ell \left[ V_{1,\pi_{TS}^\ell}^{\tilde{\mathcal{E}}_\ell^*}(s_1^\ell) \right] - \mathbb{E}_\ell \left[ V_{1,\pi_{TS}^\ell}^{\mathcal{E}}(s_1^\ell) \right] \leq 0 \tag{36}$$

Lastly, by the partition property,

$$\mathbb{E}_\ell \left[ V_{1,\pi_{\mathcal{E}}^*}^{\mathcal{E}}(s_1^\ell) - V_{1,\pi_{\mathcal{E}}^*}^{\tilde{\mathcal{E}}_\ell^*}(s_1^\ell) \right] \leq \varepsilon \,. \tag{37}$$

where in the expectation above, $\zeta(\tilde{\mathcal{E}}_\ell^*) = \zeta(\mathcal{E})$, i.e., $\mathcal{E}$ and $\tilde{\mathcal{E}}_\ell^*$ are clearly independent only after conditioning on $\zeta = k$, as also required in Eq. (11). Adding the above two inequalities gives

$$\mathbb{E}_\ell \left[ V_{1,\pi_{\mathcal{E}}^*}^{\mathcal{E}}(s_1^\ell) - V_{1,\pi_{TS}^\ell}^{\mathcal{E}}(s_1^\ell) \right] - \mathbb{E}_\ell \left[ V_{1,\pi_{\mathcal{E}}^*}^{\tilde{\mathcal{E}}_\ell^*}(s_1^\ell) - V_{1,\pi_{TS}^\ell}^{\tilde{\mathcal{E}}_\ell^*}(s_1^\ell) \right] \leq \varepsilon \tag{38}$$

finishing the proof. Notice that Eq. (37) is the only place where the property of $\varepsilon-$value partitioning is used. $\qquad\square$

*Remark* 8. We note that if $\tilde{\mathcal{E}}_\ell^*$ is any convex combination of environments in $\Theta_{\zeta(\mathcal{E})}$, then by the $\varepsilon-$value partition property $\mathbb{E}_\ell[V_{1,\pi_{\mathcal{E}}^*}^{\mathcal{E}}(s_1^\ell) - V_{1,\pi_{\mathcal{E}}^*}^{\tilde{\mathcal{E}}_\ell^*}(s_1^\ell)] \leq \varepsilon$. The distinct property that surrogate environments satisfy is

$$\mathbb{E}_\ell[V_{1,\pi_{\mathrm{TS}}^\ell}^{\tilde{\mathcal{E}}_{k,\ell}^*}(s_1^\ell)] \leq \mathbb{E}_\ell[V_{1,\pi_{\mathrm{TS}}^\ell}^{\mathcal{E}}(s_1^\ell)|\mathcal{E} \in \Theta_k^\varepsilon], \ \forall k \in [K]. \tag{39}$$

and therefore, satisfy Eq. (33).

*Remark* 9. For the purpose of fully addressing the issue present in [18] in proving the above lemma, we show an alternative construction which does not assume a layered $\varepsilon-$value partition. One can always find a decreasing sequence $\mathbb{E}_\ell\left[V_{1,\pi_{\mathrm{TS}}^\ell}^{\mathcal{E}^1}(s_1^\ell)\right] \geq \mathbb{E}_\ell\left[V_{1,\pi_{\mathrm{TS}}^\ell}^{\mathcal{E}^2}(s_1^\ell)\right] \geq \ldots$ with limit $\inf_{\mathcal{E}_0 \in \Theta_k^\varepsilon} \mathbb{E}_\ell\left[V_{1,\pi_{\mathrm{TS}}^\ell}^{\mathcal{E}_0}(s_1^\ell)\right]$. Then we claim there exists some $J$ such that the above inequality is true for $\tilde{\mathcal{E}}_{k,\ell}^* = \mathcal{E}^J$. Otherwise, we have

$$\mathbb{E}_\ell\left[V_{1,\pi_{\mathrm{TS}}^\ell}^{\mathcal{E}^i}(s_1^\ell)\right] > \int_{\mathcal{E}_0 \in \Theta_k^\varepsilon} \mathbb{P}\left(\mathcal{E}_\ell = \mathcal{E}_0 | \mathcal{E}_\ell \in \Theta_k^\varepsilon\right) \mathbb{E}_\ell\left[V_{1,\pi_{\mathrm{TS}}^\ell}^{\mathcal{E}_0}(s_1^\ell)\right] \mathrm{d}\rho(\mathcal{E}_0) \tag{40}$$

for all $i$. Taking the limit $i \to \infty$, we get

$$\inf_{\mathcal{E}_0 \in \Theta_k^\varepsilon} \mathbb{E}_\ell\left[V_{1,\pi_{\mathrm{TS}}^\ell}^{\mathcal{E}_0}(s_1^\ell)\right] \geq \int_{\mathcal{E}_0 \in \Theta_k^\varepsilon} \mathbb{P}\left(\mathcal{E}_\ell = \mathcal{E}_0 | \mathcal{E}_\ell \in \Theta_k^\varepsilon\right) \mathbb{E}_\ell\left[V_{1,\pi_{\mathrm{TS}}^\ell}^{\mathcal{E}_0}(s_1^\ell)\right] \mathrm{d}\rho(\mathcal{E}_0) \tag{41}$$

which can only be true if $\mathbb{E}_\ell\left[V_{1,\pi_{\mathrm{TS}}^\ell}^{\mathcal{E}_0}(s_1^\ell)\right] = \inf_{\mathcal{E}_0 \in \Theta_k^\varepsilon} \mathbb{E}_\ell\left[V_{1,\pi_{\mathrm{TS}}^\ell}^{\mathcal{E}_0}(s_1^\ell)\right]$ almost everywhere. Any $\mathcal{E}_0$ satisfying this equality would also satisfy our requirement for $\tilde{\mathcal{E}}_{k,\ell}^*$. It is important to note that $\tilde{\mathcal{E}}_{k,\ell}^*$ depends on both $k$ and $\mathcal{D}_\ell$, as mentioned in the lemma's statement. The finishes the construction of $\tilde{\mathcal{E}}_\ell^*$ and the rest follows similar to the above proof.

*Remark* 10 (Incorrect proof by [18] of the lemma). Eq. (34) is different from what appears in [18, App. B.1], where we have corrected for the abuse of notation of $\mathcal{E}$ occurring in e.g. "$\mathbb{P}\left(\mathcal{E}_\ell = \mathcal{E}|\mathcal{E}_\ell \in \Theta_k^\varepsilon\right) \mathbb{E}_\ell\left[V_{1,\pi_{\mathcal{E}}^*}^{\mathcal{E}}(s_1^\ell)\right]$". We further note the use of summation over $\mathcal{E} \in \Theta_k^\varepsilon$ in their equation, instead of an integral which is required even in the tabular case. This is not easily fixed by just replacing sum with integral, since the application of [18, Lemma D.1] used afterwards in their proof depends on having a finite sum. We mention this lemma below:

(Lemma 1 in [14]) Let $\{a_i\}_{i=1}^N$ and $\{b_i\}_{i=1}^N$ be two sequences of real numbers, where $N < \infty$. Let $\{p_i\}_{i=1}^N$ be such that $p_i \geq 0$ for all $i$ and $\sum_{i=1}^N p_i = 1$. Then there exists indices $j, k \in [N]$ and $r \in [0, 1]$ such that

$$ra_j + (1-r)a_k \leq \sum_{i=1}^N a_i p_i, rb_j + (1-r)b_k \leq \sum_{i=1}^L b_i p_i \,.$$

The application of this lemma is key to the proof as the authors cite it to find the right surrogate environments that satisfy Eq. (39). However, even in this application, it is not clear what exactly is the 'second' set of numbers ($b_j$ above), as is required for that Lemma to be a nontrivial result, which makes its usage even more questionable.

# D  Proof of Theorem 3

We restate the theorem for ease of reference as it contains multiple statements.

**Theorem.** *Given a Bayesian RL problem, we have*

$$\mathfrak{BR}_L(\pi_{TS}) \leq \inf_{\varepsilon > 0}\left(\lambda\sqrt{\log(K_{\mathrm{surr}}(\varepsilon))T} + L\varepsilon\right) + T_0 \tag{42}$$

*where $T_0$ does not depend on $T$. This can be further upper bounded by*

$$\mathfrak{BR}_L(\pi_{TS}) \leq \widetilde{O}(\lambda\sqrt{d_{l_1}T})\,. \tag{43}$$

*for large enough $T$. Given a homogeneous $l_1$ dimension $d_{\mathrm{hom}} = d_{l_1,h}, \forall h$, this simplifies to*

$$\mathfrak{BR}_L(\pi_{TS}) \leq \widetilde{O}(\lambda\sqrt{Hd_{\mathrm{hom}}T})\,. \tag{44}$$

*Proof.* The proof starts by employing the surrogate environment learning bound from Appendix C:

$$\mathbb{E}_\ell\left[V_{1,\pi_{\mathcal{E}}^*}^{\mathcal{E}}(s_1^\ell) - V_{1,\pi_{\mathrm{TS}}^\ell}^{\mathcal{E}}(s_1^\ell)\right] - \varepsilon \le \mathbb{E}_\ell\left[V_{1,\pi_{\mathcal{E}}^*}^{\tilde{\mathcal{E}}_\ell^*}(s_1^\ell) - V_{1,\pi_{\mathrm{TS}}^\ell}^{\tilde{\mathcal{E}}_\ell^*}(s_1^\ell)\right]. \tag{45}$$

We have

$$\begin{aligned}
\mathfrak{BR}_L(\pi_{\mathrm{TS}}) &= \sum_{\ell=1}^L \mathbb{E}\left[\mathbb{E}_\ell\left[V_{1,\pi_{\mathcal{E}}^*}^{\mathcal{E}}(s_1^\ell) - V_{1,\pi_{\mathrm{TS}}^\ell}^{\mathcal{E}}(s_1^\ell)\right]\right] \\
&= \sum_{\ell=1}^L \mathbb{E}\left[\mathbb{E}_\ell\left[V_{1,\pi_{\mathcal{E}}^*}^{\mathcal{E}}(s_1^\ell) - V_{1,\pi_{\mathrm{TS}}^\ell}^{\mathcal{E}}(s_1^\ell)\right] - \varepsilon\right] + L\varepsilon \\
&= \sum_{\ell=1}^L \mathbb{E}\left[\mathbb{E}_\ell\left[V_{1,\pi_{\mathcal{E}}^*}^{\tilde{\mathcal{E}}_\ell^*}(s_1^\ell) - V_{1,\pi_{\mathrm{TS}}^\ell}^{\tilde{\mathcal{E}}_\ell^*}(s_1^\ell)\right]\right] + L\varepsilon
\end{aligned} \tag{46}$$

Given that $\pi_{\mathrm{TS}}^\ell$ is independent from $\tilde{\mathcal{E}}_\ell^*$ and the independence of the latter's prior over different layers (due to the layered $\varepsilon-$value partition), we observe that $\mathbb{E}_\ell[V_{1,\pi_{\mathrm{TS}}^\ell}^{\tilde{\mathcal{E}}_\ell^*}(s_1^\ell)] = \mathbb{E}_\ell[V_{1,\pi_{\mathrm{TS}}^\ell}^{\bar{\mathcal{E}}_\ell^*}(s_1^\ell)]$. Again, since $\pi_{\mathrm{TS}}^\ell$ is also independent from $\bar{\mathcal{E}}_\ell^*$, and that $\pi_{\mathrm{TS}}^\ell$ and $\mathcal{E}$ have the same laws conditional on $\mathcal{D}_\ell$, we can rewrite the latter as $\mathbb{E}_\ell[V_{1,\pi_{\mathcal{E}}^*}^{\bar{\mathcal{E}}_\ell^*}(s_1^\ell)]$. This comes with the obvious note that $\bar{\mathcal{E}}_\ell^*$ and $\mathcal{E}$ are independent, in contrast to $\tilde{\mathcal{E}}_\ell^*$ and $\mathcal{E}$ that are dependent through $\zeta$. This allows us to rewrite

$$\mathbb{E}_\ell\left[V_{1,\pi_{\mathcal{E}}^*}^{\tilde{\mathcal{E}}_\ell^*}(s_1^\ell) - V_{1,\pi_{\mathrm{TS}}^\ell}^{\tilde{\mathcal{E}}_\ell^*}(s_1^\ell)\right] = \mathbb{E}_\ell\left[V_{1,\pi_{\mathcal{E}}^*}^{\tilde{\mathcal{E}}_\ell^*}(s_1^\ell) - V_{1,\pi_{\mathcal{E}}^*}^{\bar{\mathcal{E}}_\ell^*}(s_1^\ell)\right] \tag{47}$$

Due to our construction of $\bar{\mathcal{E}}_\ell^*$ in Appendix C, we can substitute $\bar{\mathcal{E}}_\ell^* = \bar{\mathcal{E}}_\ell = \mathbb{E}_\ell[\mathcal{E}]$. Using Lemma 8, we can rewrite the above mean as

$$\begin{aligned}
= \sum_{h=1}^H \mathbb{E}_\ell\Bigg[\mathbb{E}_{\pi_{\mathcal{E}}^*}^{\bar{\mathcal{E}}_\ell}\Bigg[&\mathbb{E}_{(s',r')\sim(P_h^{\tilde{\mathcal{E}}_\ell^*}\otimes r_h^{\tilde{\mathcal{E}}_\ell^*})(\cdot|s_h,a_h)}[r' + V_{h+1,\pi_{\mathcal{E}}^*}^{\tilde{\mathcal{E}}_\ell^*}(s')] \\
&-\mathbb{E}_{(s',r')\sim(P_h^{\bar{\mathcal{E}}_\ell}\otimes r_h^{\bar{\mathcal{E}}_\ell})(\cdot|s_h,a_h)}[r' + V_{h+1,\pi_{\mathcal{E}}^*}^{\tilde{\mathcal{E}}_\ell^*}(s')]\Bigg]\Bigg].
\end{aligned} \tag{48}$$

Denoting

$$\begin{aligned}
\Delta_h^{\tilde{\mathcal{E}}_\ell^*}(s_h,a_h) := &\mathbb{E}_{(s',r')\sim(P_h^{\tilde{\mathcal{E}}_\ell^*}\otimes r_h^{\tilde{\mathcal{E}}_\ell^*})(\cdot|s_h,a_h)}[r' + V_{h+1,\pi_{\mathcal{E}}^*}^{\tilde{\mathcal{E}}_\ell^*}(s')] \\
&- \mathbb{E}_{(s',r')\sim(P_h^{\bar{\mathcal{E}}_\ell}\otimes r_h^{\bar{\mathcal{E}}_\ell})(\cdot|s_h,a_h)}[r' + V_{h+1,\pi_{\mathcal{E}}^*}^{\tilde{\mathcal{E}}_\ell^*}(s')],
\end{aligned} \tag{49}$$

we have

$$= \sum_{h=1}^H \mathbb{E}_\ell\left[\int_{s,a} d_{h,\pi_{\mathcal{E}}^*}^{\bar{\mathcal{E}}_\ell}(s,a)\Delta_h^{\tilde{\mathcal{E}}_\ell^*}(s,a)\,\mathrm{d}\mu_{\mathcal{S}\times\mathcal{A}}\right]. \tag{50}$$

Let $\mathcal{B}_\ell := \{(s,a,h) \mid \mathbb{E}_\ell\left[d_{h,\pi_{\mathcal{E}}^*}^{\bar{\mathcal{E}}_\ell}(s,a)\right] \ne 0\}$ and let $\int_{(s,a,h)} := \sum_h \int_{(s,a)}$ denote the integral over the space $[H] \times \mathcal{S} \times \mathcal{A}$ where we use the product of counting measure on $[H]$ and $\mu_{\mathcal{S}\times\mathcal{A}}$. We apply Cauchy-Schwarz inequality using the similar technique in [18, App. A.2], which we modify to

include the value diameter (Definition 9). Since $\Delta_h^{\tilde{\mathcal{E}}_\ell^*}(s,a) \le 2H$, we have

$$
\mathbb{E}_\ell\left[\int_{(s,a,h)} d_{h,\pi^*}^{\bar{\mathcal{E}}_\ell}(s,a)\Delta_h^{\tilde{\mathcal{E}}_\ell^*}(s,a)\right]
$$

$$
= \mathbb{E}_\ell\left[\int_{(s,a,h)\notin\mathcal{B}_\ell} d_{h,\pi^*}^{\bar{\mathcal{E}}_\ell}(s,a)\Delta_h^{\tilde{\mathcal{E}}_\ell^*}(s,a)\right] + \mathbb{E}_\ell\left[\int_{(s,a,h)\in\mathcal{B}_\ell} d_{h,\pi^*}^{\bar{\mathcal{E}}_\ell}(s,a)\Delta_h^{\tilde{\mathcal{E}}_\ell^*}(s,a)\right]
$$

$$
\le 2H\mathbb{E}_\ell\left[\int_{(s,a,h)\notin\mathcal{B}_\ell} d_{h,\pi^*}^{\bar{\mathcal{E}}_\ell}(s,a)\right] + \mathbb{E}_\ell\left[\int_{(s,a,h)\in\mathcal{B}_\ell} d_{h,\pi^*}^{\bar{\mathcal{E}}_\ell}(s,a)\Delta_h^{\tilde{\mathcal{E}}_\ell^*}(s,a)\right]
$$

$$
= \mathbb{E}_\ell\left[\int_{(s,a,h)\in\mathcal{B}_\ell} d_{h,\pi^*}^{\bar{\mathcal{E}}_\ell}(s,a)\Delta_h^{\tilde{\mathcal{E}}_\ell^*}(s,a)\right]
$$

$$
= \mathbb{E}_\ell\left[\int_{(s,a,h)\in\mathcal{B}_\ell} \frac{\lambda_\mathcal{E} d_{h,\pi^*}^{\bar{\mathcal{E}}_\ell}(s,a)}{\mathbb{E}_\ell\left[d_{h,\pi^*}^{\bar{\mathcal{E}}_\ell}(s,a)\right]^{1/2}}\mathbb{E}_\ell\left[d_{h,\pi^*}^{\bar{\mathcal{E}}_\ell}(s,a)\right]^{1/2}\frac{\Delta_h^{\tilde{\mathcal{E}}_\ell^*}(s,a)}{\lambda_\mathcal{E}}\right]
$$

$$
\le \left(\mathbb{E}_\ell\left[\int_{(s,a,h)\in\mathcal{B}_\ell} \frac{(\lambda_\mathcal{E} d_{h,\pi^*}^{\bar{\mathcal{E}}_\ell}(s,a))^2}{\mathbb{E}_\ell\left[d_{h,\pi^*}^{\bar{\mathcal{E}}_\ell}(s,a)\right]}\right]\right)^{1/2}
$$

$$
\left(\mathbb{E}_\ell\left[\int_{(s,a,h)\in\mathcal{B}_\ell} \mathbb{E}_\ell\left[d_{h,\pi^*}^{\bar{\mathcal{E}}_\ell}(s,a)\right](\frac{\Delta_h^{\tilde{\mathcal{E}}_\ell^*}(s,a)}{\lambda_\mathcal{E}})^2\right]\right)^{1/2}
$$

$$
\le \left(\int_{(s,a,h)\in\mathcal{B}_\ell} \frac{\mathbb{E}_\ell\left[(\lambda_\mathcal{E} d_{h,\pi^*}^{\bar{\mathcal{E}}_\ell}(s,a))^2\right]}{\mathbb{E}_\ell\left[d_{h,\pi^*}^{\bar{\mathcal{E}}_\ell}(s,a)\right]}\right)^{1/2} \left(\mathbb{E}_\ell\left[\int_{(s,a,h)} \mathbb{E}_\ell\left[d_{h,\pi^*}^{\bar{\mathcal{E}}_\ell}(s,a)\right](\frac{\Delta_h^{\tilde{\mathcal{E}}_\ell^*}(s,a)}{\lambda_\mathcal{E}})^2\right]\right)^{1/2}
$$

$$
= \sqrt{\mathcal{T}^\ell \cdot \mathcal{I}^\ell},
$$

where we used $\mathcal{T}^\ell$ and $\mathcal{I}^\ell$ to denote the first and the second term respectively. Note that the total regret of each episode is at most $H$. Therefore, going back to the Bayesian regret formulationa and using Cauchy-Schwarz, we get

$$
\mathfrak{BR}_L(\pi_{\mathrm{TS}}) \le \mathbb{E}\left[\sum_{\ell=L_0+1}^{L} \sqrt{\mathcal{T}^\ell \cdot \mathcal{I}^\ell}\right] + L_0 H + L\varepsilon \tag{51}
$$

$$
\le \mathbb{E}\left[(\sum_{\ell=L_0+1}^{L} \mathcal{T}^\ell)^{1/2} \cdot (\sum_{\ell=L_0+1}^{L} \mathcal{I}^\ell)^{1/2}\right] + L_0 H + L\varepsilon \tag{52}
$$

$$
\le \sqrt{\mathbb{E}\left[\sum_{\ell=L_0+1}^{L} \mathcal{T}^\ell\right] \cdot \mathbb{E}\left[\sum_{\ell=L_0+1}^{L} \mathcal{I}^\ell\right]} + L_0 H + L\varepsilon \tag{53}
$$

$$
\le \sqrt{L\left(\sup_{L_0+1\le\ell\le L}\mathbb{E}\left[\mathcal{T}^\ell\right]\right) \cdot \mathbb{E}\left[\sum_{\ell=1}^{L}\mathcal{I}^\ell\right]} + L_0 H + L\varepsilon, \tag{54}
$$

for every $0 \le L_0 < L$. We estimate each term separately in Appendix F and Appendix E.

*Remark* 11. While the spirit of the argument, in applying surrogate learning coupled with information ratio and Cauchy Schwarz is similar to [18], the technical aspects are different for estimating $\sum_\ell \mathcal{I}^\ell$, and more importantly, the entire analysis is different for $\sum_\ell \mathcal{T}^\ell$, as can be observed in what follows.

We gather the results to finish the proof. In Appendix E, we show that $\mathbb{E}[\sum_\ell \mathcal{I}^\ell] \le \frac{1}{2}\log(K_{\mathrm{surr}}(\varepsilon))$.

In Appendix F, we show that $\limsup \mathbb{E}\left[\mathcal{T}^\ell\right]$ is bounded by $\lambda^2 H$. Thus $\sup_{L_0+1\leq\ell\leq L}\mathbb{E}\left[\mathcal{T}^\ell\right] \leq 2\lambda^2 H$ for large enough $L_0 > 0$. Hence,

$$\mathfrak{BR}_L(\pi_{\text{TS}}) \leq \lambda\sqrt{\log(K_{\text{surr}}(\varepsilon))T} + L\varepsilon + T_0 \tag{55}$$

for all $\varepsilon > 0$, where $T_0 = L_0 H$. Taking the infimum over $\varepsilon$ gives the desired regret bound. The next statement of the theorem in Eq. (43) is an application of Lemma 1 followed by selecting $\varepsilon = 1/L$. Notice that due to nonzero $\varepsilon$ effects, polylogarithmic terms in $H, L$ are picked up when comparing $d_{l_1}$ with $d_{l_1}(\varepsilon)$. More precisely, according to Lemma 1, one compares $\log(K_{\text{surr}}(\varepsilon))$ with $\sum_h \log(L_h^P(\varepsilon/(2H)^2)) + \log(L_h^R(\varepsilon/(4H)))$. From the definition of $d_{l_1,h}^P$ we have $\log(L_h^P(\varepsilon/(2H)^2)) \sim O(d_{l_1,h}^P \log((2H)^2/\varepsilon))$, so this includes a logarithmic factor of $H^2$, and choosing $\varepsilon = 1/L$ means a logarithmic factor of $H^2 L$. A similar argument can be made for $L_h^R$.

Finally, the last statement of the theorem follows by definition: $d_{l_1}L = (\sum_h d_{l_1,h})L = d_{\text{hom}}HL = d_{\text{hom}}T$. This finishes the proof of the main theorem. $\qquad\square$

*Remark* 12. Equations similar to Eq. (46) can be found in [18, App. B.2]. However, a small correction must be made to their derivation. The authors first apply Cauchy-Schwarz and then take the square of Eq. (45) to replace the original regret with the surrogate regret. As seen later in the proof, we do it in the opposite order, because in (45), the left side may be negative, so we can not assume the square of that estimation to be also correct. While outside the focus of this paper, we note that a side-effect of this correction is another one to their definition of surrogate-IDS in [18, Eq. (4.3)], wherein minimization should be over the square root of their information ratio.

# E   Bounding $\mathbb{E}[\sum_\ell \mathcal{I}^\ell]$

We start by proving $\mathcal{I}^\ell \leq \frac{1}{2}\mathbb{I}_\ell^{\pi_{\text{TS}}^\ell}(\tilde{\mathcal{E}}_\ell^*; \mathcal{H}_{\ell,H}), \forall \ell \in [L]$. We recall that TS property implies $\mathbb{E}_\ell[d_{h,\pi_{\mathcal{E}}^*}^{\bar{\mathcal{E}}_\ell}(s,a)] = \mathbb{E}_\ell[d_{h,\pi_{\text{TS}}^\ell}^{\bar{\mathcal{E}}_\ell}(s,a)]$. So,

$$\mathcal{I}^\ell = \sum_{h=1}^H \mathbb{E}_\ell\left[\int_{s,a}\mathbb{E}_\ell[d_{h,\pi_{\mathcal{E}}^*}^{\bar{\mathcal{E}}_\ell}(s,a)]\frac{\Delta_h^{\tilde{\mathcal{E}}_\ell^*}(s,a)^2}{\lambda_{\mathcal{E}}^2}\,\mathrm{d}\mu_{\mathcal{S}\times\mathcal{A}}\right] \tag{56}$$

$$= \sum_{h=1}^H \mathbb{E}_\ell\left[\int_{s,a}\mathbb{E}_\ell[d_{h,\pi_{\text{TS}}^\ell}^{\bar{\mathcal{E}}_\ell}(s,a)]\frac{\Delta_h^{\tilde{\mathcal{E}}_\ell^*}(s,a)^2}{\lambda_{\mathcal{E}}^2}\,\mathrm{d}\mu_{\mathcal{S}\times\mathcal{A}}\right]. \tag{57}$$

Next, we swap the two integrals, one represented by $\mathbb{E}_\ell$ and the one over $s, a$,

$$= \sum_{h=1}^H \int_{s,a}\mathbb{E}_\ell[d_{h,\pi_{\text{TS}}^\ell}^{\bar{\mathcal{E}}_\ell}(s,a)]\mathbb{E}_\ell\left[\frac{\Delta_h^{\tilde{\mathcal{E}}_\ell^*}(s,a)^2}{\lambda_{\mathcal{E}}^2}\right]\mathrm{d}\mu_{\mathcal{S}\times\mathcal{A}} \tag{58}$$

Note that, given $\mathcal{D}_\ell$, $\Delta_h^{\tilde{\mathcal{E}}_\ell^*}$ is independent of $d_{h,\pi_{\text{TS}}^\ell}^{\bar{\mathcal{E}}_\ell}(s,a)$. Therefore, using the identity $\mathbb{E}[XY] = \mathbb{E}[X]\mathbb{E}[Y]$ for two independent random variables, and swapping back the two integrals,

$$= \sum_{h=1}^H \mathbb{E}_\ell\left[\int_{s,a}d_{h,\pi_{\text{TS}}^\ell}^{\bar{\mathcal{E}}_\ell}(s,a)\frac{\Delta_h^{\tilde{\mathcal{E}}_\ell^*}(s,a)^2}{\lambda_{\mathcal{E}}^2}\,\mathrm{d}\mu_{\mathcal{S}\times\mathcal{A}}\right] = \sum_{h=1}^H \mathbb{E}_\ell\left[\mathbb{E}_{\pi_{\text{TS}}^\ell}^{\bar{\mathcal{E}}_\ell}\left[\frac{\Delta_h^{\tilde{\mathcal{E}}_\ell^*}(s,a)^2}{\lambda_{\mathcal{E}}^2}\right]\right] \tag{59}$$

Finally, notice that $\frac{\Delta_h^{\tilde{\mathcal{E}}_\ell^*}(s,a)^2}{\lambda_{\mathcal{E}}^2}$ can be estimated by Pinsker's inequality (Lemma 9) as

$$
\begin{aligned}
&\frac{\Delta_h^{\tilde{\mathcal{E}}_\ell^*}(s,a)^2}{\lambda_{\mathcal{E}}^2} \\
&= \left( \mathbb{E}_{(s',r')\sim(P_h^{\tilde{\mathcal{E}}_\ell}\otimes r_h^{\tilde{\mathcal{E}}_\ell})(\cdot|s_h,a_h)} \left[ \frac{r' + V_{h+1,\pi_{\mathcal{E}}^*}^{\tilde{\mathcal{E}}_\ell^*}(s')}{\lambda_{\mathcal{E}}} \right] \right. \\
&\qquad \left. - \mathbb{E}_{(s',r')\sim(P_h^{\bar{\mathcal{E}}_\ell}\otimes r_h^{\bar{\mathcal{E}}_\ell})(\cdot|s_h,a_h)} \left[ \frac{r' + V_{h+1,\pi_{\mathcal{E}}^*}^{\tilde{\mathcal{E}}_\ell^*}(s')}{\lambda_{\mathcal{E}}} \right] \right)^2 \\
&= \left( \mathbb{E}_{(s',r')\sim(P_h^{\tilde{\mathcal{E}}_\ell}\otimes r_h^{\tilde{\mathcal{E}}_\ell})(\cdot|s_h,a_h)} \left[ \frac{r' + V_{h+1,\pi_{\mathcal{E}}^*}^{\tilde{\mathcal{E}}_\ell^*}(s') - r_h^{\inf}(s_h,a_h) - \inf_s V_{h+1,\pi_{\mathcal{E}}^*}^{\tilde{\mathcal{E}}_\ell^*}(s)}{\lambda_{\mathcal{E}}} \right] \right. \\
&\qquad \left. - \mathbb{E}_{(s',r')\sim(P_h^{\bar{\mathcal{E}}_\ell}\otimes r_h^{\bar{\mathcal{E}}_\ell})(\cdot|s_h,a_h)} \left[ \frac{r' + V_{h+1,\pi_{\mathcal{E}}^*}^{\tilde{\mathcal{E}}_\ell^*}(s') - r_h^{\inf}(s_h,a_h) - \inf_s V_{h+1,\pi_{\mathcal{E}}^*}^{\tilde{\mathcal{E}}_\ell^*}(s)}{\lambda_{\mathcal{E}}} \right] \right)^2 \\
&\leq \frac{1}{2} D_{\mathrm{KL}} \left( (P_h^{\tilde{\mathcal{E}}_\ell^*} \otimes r_h^{\tilde{\mathcal{E}}_\ell^*})(\cdot|s_h,a_h) || (P_h^{\bar{\mathcal{E}}_\ell} \otimes r_h^{\bar{\mathcal{E}}_\ell})(\cdot|s_h,a_h) \right)
\end{aligned}
$$

where we note the trick of adding and subtracting the constant term $r_h^{\inf}(s_h,a_h) + \inf_s V_{h+1,\pi_{\mathcal{E}}^*}^{\tilde{\mathcal{E}}_\ell^*}(s)$ to the expected values in order to make the expression inside between zero and $\lambda_{\mathcal{E}}$. This enables the application of Pinsker's inequality which requires the random variable $X$ in $(\mathbb{E}_P[X] - \mathbb{E}_Q[X])^2 \leq \frac{1}{2}D_{\mathrm{KL}}(P||Q)$ to be smaller than one. Therefore

$$
\mathcal{I}^\ell \leq \frac{1}{2} \sum_{h=1}^H \mathbb{E}_\ell \left[ \mathbb{E}_{\pi_{\mathrm{TS}}^\ell}^{\bar{\mathcal{E}}_\ell} \left[ D_{\mathrm{KL}} \left( (P_h^{\tilde{\mathcal{E}}_\ell^*} \otimes r_h^{\tilde{\mathcal{E}}_\ell^*})(\cdot|s_h,a_h) || (P_h^{\bar{\mathcal{E}}_\ell} \otimes r_h^{\bar{\mathcal{E}}_\ell})(\cdot|s_h,a_h) \right) \right] \right]. \tag{60}
$$

Lastly, we use Lemma 10, wherein we show a fact similar to [18, App. C.1.] but for $\tilde{\mathcal{E}}_\ell^*$ instead of $\mathcal{E}$, proving the above equals $\frac{1}{2}\mathbb{I}_\ell^{\pi_{\mathrm{TS}}^\ell}(\tilde{\mathcal{E}}_\ell^*; \mathcal{H}_{\ell,H})$.

Next, observe that $\mathbb{I}_\ell^{\pi_{\mathrm{TS}}^\ell}(\tilde{\mathcal{E}}_\ell^*; \mathcal{H}_{\ell,H}) \leq \mathbb{I}_\ell^{\pi_{\mathrm{TS}}^\ell}(\zeta; \mathcal{H}_{\ell,H})$. Indeed, recall that the definition of $\mathbb{I}_\ell$ conditions on a $\mathcal{D}_\ell$; and we know that conditional on a $\mathcal{D}_\ell$, the surrogate environment $\tilde{\mathcal{E}}_\ell^*$ is only dependent on $\zeta$ by construction, hence the data processing inequality applies and we have

$$
\mathcal{I}^\ell \leq \frac{1}{2}\mathbb{I}_\ell^{\pi_{\mathrm{TS}}^\ell}(\zeta; \mathcal{H}_{\ell,H}). \tag{61}
$$

Next, we use the mutual information chain rule, observing that

$$
\mathbb{E}\left[ \mathbb{I}_\ell^{\pi_{\mathrm{TS}}^\ell}(\zeta; \mathcal{H}_{\ell,H}) \right] = \mathbb{I}(\zeta; \mathcal{H}_{\ell,H}|\mathcal{H}_{\ell-1,H}, \ldots, \mathcal{H}_{1,H}), \tag{62}
$$

and therefore

$$
\mathbb{I}(\zeta; \mathcal{D}_{L+1}) = \mathbb{I}(\zeta; (\mathcal{H}_{1,H}, \ldots, \mathcal{H}_{L,H})) = \sum_{\ell=1}^L \mathbb{E}\left[ \mathbb{I}_\ell^{\pi_{\mathrm{TS}}^\ell}(\zeta; \mathcal{H}_{\ell,H}) \right]. \tag{63}
$$

Applied to the above, and noting that $\mathbb{I}(\zeta; \mathcal{D}_{L+1}) \leq H(\zeta) \leq \log(K_{\mathrm{surr}}(\varepsilon))$, this finishes our estimation of $\mathbb{E}[\sum_\ell \mathcal{I}^\ell] \leq \frac{1}{2}\log(K_{\mathrm{surr}}(\varepsilon))$.

# F  Bounding $\mathbb{E}[\mathcal{T}^\ell]$

This is where we use analysis tools from posterior consistency. We focus on bounding

$$
\mathcal{T}^\ell = \int_{(s,a,h)\in\mathcal{B}_\ell} \frac{\mathbb{E}_\ell\left[ (\lambda_{\mathcal{E}} d_{h,\pi_{\mathcal{E}}^*}^{\bar{\mathcal{E}}_\ell}(s,a))^2 \right]}{\mathbb{E}_\ell\left[ d_{h,\pi_{\mathcal{E}}^*}^{\bar{\mathcal{E}}_\ell}(s,a) \right]}. \tag{64}
$$

Note that we have

$$\mathbb{E}_\ell \left[ d_{h,\pi_\mathcal{E}^*}^{\bar{\mathcal{E}}_\ell}(s,a) \right] = \mathbb{E}_\ell \left[ d_{h,\pi_\mathcal{E}^*}^{\mathcal{E}'}(s,a) \right],$$

where $\mathcal{E}'$ is sampled from the posterior $\mathbb{P}_\ell$ independent of $\mathcal{E}$. Similarly

$$\begin{aligned}
\mathbb{E}_\ell \left[ (\lambda_\mathcal{E} d_{h,\pi_\mathcal{E}^*}^{\bar{\mathcal{E}}_\ell}(s,a))^2 \right] &= (\mathbb{E}_\ell)_{\mathcal{E}\sim\mathbb{P}_\ell} \left[ \lambda_\mathcal{E}^2 \left( (\mathbb{E}_\ell)_{\mathcal{E}'\sim\mathbb{P}_\ell} \left[ d_{h,\pi_\mathcal{E}^*}^{\mathcal{E}'}(s,a) \right] \right)^2 \right] \\
&\le (\mathbb{E}_\ell)_{\mathcal{E}\sim\mathbb{P}_\ell} \left[ \lambda_\mathcal{E}^2 (\mathbb{E}_\ell)_{\mathcal{E}'\sim\mathbb{P}_\ell} \left[ d_{h,\pi_\mathcal{E}^*}^{\mathcal{E}'}(s,a)^2 \right] \right] \\
&= \mathbb{E}_\ell \left[ (\lambda_\mathcal{E} d_{h,\pi_\mathcal{E}^*}^{\mathcal{E}'}(s,a))^2 \right].
\end{aligned}$$

For any $\ell, s, a, h$ and $\mathcal{D}_\ell$, we define

$$g_\ell(s,a,h,\mathcal{D}_\ell) := \frac{\mathbb{E}_\ell \left[ (\lambda_\mathcal{E} d_{h,\pi_\mathcal{E}^*}^{\mathcal{E}'}(s,a))^2 \right]}{\mathbb{E}_\ell \left[ d_{h,\pi_\mathcal{E}^*}^{\mathcal{E}'}(s,a) \right]}, \tag{65}$$

whenever $(s,a,h) \in \mathcal{B}_\ell$ and $g_\ell(s,a,h,\mathcal{D}_\ell) := 0$ otherwise. Clearly we have

$$\mathcal{T}^\ell \le \int_{(s,a,h)} g_\ell(s,a,h,\mathcal{D}_\ell). \tag{66}$$

Moreover, given our assumption on state action occupation density, we have $M_d :=$ $\sup_{s,a,h,\pi,\mathcal{E}} d_{h,\pi}^{\mathcal{E}}(s,a) < \infty$, which implies $g_\ell \le M_d H^2 < \infty$ and $\mathcal{T}^\ell \le M_d H^3 < \infty$. Let $\mathcal{E}_0$ be the true environment. This means $\mathbb{E}[\cdot|\mathcal{E}_0] = \mathbb{E}_{\mathcal{D}_\ell\sim\mathbb{P}(\cdot|\mathcal{E}_0)}[\cdot]$.

According to Corollary 12, we have

$$\lim_{\ell\to\infty} \mathbb{E}_\ell \left[ d_{h,\pi_\mathcal{E}^*}^{\mathcal{E}'}(s,a) \right] = d_{h,\pi_{\mathcal{E}_0}^*}^{\mathcal{E}_0}(s,a), \tag{67}$$

$$\lim_{\ell\to\infty} \mathbb{E}_\ell \left[ (\lambda_\mathcal{E} d_{h,\pi_\mathcal{E}^*}^{\mathcal{E}'}(s,a))^2 \right] = (\lambda_{\mathcal{E}_0} d_{h,\pi_{\mathcal{E}_0}^*}^{\mathcal{E}_0}(s,a))^2. \tag{68}$$

According to Assumption 2, $d_{h,\pi_{\mathcal{E}_0}^*}^{\mathcal{E}_0}(s,a) \ne 0$ for almost every $\mathcal{E}_0, s, a$ and $h$. For any such values of $(\mathcal{E}_0, s, a, h)$ and almost every $\mathcal{D}_\ell$ sampled from true environment $\mathcal{E}_0$, we conclude that

$$\lim_{\ell\to\infty} g_\ell(s,a,h,\mathcal{D}_\ell) = \lambda_{\mathcal{E}_0}^2 d_{h,\pi_{\mathcal{E}_0}^*}^{\mathcal{E}_0}(s,a). \tag{69}$$

Therefore, using dominated convergence theorem, for almost every $\mathcal{E}_0$ we have

$$\lim_{\ell\to\infty} \mathbb{E}[\mathcal{T}^\ell|\mathcal{E}_0] \le \lim_{\ell\to\infty} \mathbb{E}_{\mathcal{D}_\ell\sim\mathbb{P}(\cdot|\mathcal{E}_0)} \left[ \int_{(s,a,h)} g_\ell(s,a,h,\mathcal{D}_\ell) \right] \tag{70}$$

$$= \lim_{\ell\to\infty} \int_{(s,a,h)} \mathbb{E}_{\mathcal{D}_\ell\sim\mathbb{P}(\cdot|\mathcal{E}_0)} \left[ g_\ell(s,a,h,\mathcal{D}_\ell) \right] \tag{71}$$

$$= \int_{(s,a,h)} \mathbb{E}_{\mathcal{D}_\ell\sim\mathbb{P}(\cdot|\mathcal{E}_0)} \left[ \lim_{\ell\to\infty} g_\ell(s,a,h,\mathcal{D}_\ell) \right] \tag{72}$$

$$= \int_{(s,a,h)} \mathbb{E}_{\mathcal{D}_\ell\sim\mathbb{P}(\cdot|\mathcal{E}_0)} \left[ \lambda_{\mathcal{E}_0}^2 d_{h,\pi_{\mathcal{E}_0}^*}^{\mathcal{E}_0}(s,a) \right] \tag{73}$$

$$= \int_{(s,a,h)} \lambda_{\mathcal{E}_0}^2 d_{h,\pi_{\mathcal{E}_0}^*}^{\mathcal{E}_0}(s,a) \tag{74}$$

$$= \lambda_{\mathcal{E}_0}^2 H. \tag{75}$$

Therefore we may use dominated convergence theorem again to see that

$$\lim_{\ell\to\infty} \mathbb{E}[\mathcal{T}^\ell] = \lim_{\ell\to\infty} \mathbb{E}[\mathbb{E}[\mathcal{T}^\ell|\mathcal{E}_0]] = \mathbb{E}[\lim_{\ell\to\infty} \mathbb{E}[\mathcal{T}^\ell|\mathcal{E}_0]] \le \mathbb{E}[\lambda_{\mathcal{E}_0}^2 H] = \mathbb{E}[\lambda_\mathcal{E}^2]H = \lambda^2 H. \tag{76}$$

Therefore, there exists $L_0 > 0$ such that $\mathbb{E}[\mathcal{T}^\ell] \le 2\lambda^2 H$ for $\ell > L_0$.

# G   Proof of Corollary 5

We restate the corollary below. We prove it in a more general case where the maps $\phi^P, \phi^R$ are time inhomogeneous, i.e. $\phi_h^P, \phi_h^R$. In addition, the dimension of their target space can also depend on $h$, i.e. we use $d_f^{P,h}, d_f^{R,h}$, instead of just $d_f^P, d_f^R$. For the case where the dimensions are homogeneous, we use $d_f^{\mathrm{hom}}$. This new notation impacts the statement as follows:

**Corollary.** *For a linear Bayesian RL, for large enough $T$,*

$$\mathfrak{BR}_L(\pi_{TS}) \leq \widetilde{O}(\lambda\sqrt{d_{l_1}^f T}). \tag{77}$$

*Given a linear Bayesian RL with finitely many states and homogeneous feature space dimension $d_f^{\mathrm{hom}}$, we have $d_{l_1}^f \leq 2d_f^{\mathrm{hom}}HS$, yielding for large enough $T$,*

$$\mathfrak{BR}_L(\pi_{TS}) \leq \widetilde{O}(\lambda\sqrt{Hd_f^{\mathrm{hom}}ST}). \tag{78}$$

*Given a mixture linear Bayesian RL, for large enough $T$,*

$$\mathfrak{BR}_L(\pi_{TS}) \leq \widetilde{O}(\lambda\sqrt{MT}), \tag{79}$$

The first statement follows from the generic bound, but we need to relate the $l_1-$dimension of the environment space to that of the feature maps space, where we recall the definitions $d_{l_1}^f = d_{l_1}^{P,f} + d_{l_1}^{R,f}$ as the sum of the $l_1-$dimensions of the feature map space $\{\psi_h^{P,\mathcal{E}}\}_{\mathcal{E}\in\Theta}, \{\psi_h^{R,\mathcal{E}}\}_{\mathcal{E}\in\Theta}$ where the $l_1-$distance between feature maps is defined as $l_1(\psi_h^{\mathcal{E}}, \psi_h^{\mathcal{E}'}) = \int_s \|\psi_h^{\mathcal{E}} - \psi_h^{\mathcal{E}'}\|_1 \mu_{\mathcal{S}}$. We shall use Fact 1:

$$V_{1,\pi_{\mathcal{E}}^*}^{\mathcal{E}}(s_1^\ell) - V_{1,\pi_{\mathcal{E}}^*}^{\mathcal{E}'}(s_1^\ell) \leq H\sum_{h=1}^{H} l_1(P_h^{\mathcal{E}}, P_h^{\mathcal{E}'}) + \sum_{h=1}^{H} l_1(r_h^{\mathcal{E}}, r_h^{\mathcal{E}'}). \tag{80}$$

We estimate

$$l_1(P_h^{\mathcal{E}}, P_h^{\mathcal{E}'}) = \sup_{s,a}\int_{s'} |P_h^{\mathcal{E}}(s'|s,a) - P_h^{\mathcal{E}'}(s'|s,a)| = \sup_{s,a}\int_{s'} |\phi_h^P(s,a)\cdot(\psi_h^{P,\mathcal{E}}(s') - \psi_h^{P,\mathcal{E}}(s'))| \leq$$

$$\sup_{s,a}\int_{s'}\sum_{i=1}^{d_{f,h}^P} |\phi_h^P(s,a)_i(\psi_h^{P,\mathcal{E}}(s') - \psi_h^{P,\mathcal{E}}(s'))_i| \tag{81}$$

Since $\|\phi_h^P(s,a)\|_2 \leq 1 \implies |\phi_h^P(s,a)_i| \leq 1, \forall i \in [d]$. Therefore

$$l_1(P_h^{\mathcal{E}}, P_h^{\mathcal{E}'}) \leq \int_{s'} \|(\psi_h^{P,\mathcal{E}}(s') - \psi_h^{P,\mathcal{E}}(s'))\|_1 = l_1(\psi_h^{P,\mathcal{E}}, \psi_h^{P,\mathcal{E}}). \tag{82}$$

The similar bound can be achieved for $l_1(r_h^{\mathcal{E}}, r_h^{\mathcal{E}'})$. As a result $d_{l_1} \leq d_{l_1}^f$ and we get the first statement.

For Eq. (78), we note that if $S$ is a finite, then $\psi_h^{P,\mathcal{E}}$ can be viewed as a $d_f^{P,h} \times S$ matrix, or simply a vector with dimension that size. Therefore, we can view our problem as asking for the asymptotics of the $\varepsilon-$covering number in $\mathbb{R}^{d_f^{P,h}S}$. As long as the collection $\{\psi_h^{P,\mathcal{E}}\}_{\mathcal{E}}$ is within a finite ball, which they are (Definition 4), the covering number is well-known to scale at most as $O((\frac{\log(C_\psi)}{\varepsilon})^{d_f^{P,h}S})$ where $C_\psi$ is the radius of that ball. Applying the similar argument for the rewards, we have $d_{l_1} \leq d_{l_1}^f \leq \sum(d_f^{P,h} + d_f^{R,h})S$, which equals $2Hd_f^{\mathrm{hom}}S$ given a homogeneous feature space dimension $d_f^{\mathrm{hom}} = d_f^{P,h} = d_f^{R,h}$, for all $h \in [H]$.

The finite mixtures statement is a straightforward generalization of the above, where every $\psi_h^{P,\mathcal{E}}, \psi_h^{R,\mathcal{E}}$ is characterized with a finite $m_h^P, m_h^R$-dimensional vector instead of specifically being $d_f^{P,h}S, d_f^{R,h}S$-dimensional. We note

$$l_1(\psi_h^{P,\mathcal{E}}, \psi_h^{P,\mathcal{E}}) = \int_{s'} \|(\psi_h^{P,\mathcal{E}}(s') - \psi_h^{P,\mathcal{E}}(s'))\|_1 \leq C_\Psi\|\boldsymbol{a}_h^{P,\mathcal{E}} - \boldsymbol{a}_h^{P,\mathcal{E}'}\|_1 \tag{83}$$

where $C_\Psi = \max_{1\leq i\leq m_h^P} \|\Psi_{h,i}^P\|_1$. So the same argument above applies, where we consider the collection of finite dimensional vectors $\{\boldsymbol{a}_h^{P,\mathcal{E}}\}_{\mathcal{E}}$ and the $\varepsilon-$covering number, and similarly for $\{\boldsymbol{a}_h^{R,\mathcal{E}}\}_{\mathcal{E}}$.

### G.1 Incorrect proof for the Bayesian regret bound of linear Bayesian RL with deterministic rewards

Here, we discuss the proof of [18] for linear RLs, and the mistakes in their argument. We start by citing the similar equations in [18, App. B.4] in bounding the value difference with the feature maps difference:

For any $\mathcal{E}_1, \mathcal{E}_2 \in \Theta_k$, [...]

$$V_{1,\pi_{\mathcal{E}_1}^*}^{\mathcal{E}_1}(s_1) - V_{1,\pi_{\mathcal{E}_1}^*}^{\mathcal{E}_2}(s_1)$$

$$= \sum_{h=1}^H \mathbb{E}_{\pi_{\mathcal{E}_1}^*}^{\mathcal{E}_2} \left[ P_h^{\mathcal{E}_1}(\cdot|s_h^\ell, a_h^\ell)^\top V_{h+1,\pi_{\mathcal{E}_1}^*}^{\mathcal{E}_1}(\cdot) - P_h^{\mathcal{E}_2}(\cdot|s_h^\ell, a_h^\ell)^\top V_{h+1,\pi_{\mathcal{E}_1}^*}^{\mathcal{E}_1}(\cdot) \right]$$

$$= \sum_{h=1}^H \mathbb{E}_{\pi_{\mathcal{E}_1}^*}^{\mathcal{E}_2} \left[ \phi(s_h^\ell, a_h^\ell)^\top \sum_{s'} V_{h+1,\pi_{\mathcal{E}_1}^*}^{\mathcal{E}_1}(s') \psi_h^{\mathcal{E}_1}(s') \right.$$

$$\left. - \phi(s_h^\ell, a_h^\ell)^\top \sum_{s'} V_{h+1,\pi_{\mathcal{E}_1}^*}^{\mathcal{E}_1}(s') \psi_h^{\mathcal{E}_2}(s') \right],$$

[...] Moreover, since the value function is always bounded by $H$, we have

$$V_{1,\pi_{\mathcal{E}_1}^*}^{\mathcal{E}_1}(s_1) - V_{1,\pi_{\mathcal{E}_1}^*}^{\mathcal{E}_2}(s_1)$$

$$= H \sum_{h=1}^H \mathbb{E}_{\pi_{\mathcal{E}_1}^*}^{\mathcal{E}_2} \left[ \phi(s_h^\ell, a_h^\ell)^\top \left( \sum_{s'} \psi_h^{\mathcal{E}_1}(s') - \sum_{s'} \psi_h^{\mathcal{E}_2}(s') \right) \right]$$

$$\leq H \sum_{h=1}^H \mathbb{E}_{\pi_{\mathcal{E}_1}^*}^{\mathcal{E}_2} \left[ \left\| \phi(s_h^\ell, a_h^\ell) \right\|_2 \right] \left\| \sum_{s'} \psi_h^{\mathcal{E}_1}(s') - \sum_{s'} \psi_h^{\mathcal{E}_2}(s') \right\|_2$$

$$\leq H \sum_{h=1}^H \left\| \sum_{s'} \psi_h^{\mathcal{E}_1}(s') - \sum_{s'} \psi_h^{\mathcal{E}_2}(s') \right\|_2. \tag{84}$$

Clearly, in the first equation, the equality must be replaced by $\leq$, and more importantly, given that the value function $V_{h+1,\pi_{\mathcal{E}_1}^*}^{\mathcal{E}_1}(\cdot)$ has argument $\cdot = s'$, the $l_2-$norm should be taken on the **inside** of the integral $\int_{s'} \| \phi(s_h^\ell, a_h^\ell)^\top \left( \psi_h^{\mathcal{E}_1}(s') - \psi_h^{\mathcal{E}_2}(s') \right) \|_2$ (we note we also replaced the sum $\sum_{s'}$ with integral, as linear RLs could have infinitely many states). If our proposed correction were to be followed then the next equations would change to ones similar to ours except with an $l_2-$distance:

$$\leq H \sum_{h=1}^H \mathbb{E}_{\pi_{\mathcal{E}_1}^*}^{\mathcal{E}_2} \left[ \left\| \phi(s_h^\ell, a_h^\ell) \right\|_2 \right] \int_{s'} \left\| \psi_h^{\mathcal{E}_1}(s') - \psi_h^{\mathcal{E}_2}(s') \right\|_2 \tag{85}$$

$$\leq H \sum_{h=1}^H \int_{s'} \left\| \psi_h^{\mathcal{E}_1}(s') - \psi_h^{\mathcal{E}_2}(s') \right\|_2 \tag{86}$$

Otherwise, let us assume that the authors were correct, then we have managed to bound $V_{1,\pi_{\mathcal{E}_1}^*}^{\mathcal{E}_1}(s_1) - V_{1,\pi_{\mathcal{E}_1}^*}^{\mathcal{E}_2}(s_1) \leq H \sum_{h=1}^H \left\| \sum_{s'} \psi_h^{\mathcal{E}_1}(s') - \sum_{s'} \psi_h^{\mathcal{E}_2}(s') \right\|_2$. We show that this is a bound by zero for an important subclass of linear RLs, i.e. all tabular RLs.

It is a well-known fact that tabular RLs can be viewed as linear RLs. The mapping works as follows. First let us enumerate the set $\{(s,a) \in \mathcal{S} \times \mathcal{A}\}$ by $1, \ldots, SA$. Call this assignment $N(s,a) \in [SA]$. Then define $\phi(s,a) = e_{N(s,a)} \in \mathbb{R}^{SA}$, which is the Euclidean basis state on axis $N(s,a)$. Let $\psi_h^{\mathcal{E}}(s') = (P_h^{\mathcal{E}}(s'|s,a))_{s,a}$. Then clearly all the conditions $\|\phi\|_2 \leq 1, \|\sum_{s'} \psi(s')\|_2 \leq C_\psi$ and most importantly $P_h(\cdot|s,a) = \langle \phi(s,a), \psi_h^{\mathcal{E}}(s') \rangle$, are satisfied. However we note that for any $\mathcal{E}, h$ we have $\sum_{s'} \psi_h^{\mathcal{E}}(s') = (\sum_{s'} P_h^{\mathcal{E}}(s'|s,a))_{s,a} = (1)_{s,a}$ which is the all one vector in $\mathbb{R}^{SA}$. In that case, $\sum_{s'} \psi_h^{\mathcal{E}}(s') - \sum_{s'} \psi_h^{\mathcal{E}'}(s')$ in Eq. (84) is the zero vector, with zero norm.

Therefore, were the estimation in [18, App. B.4] correct, for all tabular Bayesian RLs with deterministic reward, the difference of all value functions of the form $V_{1,\pi_{\mathcal{E}_1}^*}^{\mathcal{E}_1} - V_{1,\pi_{\mathcal{E}_2}^*}^{\mathcal{E}_2}$ would be bounded above by zero, meaning we have estimated $K_{\mathrm{surr}}(\varepsilon) = 1$ for all $\varepsilon$, which since $\log(1) = 0$, implies a constant regret bound as well. This counterexample further demonstrates the mistake above.

Overall, this makes the proof for [18, Theorem 4.10] incorrect and invalidates their claim of a regret bound $\widetilde{O}(d_f^{\mathrm{hom}} H^{3/2} \sqrt{T})$.

*Remark* 13. Another important gap in the proof of [18, Theorem 4.10] can be found in [18, App. B.5], where the surrogate regret is claimed to be bounded by a conditional mutual information by $\pi^*$ instead of $\pi_{\mathrm{TS}}^\ell$. This is explained in further details in Appendix J.1.

# H  Proof of Corollary 6

We restate the corollary.

**Corollary.** *Given a finite mixtures Bayesian RL problem, for large enough $T$,*

$$\mathfrak{BR}_L(\pi_{TS}) \leq \widetilde{O}(\lambda \sqrt{d_{l_1}^m T}) . \tag{87}$$

*Assuming the restricted finite mixtures model, for large enough $T$,*

$$\mathfrak{BR}_L(\pi_{TS}) \leq \widetilde{O}\left(\lambda \sqrt{MT}\right) . \tag{88}$$

*which, given a uniform dimension $m = m_h^P = m_h^R$, yields $\widetilde{O}(\lambda \sqrt{HmT})$.*

Given Theorem 3 and Fact 1, we need to estimate the $d_{l_1}$ of $\Theta$ by that of $\{a_h^{P,\mathcal{E}}(s,a)\}_{\mathcal{E} \in \Theta}$ and $\{a_h^{R,\mathcal{E}}(s,a)\}_{\mathcal{E} \in \Theta}$. Indeed writing the $l_1-$distance of two transition functions:

$$\sup_{s,a} \|P_h^{\mathcal{E}}(\cdot|s,a) - P_h^{\mathcal{E}'}(\cdot|s,a)\|_1 = \sup_{s,a} \|\sum_{i=1}^{m_h}(a_{h,i}^{P,\mathcal{E}}(s,a) - a_{h,i}^{P,\mathcal{E}'}(s,a))Z_{h,i}^P(\cdot|s,a)\|_1 \leq$$

$$\sup_{s,a} \sum_{i=1}^{m_h} \|(a_{h,i}^{P,\mathcal{E}}(s,a) - a_{h,i}^{P,\mathcal{E}'}(s,a))Z_{h,i}^P(\cdot|s,a)\|_1 = \sup_{s,a} \sum_{i=1}^{m_h} |a_{h,i}^{P,\mathcal{E}}(s,a) - a_{h,i}^{P,\mathcal{E}'}(s,a)| = \tag{89}$$

$$\sup_{s,a} \|a_h^{P,\mathcal{E}}(s,a) - a_h^{P,\mathcal{E}'}(s,a)\|_1$$

where we used the triangle inequality and the fact that the density functions are positive and their integral equals one.

For the second statement, we are faced with the problem of finding an $l_1-$covering number for a collection of vectors on the $m_h-$dimensional simplex. It is a standard fact that the covering of the latter is of order $O\left((\frac{1}{\varepsilon})^{m_h}\right)$, implying Eq. (88).

# I  Useful Lemmas

**Lemma 8.** *For any two environments $\mathcal{E}, \mathcal{E}'$ with potentially different transition and reward functions, and any policy $\pi$, we have*

$$V_{1,\pi}^{\mathcal{E}}(s_1) - V_{1,\pi}^{\mathcal{E}'}(s_1)$$

$$= \sum_{h=1}^{H} \mathbb{E}_{\pi}^{\mathcal{E}'} \left[ \mathbb{E}_{s' \sim P_h^{\mathcal{E}}(\cdot|s_h,a_h)}[V_{h+1,\pi}^{\mathcal{E}}(s')] - \mathbb{E}_{s' \sim P_h^{\mathcal{E}'}(\cdot|s_h,a_h)}[V_{h+1,\pi}^{\mathcal{E}}(s')] \right]$$

$$+ \sum_{h=1}^{H} \mathbb{E}_{\pi}^{\mathcal{E}'} [r_h^{\mathcal{E}}(s_h,a_h) - r_h^{\mathcal{E}'}(s_h,a_h)]$$

$$= \sum_{h=1}^{H} \mathbb{E}_{\pi}^{\mathcal{E}'} \left[ \mathbb{E}_{(s',r') \sim (P_h^{\mathcal{E}} \otimes r_h^{\mathcal{E}})(\cdot|s_h,a_h)}[r' + V_{h+1,\pi}^{\mathcal{E}}(s')] \right.$$

$$\left. - \mathbb{E}_{(s',r') \sim (P_h^{\mathcal{E}'} \otimes r_h^{\mathcal{E}'})(\cdot|s_h,a_h)}[r' + V_{h+1,\pi}^{\mathcal{E}}(s')] \right],$$

*where $V_{H+1,\pi^*}^{\mathcal{E}}(\cdot) := 0$ and the expectation $\mathbb{E}_{\pi}^{\mathcal{E}'}$ is with respect to $s_h, a_h$.*

Note that when rewards are deterministic, we have

$$V_{1,\pi}^{\mathcal{E}}(s_1) - V_{1,\pi}^{\mathcal{E}'}(s_1) = \sum_{h=1}^{H} \mathbb{E}_\pi^{\mathcal{E}'} \left[ \mathbb{E}_{s' \sim P_h^{\mathcal{E}}(\cdot|s_h,a_h)}[V_{h+1,\pi}^{\mathcal{E}}(s')] - \mathbb{E}_{s' \sim P_h^{\mathcal{E}'}(\cdot|s_h,a_h)}[V_{h+1,\pi}^{\mathcal{E}}(s')] \right],$$

which is the statements of [18, Lemma D.3].

*Proof.* We have

$$V_{1,\pi}^{\mathcal{E}}(s_1) - V_{1,\pi}^{\mathcal{E}'}(s_1) = (V_{1,\pi}^{\mathcal{E}}(s_1) - V_{1,\pi}^{\mathcal{E}'_r\mathcal{E}}(s_1)) + (V_{1,\pi}^{\mathcal{E}'_r\mathcal{E}}(s_1) - V_{1,\pi}^{\mathcal{E}'}(s_1)) \tag{90}$$

where $\mathcal{E}'_{r\mathcal{E}}$ is an environment with the transition functions of $\mathcal{E}'$ but reward functions of $\mathcal{E}$. The first term above may be rewritten using [18, Lemma D.3]. The second term may be rewritten using the direct definition of value function as $V_{1,\pi}^{\mathcal{G}} = \mathbb{E}_\pi^{\mathcal{G}}[\sum_{h=1}^{H} r^{\mathcal{G}}(s_h,a_h)]$, which completes the proof. $\square$

Pinsker's lemma is at the center of relating the two concepts of regret and mutual information. We cite the following variant of the Pinsker's inequality from Fact 9 in [35].

**Lemma 9.** *For any distribution $P$ and $Q$ such that $P$ is absolutely continuous with respect to $Q$, any random variable $X : \Omega \to \mathcal{X}$ and any $g : \mathcal{X} \to \mathbb{R}$ such that $\sup g - \inf g \leq 1$, we have*

$$\mathbb{E}_P[g(x)] - \mathbb{E}_Q[g(x)] \leq \sqrt{\frac{1}{2} D_{\mathrm{KL}}(P||Q)}. \tag{91}$$

## J  Mutual information of surrogate environment and history

Recall that by performing the information ratio trick, Cauchy-Schwarz and Pinsker's inequality, we obtained the following term in our bound of the squared regret:

$$\frac{1}{2} \sum_{h=1}^{H} \mathbb{E}_\ell \left[ \mathbb{E}_{\pi_{\mathrm{TS}}^\ell}^{\bar{\mathcal{E}}_\ell} \left[ D_{\mathrm{KL}} \left( (P_h^{\tilde{\mathcal{E}}_\ell^*} \otimes r_h^{\tilde{\mathcal{E}}_\ell^*})(\cdot|s_{h-1}^\ell,a_{h-1}^\ell)||(P_h^{\bar{\mathcal{E}}_\ell} \otimes r_h^{\bar{\mathcal{E}}_\ell})(\cdot|s_{h-1}^\ell,a_{h-1}^\ell)) \right] \right].$$

Now we would like to show that the above is $\frac{1}{2}\mathbb{I}_\ell^{\pi_{\mathrm{TS}}^\ell}(\tilde{\mathcal{E}}_\ell^*; \mathcal{H}_{\ell,H})$. To be more careful in our arguments, we need to be reminded of what the random variable $\mathcal{H}_{\ell,H}$ is. We must view it as $\mathcal{H}_{\ell,H} = \mathcal{H}_{\ell,H}(\mathcal{E}, \pi_{\mathrm{TS}}^\ell)$ or $\mathcal{H}_{\ell,H}(\mathcal{E}, \pi_{\mathcal{E}_{\mathrm{TS}}}^*)$ where $\mathcal{E}, \mathcal{E}_{\mathrm{TS}}$ are two independent samples of $\mathbb{P}_\ell(\cdot)$, and $\mathcal{E}$ represents the same $\mathcal{E}$ in the regret above in $\pi_\mathcal{E}^*$, i.e. the true environment. Also note that $\mathcal{E}, \tilde{\mathcal{E}}_\ell^*$ are dependent, as we set $\tilde{\mathcal{E}}_\ell^*, \mathcal{E}$ to have the same $\zeta$ value.

**Lemma 10.** *With $\tilde{\mathcal{E}}_\ell^*$ defined according to the proof in Appendix C, in a Bayesian RL, we have*

$$\mathbb{I}_\ell^{\pi_{\mathrm{TS}}^\ell}(\tilde{\mathcal{E}}_\ell^*; \mathcal{H}_{\ell,H}) = \sum_{h=1}^{H} \mathbb{E}_\ell \left[ \mathbb{E}_{\pi_{\mathrm{TS}}^\ell}^{\bar{\mathcal{E}}_\ell} \left[ D_{\mathrm{KL}} \left( (P_h^{\tilde{\mathcal{E}}_\ell^*} \otimes r_h^{\tilde{\mathcal{E}}_\ell^*})(\cdot|s_{h-1}^\ell,a_{h-1}^\ell)||(P_h^{\bar{\mathcal{E}}_\ell} \otimes r_h^{\bar{\mathcal{E}}_\ell})(\cdot|s_{h-1}^\ell,a_{h-1}^\ell)) \right] \right]$$

*As a special case, when rewards are deterministic, we have*

$$\mathbb{I}_\ell^{\pi_{\mathrm{TS}}^\ell}(\tilde{\mathcal{E}}_\ell^*; \mathcal{H}_{\ell,H}) = \sum_{h=1}^{H} \mathbb{E}_\ell \left[ \mathbb{E}_{\pi_{\mathrm{TS}}^\ell}^{\bar{\mathcal{E}}_\ell} \left[ D_{KL}(P_h^{\tilde{\mathcal{E}}_\ell^*}(\cdot|s_h,a_h)||P_h^{\bar{\mathcal{E}}_\ell}(\cdot|s_h,a_h)) \right] \right].$$

*Proof.* Using the chain rule of mutual information,

$$\begin{aligned}
\mathbb{I}_\ell^{\pi_{\mathrm{TS}}^\ell}(\tilde{\mathcal{E}}_\ell^*; \mathcal{H}_{\ell,H}) &= \sum_{h=1}^{H} \mathbb{I}_\ell^{\pi_{\mathrm{TS}}^\ell} \left( \tilde{\mathcal{E}}_\ell^*; (s_h^\ell, a_h^\ell, r_h^\ell)|\mathcal{H}_{\ell,h-1} \right) \\
&= \sum_{h=1}^{H} \mathbb{I}_\ell^{\pi_{\mathrm{TS}}^\ell} \left( \tilde{\mathcal{E}}_\ell^*; s_h^\ell|\mathcal{H}_{\ell,h-1} \right) + \sum_{h=1}^{H} \mathbb{I}_\ell^{\pi_{\mathrm{TS}}^\ell} \left( \tilde{\mathcal{E}}_\ell^*; a_h^\ell|s_h^\ell, \mathcal{H}_{\ell,h-1} \right) \\
&\quad + \sum_{h=1}^{H} \mathbb{I}_\ell^{\pi_{\mathrm{TS}}^\ell} \left( \tilde{\mathcal{E}}_\ell^*; r_h^\ell|s_h^\ell, a_h^\ell, \mathcal{H}_{\ell,h-1} \right).
\end{aligned} \tag{92}$$

Let us note what is meant by $\mathbb{I}_{\ell}^{\pi_{\mathrm{TS}}^{\ell}}(\cdot)$ is $\mathbb{I}_{\ell}(\cdot|\pi_{\mathrm{TS}}^{\ell})$. In [18], the policy $\pi$ used is fixed/independent from the random variables involved in the mutual information (given $\mathcal{D}_{\ell}$). Here, the same holds as $\pi_{\mathrm{TS}}^{\ell}$ and $\tilde{\mathcal{E}}_{\ell}^{*}$ are independent.

- For the first term in Eq. (92), by using $\mathbb{I}(X;Y) = \int D_{\mathrm{KL}}(P(Y|x)||P(Y))\,\mathrm{d}\mathbb{P}(x)$, and the definition of conditional mutual information, we have

$$
\mathbb{I}_{\ell}\left(\tilde{\mathcal{E}}_{\ell}^{*}; s_{h}^{\ell}|\mathcal{H}_{\ell,h-1}, \pi_{\mathrm{TS}}^{\ell}\right)
$$
$$
= \int\int D_{\mathrm{KL}}\left(\mathbb{P}_{\ell}\left(s_{h}^{\ell} = \cdot|\mathcal{H}_{\ell,h-1}, \pi_{\mathcal{E}_{\mathrm{TS}}}^{*}, \tilde{\mathcal{E}}_{\ell}^{*}\right)||\mathbb{P}_{\ell}\left(s_{h}^{\ell} = \cdot|\mathcal{H}_{\ell,h-1}, \pi_{\mathcal{E}_{\mathrm{TS}}}^{*}\right)\right)
$$
$$
\mathrm{d}\mathbb{P}_{\ell}(\tilde{\mathcal{E}}_{\ell}^{*}|\mathcal{H}_{\ell,h-1}, \pi_{\mathcal{E}_{\mathrm{TS}}}^{*})\,\mathrm{d}\mathbb{P}_{\ell}(\mathcal{H}_{\ell,h-1}, \pi_{\mathcal{E}_{\mathrm{TS}}}^{*}) \tag{93}
$$
$$
= \int\int D_{\mathrm{KL}}\left(P_{h}^{\tilde{\mathcal{E}}_{\ell}^{*}}\left(\cdot|s_{h-1}^{\ell}, a_{h-1}^{\ell}\right)||\mathbb{P}_{\ell}\left(s_{h}^{\ell} = \cdot|\mathcal{H}_{\ell,h-1}, \pi_{\mathcal{E}_{\mathrm{TS}}}^{*}\right)\right)
$$
$$
\mathrm{d}\mathbb{P}_{\ell}(\tilde{\mathcal{E}}_{\ell}^{*}|\mathcal{H}_{\ell,h-1}, \pi_{\mathcal{E}_{\mathrm{TS}}}^{*})\,\mathrm{d}\mathbb{P}_{\ell}(\mathcal{H}_{\ell,h-1}, \pi_{\mathcal{E}_{\mathrm{TS}}}^{*}).
$$

Where we substituted $\mathbb{P}_{\ell}\left(s_{h}^{\ell} = \cdot|\mathcal{H}_{\ell,h-1}, \pi_{\mathcal{E}_{\mathrm{TS}}}^{*}, \tilde{\mathcal{E}}_{\ell}^{*}\right) = P_{h}^{\tilde{\mathcal{E}}_{\ell}^{*}}\left(\cdot|s_{h-1}^{\ell}, a_{h-1}^{\ell}\right)$. Let us see why this is the case. Let us analyze the meaning of the conditional on $(\mathcal{H}_{\ell,h-1}, \pi_{\mathcal{E}_{\mathrm{TS}}}^{*}, \tilde{\mathcal{E}}_{\ell}^{*})$. Recall that $\mathcal{H}_{\ell,h-1} = \mathcal{H}_{\ell,h-1}(\mathcal{E}, \pi_{\mathcal{E}_{\mathrm{TS}}}^{*})$. Since $\tilde{\mathcal{E}}_{\ell}^{*}$ is given, the random variable $\mathcal{E}$ can only go over the partition $\Theta_{\zeta(\tilde{\mathcal{E}}_{\ell}^{*})}^{\varepsilon}$. Of course, we can also drop all conditionals on previous state transitions, except for the last one $s_{h-1}^{\ell}(\mathcal{E}), a_{h-1}^{\ell}(\mathcal{E})$. Note that the policy $\pi_{\mathcal{E}_{\mathrm{TS}}}^{*}$ is also irrelevant in this conditional, since the next state only depends on probability transitions and not on policy, hence why also we are not using the full notation $s_{h-1}^{\ell}(\mathcal{E}, \pi_{\mathcal{E}_{\mathrm{TS}}}^{*}), a_{h-1}^{\ell}(\mathcal{E}, \pi_{\mathcal{E}_{\mathrm{TS}}}^{*})$. This implies

$$
\mathbb{P}_{\ell}\left(s_{h}^{\ell}(\mathcal{E}) = \cdot|\mathcal{H}_{\ell,h-1}, \pi_{\mathcal{E}_{\mathrm{TS}}}^{*}, \tilde{\mathcal{E}}_{\ell}^{*}\right) = \mathbb{P}_{\ell}\left(s_{h}^{\ell}(\mathcal{E}) = \cdot|s_{h-1}^{\ell}(\mathcal{E}), a_{h-1}^{\ell}(\mathcal{E}), \tilde{\mathcal{E}}_{\ell}^{*}\right) \tag{94}
$$
$$
= \int_{\mathcal{E}} P_{h}^{\mathcal{E}}\left(s_{h}^{\ell} = \cdot|s_{h-1}^{\ell}, a_{h-1}^{\ell}\right)\mathrm{d}\mathbb{P}_{\ell}(\mathcal{E}|\zeta(\mathcal{E}) \tag{95}
$$
$$
= \zeta(\tilde{\mathcal{E}}_{\ell}^{*})) \tag{96}
$$
$$
= P_{h}^{\mathbb{E}_{\ell}[\mathcal{E}|\zeta(\mathcal{E})=\zeta(\tilde{\mathcal{E}}_{\ell}^{*})]}\left(s_{h}^{\ell} = \cdot|s_{h-1}^{\ell}, a_{h-1}^{\ell}\right) \tag{97}
$$

However, recall that we defined $\tilde{\mathcal{E}}_{\ell}^{*}$ to be the posterior mean of $\mathcal{E}$ over $\Theta_{k}^{\varepsilon}$, i.e. $\mathbb{E}_{\ell}[\mathcal{E}|\zeta(\mathcal{E}) = \zeta(\tilde{\mathcal{E}}_{\ell}^{*})] = \tilde{\mathcal{E}}_{\ell}^{*}$. Hence, the average above yields $P_{h}^{\tilde{\mathcal{E}}_{\ell}^{*}}\left(\cdot|s_{h-1}^{\ell}, a_{h-1}^{\ell}\right)$, as desired. Next, for the second term in the KL-divergence,

$$
\mathbb{P}_{\ell}\left(s_{h}^{\ell}(\mathcal{E}) = \cdot|\mathcal{H}_{\ell,h-1}, \pi_{\mathcal{E}_{\mathrm{TS}}}^{*}\right) = \int\mathbb{P}_{\ell}\left(s_{h}^{\ell}(\mathcal{E}) = \cdot|\mathcal{H}_{\ell,h-1}, \pi_{\mathcal{E}_{\mathrm{TS}}}^{*}, \mathcal{E}\right)\mathrm{d}\mathbb{P}_{\ell}(\mathcal{E}|\mathcal{H}_{\ell,h-1}, \pi_{\mathcal{E}_{\mathrm{TS}}}^{*})
$$
$$
= \int P_{h}^{\mathcal{E}}(s_{h}^{\ell} = \cdot|s_{h-1}^{\ell}, a_{h-1}^{\ell})\,\mathrm{d}\mathbb{P}_{\ell}(\mathcal{E}|\mathcal{H}_{\ell,h-1}, \pi_{\mathcal{E}_{\mathrm{TS}}}^{*})
$$
$$
= \int P_{h}^{\mathcal{E}}(s_{h}^{\ell} = \cdot|s_{h-1}^{\ell}, a_{h-1}^{\ell})\,\mathrm{d}\mathbb{P}_{\ell}(\mathcal{E})
$$
$$
= P_{h}^{\bar{\mathcal{E}}_{\ell}}\left(s_{h}^{\ell} = \cdot|s_{h-1}^{\ell}, a_{h-1}^{\ell}\right). \tag{98}
$$

In the above equations, $\mathcal{H}_{\ell,h-1}, \pi_{\mathcal{E}_{\mathrm{TS}}}^{*}$ are given in the conditional, and the true environment $\mathcal{E}$ is being integrated. The second equality was explained in the previous case. Let us explain why $\mathrm{d}\mathbb{P}_{\ell}(\mathcal{E}|\mathcal{H}_{\ell,h-1}, \pi_{\mathcal{E}_{\mathrm{TS}}}^{*}) = \mathrm{d}\mathbb{P}_{\ell}(\mathcal{E})$ in the third equality. Due to the independence of priors over different layers, the conditional on $\mathcal{H}_{\ell,h-1}$ impacts transition functions of prior layers (i.e. $P_{1}^{\mathcal{E}}, \ldots, P_{h-1}^{\mathcal{E}}$), while the transition function in question is the one at layer $h$. Therefore, this conditional can be dropped, as well as $\pi_{\mathcal{E}_{\mathrm{TS}}}^{*}$ since $\mathcal{E}_{\mathrm{TS}}, \mathcal{E}$ are two independent samples of $\mathbb{P}_{\ell}$. Finally, the last equation is by the definition of probability kernel $P_{h}^{\bar{\mathcal{E}}_{\ell}}$. Eqs. (93) and (98) imply $\mathbb{I}_{\ell}^{\pi_{\mathrm{TS}}^{\ell}}\left(\tilde{\mathcal{E}}_{\ell}^{*}; s_{h}^{\ell}|\mathcal{H}_{\ell,h-1}, \pi_{\mathcal{E}_{\mathrm{TS}}}^{*}\right) = $

$$
\int\int D_{\mathrm{KL}}\left(P_{h}^{\tilde{\mathcal{E}}_{\ell}^{*}}(\cdot|s_{h-1}^{\ell}, a_{h-1}^{\ell})||P_{h}^{\bar{\mathcal{E}}_{\ell}}(\cdot|s_{h-1}^{\ell}, a_{h-1}^{\ell})\right)\mathrm{d}\mathbb{P}_{\ell}(\tilde{\mathcal{E}}_{\ell}^{*})\,\mathrm{d}\mathbb{P}_{\ell}(\mathcal{H}_{\ell,h-1}, \pi_{\mathcal{E}_{\mathrm{TS}}}^{*}). \tag{99}
$$

where we note we also dropped the conditionals on $\mathcal{H}_{\ell,h-1}, \pi^*_{\mathcal{E}_{\mathrm{TS}}}$ in $\mathrm{d}\mathbb{P}_\ell(\tilde{\mathcal{E}}^*_\ell)$, by the similar argument in the previous case for $\mathrm{d}\mathbb{P}_\ell(\mathcal{E}|\mathcal{H}_{\ell,h-1}, \pi^*_{\mathcal{E}_{\mathrm{TS}}})$ as the integrand is transitions at the $h$−th step. We continue by focusing on the outer integral with respect to

$$\mathrm{d}\mathbb{P}_\ell(\mathcal{H}_{\ell,h-1}, \pi^*_{\mathcal{E}_{\mathrm{TS}}}) = P(\mathcal{H}_{\ell,h-1}(\mathcal{E}, \pi^*_{\mathcal{E}_{\mathrm{TS}}})|\mathcal{E}, \pi^*_{\mathcal{E}_{\mathrm{TS}}})\, \mathrm{d}\mu^{\otimes(h-1)}_{\mathcal{S}\times\mathcal{A}}\, \mathrm{d}\mathbb{P}_\ell(\mathcal{E})\, \mathrm{d}\mathbb{P}_\ell(\pi^*_{\mathcal{E}_{\mathrm{TS}}}) \quad (100)$$

and note that since only transitions at the $(h-1)$-th step are inside the inner integral, one can marginalize prior $(h-2, \ldots, 1)$ state-action-reward tuples, yielding

$$= \int_{s^\ell_{h-1}, a^\ell_{h-1}, \mathcal{E}, \pi^*_{\mathcal{E}_{\mathrm{TS}}}} P(s^\ell_{h-1}, a^\ell_{h-1}|\mathcal{E}, \pi^*_{\mathcal{E}_{\mathrm{TS}}})$$

$$\left( \int_{\tilde{\mathcal{E}}^*_\ell} D_{\mathrm{KL}} \left( P^{\tilde{\mathcal{E}}^*_\ell}_h(\cdot|s^\ell_{h-1}, a^\ell_{h-1}) || P^{\bar{\mathcal{E}}_\ell}_h(\cdot|s^\ell_{h-1}, a^\ell_{h-1}) \right) \mathrm{d}\mathbb{P}_\ell(\tilde{\mathcal{E}}^*_\ell) \right)$$

$$\mathrm{d}\mu_{\mathcal{S}\times\mathcal{A}}\, \mathrm{d}\mathbb{P}_\ell(\mathcal{E})\, \mathrm{d}\mathbb{P}_\ell(\pi^*_{\mathcal{E}_{\mathrm{TS}}})$$

$$= \int_{s^\ell_{h-1}, a^\ell_{h-1}, \pi^*_{\mathcal{E}_{\mathrm{TS}}}, \tilde{\mathcal{E}}^*_\ell} \left( \int_{\mathcal{E}} P(s^\ell_{h-1}, a^\ell_{h-1}|\mathcal{E}, \pi^*_{\mathcal{E}_{\mathrm{TS}}})\, \mathrm{d}\mathbb{P}_\ell(\mathcal{E}) \right)$$

$$D_{\mathrm{KL}} \left( P^{\tilde{\mathcal{E}}^*_\ell}_h(\cdot|s^\ell_{h-1}, a^\ell_{h-1}) || P^{\bar{\mathcal{E}}_\ell}_h(\cdot|s^\ell_{h-1}, a^\ell_{h-1}) \right) \mathrm{d}\mathbb{P}_\ell(\tilde{\mathcal{E}}^*_\ell, \pi^*_{\mathcal{E}_{\mathrm{TS}}})\, \mathrm{d}\mu_{\mathcal{S}\times\mathcal{A}}$$

where we simply rearranged the measures and integrals, and note the independence $\mathbb{P}_\ell(\tilde{\mathcal{E}}^*_\ell)\mathbb{P}_\ell(\pi^*_{\mathcal{E}_{\mathrm{TS}}}) = \mathbb{P}_\ell(\tilde{\mathcal{E}}^*_\ell, \pi^*_{\mathcal{E}_{\mathrm{TS}}})$. For the outer integral, notice that $P(s^\ell_{h-1}, a^\ell_{h-1}|\mathcal{E}, \pi^*_{\mathcal{E}_{\mathrm{TS}}}) = d^{\mathcal{E}}_{h,\pi^*_{\mathcal{E}_{\mathrm{TS}}}}(s^\ell_{h-1}, a^\ell_{h-1})$ by definition. So using the linearity of expectation and independence of priors over different layers

$$\int_{\mathcal{E}} d^{\mathcal{E}}_{h,\pi^*_{\mathcal{E}_{\mathrm{TS}}}}(s^\ell_{h-1}, a^\ell_{h-1})\, \mathrm{d}\mathbb{P}_\ell(\mathcal{E}) = d^{\bar{\mathcal{E}}_\ell}_{h,\pi^*_{\mathcal{E}_{\mathrm{TS}}}}(s^\ell_{h-1}, a^\ell_{h-1}). \quad (101)$$

Putting it all together, and going back to the notation $\pi^*_{\mathcal{E}_{\mathrm{TS}}} \to \pi^\ell_{\mathrm{TS}}$ :

$$\mathbb{I}_\ell \left( \tilde{\mathcal{E}}^*_\ell; s^\ell_h | \mathcal{H}_{\ell,h-1}, \pi^\ell_{\mathrm{TS}} \right)$$

$$= \int_{s^\ell_{h-1}, a^\ell_{h-1}, \pi^\ell_{\mathrm{TS}}, \tilde{\mathcal{E}}^*_\ell} d^{\bar{\mathcal{E}}_\ell}_{h,\pi^\ell_{\mathrm{TS}}}(s^\ell_{h-1}, a^\ell_{h-1})$$

$$D_{\mathrm{KL}} \left( P^{\tilde{\mathcal{E}}^*_\ell}_h(\cdot|s^\ell_{h-1}, a^\ell_{h-1}) || P^{\bar{\mathcal{E}}_\ell}_h(\cdot|s^\ell_{h-1}, a^\ell_{h-1}) \right) \mathrm{d}\mathbb{P}_\ell(\tilde{\mathcal{E}}^*_\ell, \pi^\ell_{\mathrm{TS}})\, \mathrm{d}\mu_{\mathcal{S}\times\mathcal{A}}$$

$$= \int_{\pi^\ell_{\mathrm{TS}}, \tilde{\mathcal{E}}^*_\ell} \mathbb{E}^{\bar{\mathcal{E}}_\ell}_{\pi^\ell_{\mathrm{TS}}} \left[ D_{\mathrm{KL}} \left( P^{\tilde{\mathcal{E}}^*_\ell}_h(\cdot|s^\ell_{h-1}, a^\ell_{h-1}) || P^{\bar{\mathcal{E}}_\ell}_h(\cdot|s^\ell_{h-1}, a^\ell_{h-1}) \right) \right] \mathrm{d}\mathbb{P}_\ell(\tilde{\mathcal{E}}^*_\ell, \pi^\ell_{\mathrm{TS}})$$

$$= \mathbb{E}_\ell \left[ \mathbb{E}^{\bar{\mathcal{E}}_\ell}_{\pi^\ell_{\mathrm{TS}}} \left[ D_{\mathrm{KL}} \left( P^{\tilde{\mathcal{E}}^*_\ell}_h(\cdot|s^\ell_{h-1}, a^\ell_{h-1}) || P^{\bar{\mathcal{E}}_\ell}_h(\cdot|s^\ell_{h-1}, a^\ell_{h-1}) \right) \right] \right] ,$$

where $\mathbb{E}^{\bar{\mathcal{E}}_\ell}_{\pi^\ell_{\mathrm{TS}}}$ is taken with respect to $s^\ell_{h-1}, a^\ell_{h-1}$ and $\mathbb{E}_\ell$ is taken with respect to $\pi^\ell_{\mathrm{TS}}, \tilde{\mathcal{E}}^*_\ell$.

- For the second term in Eq. (92),

$$\mathbb{I}_\ell \left( \tilde{\mathcal{E}}^*_\ell; a^\ell_h | s^\ell_h, \mathcal{H}_{\ell,h-1}, \pi^\ell_{\mathrm{TS}} \right)$$

$$= \int \int D_{\mathrm{KL}} \left( \mathbb{P}_\ell \left( a^\ell_h = \cdot | s^\ell_h, \mathcal{H}_{\ell,h-1}, \pi^\ell_{\mathrm{TS}}, \tilde{\mathcal{E}}^*_\ell \right) || \mathbb{P}_\ell \left( a^\ell_h = \cdot | s^\ell_h, \mathcal{H}_{\ell,h-1}, \pi^\ell_{\mathrm{TS}} \right) \right) .$$

where the integrals are with respect to $\mathrm{d}\mathbb{P}_\ell(\tilde{\mathcal{E}}^*_\ell | s^\ell_h, \mathcal{H}_{\ell,h-1}, \pi^\ell_{\mathrm{TS}})\, \mathrm{d}\mathbb{P}_\ell(s^\ell_h, \mathcal{H}_{\ell,h-1}, \pi^\ell_{\mathrm{TS}})$. When $s^\ell_h, \pi^\ell_{\mathrm{TS}}$ are given, both sides of the KL term are equal to $\pi^\ell_{\mathrm{TS}}(\cdot|s^\ell_h)$ and thus the above is zero.

- For the third term, we use an argument similar to the first term to see that

$$\mathbb{I}_\ell \left( \tilde{\mathcal{E}}^*_\ell; r^\ell_h | s^\ell_h, a^\ell_h, \mathcal{H}_{\ell,h-1}, \pi^\ell_{\mathrm{TS}} \right) = \mathbb{E}_\ell \left[ \mathbb{E}^{\bar{\mathcal{E}}_\ell}_{\pi^\ell_{\mathrm{TS}}} \left[ D_{\mathrm{KL}} \left( r^{\tilde{\mathcal{E}}^*_\ell}_h(\cdot|s^\ell_h, a^\ell_h) || r^{\bar{\mathcal{E}}_\ell}_h(\cdot|s^\ell_h, a^\ell_h) \right) \right] \right] .$$

$$(102)$$

Put together, it follows that

$$
\mathbb{I}_\ell^{\pi_{\mathrm{TS}}^\ell}\left(\tilde{\mathcal{E}}_\ell^*; \mathcal{H}_{\ell,H}\right)
$$

$$
= \sum_{h=1}^{H} \mathbb{E}_\ell\left[\mathbb{E}_{\pi_{\mathrm{TS}}^\ell}^{\bar{\mathcal{E}}_\ell}\left[D_{\mathrm{KL}}\left(P_h^{\tilde{\mathcal{E}}_\ell^*}(\cdot|s_{h-1}^\ell, a_{h-1}^\ell)||P_h^{\bar{\mathcal{E}}_\ell}(\cdot|s_{h-1}^\ell, a_{h-1}^\ell)\right)\right]\right]
$$

$$
+ \mathbb{E}_\ell\left[\mathbb{E}_{\pi_{\mathrm{TS}}^\ell}^{\bar{\mathcal{E}}_\ell}\left[D_{\mathrm{KL}}\left(r_h^{\tilde{\mathcal{E}}_\ell^*}(\cdot|s_h^\ell, a_h^\ell)||r_h^{\bar{\mathcal{E}}_\ell}(\cdot|s_h^\ell, a_h^\ell)\right)\right]\right]
$$

$$
= \sum_{h=1}^{H} \mathbb{E}_\ell\left[\mathbb{E}_{\pi_{\mathrm{TS}}^\ell}^{\bar{\mathcal{E}}_\ell}\left[D_{\mathrm{KL}}\left((P_h^{\tilde{\mathcal{E}}_\ell^*} \otimes r_h^{\tilde{\mathcal{E}}_\ell^*})(\cdot|s_{h-1}^\ell, a_{h-1}^\ell)||(P_h^{\bar{\mathcal{E}}_\ell} \otimes r_h^{\bar{\mathcal{E}}_\ell})(\cdot|s_{h-1}^\ell, a_{h-1}^\ell)\right)\right]\right]. \quad \square
$$

### J.1 On the rewrite of mutual information in [18, App. B.5]

We start by citing the relevant equations involved in [18, App. B.5]. For $\Sigma_h = \mathbb{E}_\ell\left[\mathbb{E}_{\pi^*}^{\bar{\mathcal{E}}_\ell^*}\left[\phi(s_h^\ell, a_h^\ell)\right]\mathbb{E}_{\pi^*}^{\bar{\mathcal{E}}_\ell^*}\left[\phi(s_h^\ell, a_h^\ell)^\top\right]\right]$, the authors claim

$$
\sum_{h=1}^{H} \mathbb{E}_\ell\left[\left\|\Sigma_h^{1/2}\sum_{s'}(\psi_h^{\tilde{\mathcal{E}}_\ell^*}(s') - \psi_h^{\bar{\mathcal{E}}_\ell^*}(s'))V_{h+1,\pi^*}^{\tilde{\mathcal{E}}_\ell^*}(s')\right\|_2^2\right] \tag{103}
$$

$$
= \mathbb{E}_\ell\left[\sum_{h=1}^{H}\mathbb{E}_{\pi^*}^{\bar{\mathcal{E}}_\ell}\left[\left(P_h^{\tilde{\mathcal{E}}_\ell^*}(\cdot|s_h^\ell, a_h^\ell)^\top V_{h+1,\pi^*}^{\tilde{\mathcal{E}}_\ell^*}(\cdot) - P_h^{\bar{\mathcal{E}}_\ell^*}(\cdot|s_h^\ell, a_h^\ell)^\top V_{h+1,\pi^*}^{\tilde{\mathcal{E}}_\ell^*}(\cdot)\right)^2\right]\right] \tag{104}
$$

$$
\leq \frac{1}{2}\sum_{h=1}^{H}\mathbb{E}_\ell\left[\mathbb{E}_{\pi_{\mathcal{E}}^*}^{\bar{\mathcal{E}}_\ell}\left[D_{\mathrm{KL}}\left(P_h^{\tilde{\mathcal{E}}_\ell^*}(\cdot|s_{h-1}^\ell, a_{h-1}^\ell)||P_h^{\bar{\mathcal{E}}_\ell}(\cdot|s_{h-1}^\ell, a_{h-1}^\ell)\right)\right]\right] \tag{105}
$$

$$
= \frac{1}{2}\mathbb{I}_\ell^{\pi^*}\left(\tilde{\mathcal{E}}_\ell^*; \mathcal{H}_{\ell,H}\right). \tag{106}
$$

For the last part in Eq. (105), the authors do not provide a proof, and cite their own [18, Lemma A.1] as support. However, that lemma is for $\mathbb{I}_\ell^\pi(\mathcal{E}; \mathcal{H}_{\ell,H})$, where $\pi$ is the **algorithm** and not the optimal policy of the true environment.

We need a rewrite of $\mathbb{I}_\ell^{\pi_{\mathcal{E}}^*}\left(\tilde{\mathcal{E}}_\ell^*; \mathcal{H}_{\ell,H}\right)$. We emphasize that in the former mutual information expression, the policy involved is the algorithm $\pi$, which clearly is not dependent on the true environment $\mathcal{E}$, unlike $\pi_{\mathcal{E}}^*$. Furthermore, the environment involved is also independent from the policy, but that is not the case here since $\tilde{\mathcal{E}}_\ell^*, \pi_{\mathcal{E}}^*$ are dependent through $\zeta$. As we shall see, it is crucial for the policy in the mutual information expression to be independent from the true environment, in order for the argument in [18, Lemma A.1].

Since our own lemma above for $\mathbb{I}_\ell^{\pi_{\mathrm{TS}}^\ell}\left(\tilde{\mathcal{E}}_\ell^*; \mathcal{H}_{\ell,H}\right)$ naturally extends [18, Lemma A.1] and takes the first step for the substitution of $\mathcal{E}$ by $\tilde{\mathcal{E}}_\ell^*$, we can analyze what happens in our own equations in Lemma 10, assuming we were to take $\pi_{\mathcal{E}}^*$, the optimal policy of the true environment $\mathcal{E}$, instead of $\pi_{\mathrm{TS}}^\ell$. We can apply the mutual information chain rule as before, and focus on

$$
\mathbb{I}_\ell\left(\tilde{\mathcal{E}}_\ell^*; s_h^\ell|\mathcal{H}_{\ell,h-1}, \pi_{\mathcal{E}}^*\right) = \int\int D_{\mathrm{KL}}\left(\mathbb{P}_\ell(s_h^\ell = \cdot|\mathcal{H}_{\ell,h-1}, \pi_{\mathcal{E}}^*, \tilde{\mathcal{E}}_\ell^*)||\mathbb{P}_\ell\left(s_h^\ell = \cdot|\mathcal{H}_{\ell,h-1}, \pi_{\mathcal{E}}^*\right)\right)
$$

$$
\mathrm{d}\mathbb{P}_\ell(\tilde{\mathcal{E}}_\ell^*|\mathcal{H}_{\ell,h-1}, \pi_{\mathcal{E}}^*)\,\mathrm{d}\mathbb{P}_\ell(\mathcal{H}_{\ell,h-1}, \pi_{\mathcal{E}}^*)
$$

Recall that the history is of the form $\mathcal{H}_{\ell,H} = \mathcal{H}_{\ell,H}(\mathcal{E}, \pi_{\mathcal{E}_{\mathrm{TS}}}^*)$. The first thing to prove above should be $\mathbb{P}_\ell\left(s_h^\ell = \cdot|\mathcal{H}_{\ell,h-1}, \pi_{\mathcal{E}}^*, \tilde{\mathcal{E}}_\ell^*\right) = P_h^{\tilde{\mathcal{E}}_\ell^*}\left(\cdot|s_{h-1}^\ell, a_{h-1}^\ell\right)$. However, note that since $\pi_{\mathcal{E}}^*$ is given, this means $\mathcal{E}$ is given, at least in (realistic) scenarios where there is uniqueness of optimal policies, and as a result, the true environment $\mathcal{E}$ in $\mathcal{H}_{\ell,H}(\mathcal{E}, \pi_{\mathcal{E}_{\mathrm{TS}}}^*)$ is determined uniquely. This implies that in fact $\mathbb{P}_\ell\left(s_h^\ell = \cdot|\mathcal{H}_{\ell,h-1}, \pi_{\mathcal{E}}^*, \tilde{\mathcal{E}}_\ell^*\right) = P_h^{\mathcal{E}}\left(s_h^\ell = \cdot|s_{h-1}^\ell, a_{h-1}^\ell\right)$. In general it would be the average $P_h^{\mathbb{E}[\mathcal{E}'|\pi_{\mathcal{E}'}^* = \pi_{\mathcal{E}}^*, \zeta(\mathcal{E}') = \zeta(\tilde{\mathcal{E}}_\ell^*)]}\left(s_h^\ell = \cdot|s_{h-1}^\ell, a_{h-1}^\ell\right)$.

Either way, in the very first step, we have shown that the dependence of the policy with the true environment can alter significantly the rewrite of the mutual information by the argument in [18, Lemma A.1]. Clearly, this does not lead to the desired rewrite in Eq. (105) and makes this claimed bound of the Bayesian regret by that mutual information (at the very least) unproven.

A more direct way to note the gap in the argument is the following. Recall that the denominator in the (surrogate) information ratio is supposed to represent the information gain by the algorithm on the true (or surrogate) environment. The surrogate mutual information ratio that one should bound is:

$$\frac{(\mathbb{E}_\ell \left[ V_{1,\pi_{\tilde{\mathcal{E}}}^*}^{\tilde{\mathcal{E}}_\ell^*}(s_1^\ell) - V_{1,\pi}^{\tilde{\mathcal{E}}_\ell^*}(s_1^\ell) \right])^2}{\mathbb{I}_\ell^\pi \left( \tilde{\mathcal{E}}_\ell^*; \mathcal{H}_{\ell,H} \right)}, \tag{107}$$

where $\pi$ is the algorithm (and we select $\pi = \pi_{\mathrm{TS}}$). Clearly the algorithm can not know about the true environment $\mathcal{E}$, which makes it questionable to try to bound the surrogate regret $\mathbb{E}_\ell \left[ V_{1,\pi_{\tilde{\mathcal{E}}}^*}^{\tilde{\mathcal{E}}_\ell^*}(s_1^\ell) - V_{1,\pi}^{\tilde{\mathcal{E}}_\ell^*}(s_1^\ell) \right]$ by a mutual information such as $\mathbb{I}_\ell^{\pi_{\tilde{\mathcal{E}}}^*} \left( \tilde{\mathcal{E}}_\ell^*; \mathcal{H}_{\ell,H} \right) := \mathbb{I}_\ell \left( \tilde{\mathcal{E}}_\ell^*; \mathcal{H}_{\ell,H} | \pi_{\tilde{\mathcal{E}}}^* \right)$, where there is assumed knowledge of the true environment in the conditional, as opposed to conditioning on the algorithm itself like in the ratio above. Therefore, the information ratio that the authors in [18, App. B.5] are (implicitly) trying to bound is not the right one.

## K  Posterior consistency

In this section we define the notion of posterior consistency and state Doob's consistency theorem. We start by describing posterior consistency in a general setting.

Let $\mathscr{X}$ be a measure space and for every $n \in \mathbb{N}$, let $X^{(n)}$ be an observation in the sample space $\mathscr{X}^n$ with distribution $P_\theta^{(n)}$ indexed by a parameter $\theta$ belonging to a separable metric space $\Omega$. For instance $X^{(n)}$ might be a sample of size $n$ from a given distribution $P_\theta$ with $P_\theta^{(n)}$ the corresponding product measure. Given a prior $\Pi$ on the Borel sets of $\Omega$, let $\Pi_n(\cdot \mid X^{(n)})$ be the posterior distribution given the observation $X^{(n)}$. Moreover, we assume that there is a measure $P_\theta^{(\infty)}$ on $\mathscr{X}^\infty$ such that $P_\theta^{(n)}$ is equal to the the image $P_\theta^{(\infty)} \circ (X^{(n)})^{-1}$ of the probability measure $P_\theta^{(\infty)}$ when pushed forward onto $\mathscr{X}^{(n)}$. We say an estimator $T := (T_n)_{n=1}^\infty$, where $T_n : \mathscr{X}^{(n)} \to \Omega$ is a measureable function for all $n \geq 1$, is a strongly consistent estimator of $\theta$ if for every $\theta_0 \in \Omega$ and almost every $X^{(\infty)} = (X^{(n)})_{n=1}^\infty$, we have

$$\lim_{n \to \infty} T_n(X^{(n)}) = \theta_0.$$

We can now describe the content of Assumption 1. Let $\Omega := \Theta$ with the measure $\Pi := \rho$ as the prior, and let $\mathscr{X}$ be the space of all single-episode histories. Also let $\theta_0 := \mathcal{E}_0$ and $P_{\theta_0}^{(\ell)}(\mathcal{D}_\ell) := \mathbb{P}(\mathcal{D}_\ell \mid \mathcal{E}_0)$ be the probability of observing the history $\mathcal{D}_\ell$ in the true environment $\mathcal{E}_0$. Existence of the measure $P_\theta^{(\infty)}$ on $\mathscr{X}^\infty$ as described above follows from the fact that for any $l' > l$, we have $P_{\theta_0}^{(\ell')}(\mathcal{D}_\ell) = P_{\theta_0}^{(\ell)}(\mathcal{D}_\ell) := \mathbb{P}(\mathcal{D}_\ell \mid \mathcal{E}_0)$. Assumption 1 states that there exists a strongly consistent estimator of the true environment $T$ such that for almost every environment $\mathcal{E}_0$ and almost every infinite history $\mathcal{D} = (\mathcal{D}_\ell)_{\ell=1}^\infty$ sampled from the environment $\mathcal{E}_0$, we have

$$\lim_{n \to \infty} T_\ell(\mathcal{D}_\ell) = \mathcal{E}_0.$$

The existence of consistent estimators is closely related to the notion of posterior consistency:

**Definition 10.** The posterior distribution $\Pi_n(\cdot \mid X^{(n)})$ is said to be strongly consistent at $\theta_0 \in \Omega$ if for every neighbourhood $U$ of $\theta_0$ and $P_{\theta_0}^{(\infty)}$-almost every $X^{(\infty)}$, we have $\Pi_n(U^c \mid X^{(n)}) \to 0$ where $X^{(n)}$ is the projection of $X^{(\infty)}$ into the space $\mathscr{X}^n$.

Here we state a version of Doob's consistency theorem that we need for our application. (Theorem 6.9 in [17])

**Theorem 11** (Doob's consistency theorem). *If there is a strongly consistent estimator $T_n : \mathscr{X}^{(n)} \to \Omega$, then the posterior is strongly consistent at $\Pi$-almost every $\theta \in \Omega$. In fact, $\int f(\theta') d\Pi_n(\theta' \mid X^{(n)}) \to f(\theta)$, almost surely $[P_\theta^{(\infty)}]$, for $\Pi$-almost every $\theta$ and every $\Pi$-integrable function $f$.*

Note that while this statement is not the exact statement of Theorem 6.9 in [17], it is equivalent to it as discussed in the paragraph following the theorem.

**Corollary 12.** *Given Assumption 1, for any $\Pi$-integrable function $f : \Theta \to \mathbb{R}$ and almost every $\mathcal{D}_\infty$ sampled from true environment $\mathcal{E}_0$, we have*

$$\lim_{\ell \to \infty} \mathbb{E}_\ell[f(\mathcal{E})] = f(\mathcal{E}_0).$$

*Similarly, if $f : \Theta \times \Theta \to \mathbb{R}$ is bounded and $(\Pi \times \Pi)$-integrable, for almost every $\mathcal{D}_\infty$ sampled from true environment $\mathcal{E}_0$, we have*

$$\lim_{\ell \to \infty} \mathbb{E}_\ell[f(\mathcal{E}, \mathcal{E}')] = f(\mathcal{E}_0, \mathcal{E}_0),$$

*where the expectation is taken over all values of $\mathcal{E}$ and $\mathcal{E}'$, sampled according to $\mathbb{P}_\ell$.*

*Proof.* The first statement immediately follows from Assumption 1 and Theorem 11. To prove the second part, we use the first part to see that for any fixed value of $\mathcal{E}' \in \Theta$ and almost every $\mathcal{D}_\infty$, we have

$$\lim_{\ell \to \infty} (\mathbb{E}_\ell)_{\mathcal{E} \sim \mathbb{P}_\ell}[f(\mathcal{E}, \mathcal{E}')] = f(\mathcal{E}_0, \mathcal{E}').$$

Now we use dominated convergence theorem to see that

$$\lim_{\ell \to \infty} (\mathbb{E}_\ell)_{\mathcal{E}, \mathcal{E}' \sim \mathbb{P}_\ell}[f(\mathcal{E}, \mathcal{E}')] = \lim_{\ell \to \infty} (\mathbb{E}_\ell)_{\mathcal{E}' \sim \mathbb{P}_\ell}[f(\mathcal{E}_0, \mathcal{E}')] = f(\mathcal{E}_0, \mathcal{E}_0). \qquad \square$$