# OpenReview forum: "Improved Bayesian Regret Bounds for Thompson Sampling in Reinforcement Learning"
_NeurIPS.cc/2023/Conference — NeurIPS 2023 poster_

### Official Review · Reviewer_DrLb · 2023-06-27

**Soundness:** 3 good
**Presentation:** 2 fair
**Contribution:** 3 good
**Rating:** 6
**Confidence:** 2

**Summary:**

The paper presents a refined analysis of TS in Bayesian regret reinforcement learning. Regret bounds are derived for tabular, linear, and finite mixture MDPs. The paper uses an information theoretical approach: the information ratio, representing the exploration-exploitation trade-off, is analyzed and bounded by the episode length. The paper uses discretization of the environment space defined by fixing a previous proof from the literature. The paper concludes by discussing related work and the optimality of the obtained regret bounds.

**Strengths:**

The paper provides state-of-the-art Bayesian regret bounds for Thompson Sampling in reinforcement learning through a refined analysis of the surrogate environments and information ratio in many different settings, including tabular, linear, and finite mixtures. The obtained bound is general and independent of the dimension of transition/reward function space. Proof sketches are provided.

**Weaknesses:**

The proposed analysis and bounds may have limited applicability to real-world RL problems beyond the considered settings.
The paper does not provide experimental results or empirical validation of the derived bounds.
Some assumptions and prior works are discussed without providing detailed explanations or comparisons.
Overall, the paper makes significant contributions to the theoretical understanding of TS in RL by providing refined analysis and state-of-the-art Bayesian regret bounds in various settings. However, practical applicability and empirical validation of the derived bounds need further investigation.

**Questions:**

Could you please provide more references that specifically highlight the theoretical analysis of TS in the context of RL and Bandit problems? Usually, bayesian regret is easier than frequentist to analyse. Do you know if your approach could be applicable to frequentist regret ?

What is the relationship between the Kolmogorov dimension for $l_1$-distance, the "value partition for surrogate learning," and the time-inhomogeneous Bayesian RL problem?

How does leveraging this relationship help in bounding the information ratio and achieving regret bounds in TS ?

Can concrete estimates of the dimension be provided for linear, tabular, and finite mixtures applications? What is the significance of isolating the contributions of the information ratio and the cumulative mutual information terms?

Can you clarify the specific experimental results or empirical evidence that support the conjecture regarding the optimality of the regret bound when substituting $H$ in place of the variable?

Are there any specific experiments conducted with TS, assuming access to an oracle, that provide insights into the performance of the regret bound in practical scenarios?

You mention related work on bounding Bayesian regret for TS, including the use of confidence regions and algorithms such as UCBVI and OPPO. How does the proposed work in this paper compare to these approaches in terms of regret bounds and optimality? Are there any notable advantages or limitations of the proposed approach compared to the existing literature?


**Limitations:**

Can you provide a motivation for using TS-based policies rather than OFU ones ? If this is about empirical performances, then it makes sense to run some experimental comparisons.

I am not sure to clearly understand what evidence or rationale is provided to support the claim that the regret in surrogate environments serves as a proxy for the main problem.

I think you could elaborate more on the methodology used in this analysis. How does it contribute to a better understanding of the trade-off between exploration and exploitation? Are there any limitations or assumptions associated with this analysis?

minor:
Lemma 11 : fix the X vs x notation

---

> ### Author Rebuttal · Authors · 2023-08-09
>
> Thank you very much for your questions and comments improving our paper. Please read the Author Rebuttal beforehand.
>
> W:
>
> 1. See Q.6 and L.1 for experiments and empirical validations.
>
> 2. See L.4 and replies to reviewers: W.2 of 4ohk and W.1 of bSE3.
>
> Q:
>
> 1. (a) Here are more refs:  [1111.1797] for multi-armed bandit, [1611.06534] for linear bandit. For RL: [1306.0940, 1607.00215] for tabular Bayesian RL problem with variants of TS. Theory of TS in general Bayesian RL only includes [2206.04640] which is the closest to our work.\
> (b) It is yet unclear if our approach could be applied to the frequentist setting, as some of the derivation need the average over the prior. We hope posterior consistency tools will play a role similar to concentration inequalities in spirit in frequentist analysis.
>
> 2. The $l_1$-distance determines the $l_1-$dim of the environments space, and is a bound on the size of any discrete set that approximates this space up to $\epsilon$ error. This surrogate set is used for surrogate learning (Lemma 1). Surrogate learning is meant to act as a proxy to the main learning problem for time-inhomogeneous Bayesian RL (Lemma 2). This proxy regret is then used to bound the main one, expressed by the $d_{l_1}$ estimation of the size of the surrogate set (Theorem 4). More precisely, the regret is isolated into an information ratio and cumulative mutual information terms, and the latter is bounded by the size of the surrogate space. This shows how this relationship helps in regret bounding.
>
> 3. These estimates are in Corollaries 5-7, Tabular: $SAH$; Linear: $l_1-$dim of the feature maps space ; Finite mixtures: $l_1-$dim of the space of mixtures coefficients.
>
> 4. This is a technique introduced in [29]. On the significance of this decomposition of regret: (1) The ratio quantifies the exploration/exploitation trade-off of the algorithm at each step. Using Pinsker's inequality relating expectation and mutual information, one can bound this ratio. (2) The cumulative mutual information terms can always be bounded by entropy measures or relevant dimensions of the environments space. Our work improves and/or finds correct estimates for both terms in different RL settings.
>
> 5. The simulation supporting our conjecture is in [25, Fig 9]. There are 3 sets of data points in that figure, and the one in blue (PSRL) has the best asymptotic behavior of $\tilde{O}(\sqrt{HSAT})$. Note that we added $\sqrt{H}$ to convert to time-inhomogeneous.
>
> 6. Yes, such works include Optimistic PSRL [2209.14414], wherein PSRL with oracle access is the most performant. See also recent practical work in this area [2305.00477] stating: "Our extensive experiments on the Atari benchmark show that PSDRL significantly outperforms previous state-of-the-art randomized value function approaches, its natural model-free counterparts, while being competitive with a state-of-the-art (model-based) reinforcement learning method in both sample efficiency and computational efficiency".
>
> 7. Our work has few reasonable assumptions, but does not assume a special RL setting like tabular. Previous UCB based works achieve competitive regret for frequentist model-free settings, and [1] has shown optimality with regret $\tilde{O}(d\sqrt{HT})$, where $d$ is the functional approximation dimension of the MDP. As frequentist model-free vs Bayesian model-based are quite different, one must be careful in comparing them. Frequentist optimal/lower regret bounds can be larger than Bayesian regret bounds; Our work achieves a regret $\tilde{O}(H\sqrt{d_{l_1}T})$, and conjectures the lower bound $\tilde{O}(\sqrt{Hd_{l_1}T})$, while not being in contradiction with the optimal bound $\tilde{O}(d\sqrt{HT})$ in frequentist model-free setting. In terms of limitations of TS: the main one is the oracle access to an optimal policy, which cannot always be satisfied efficiently. Nevertheless, clever engineering can make TS work even in large scale Deep RL, as cited above.
>
> L:
>
> 1. We cite the paper "why is PS better than optimism for RL?" directly addressing this: "Computational results...demonstrate that PSRL dramatically outperforms existing algorithms based on OFU". In addition, more up-to-date works on PSRL [2209.14414, Fig. 1,3] compares this recent OFU based UCBVI/UCBVI-B wherein PSRL shows the lowest regret. These experiments provide enough support for the empirical performance of TS.
>
> 2. We address this by explaining the intuition and evidence. Intuition: The discrete surrogate environments set $\Theta^\epsilon$ are built to approximate the original environments $\Theta$ up to a chosen accuracy $\epsilon$. The rationale behind building such a set is that any regret bound on this set is, up to some  error related to $\epsilon$, a regret bound for $\Theta$, hence the use of the phrase "proxy for the main problem". Evidence: The supporting evidence is mathematically laid out in Section 4 and 5, culminating in (Theorem 4), where it is shown that the regret of the main problem scales by the size of $\Theta^\epsilon$. This is established by replacing every environment of the main problem by their $\epsilon$-close surrogate environment, and analyzing the regret in the surrogate space (Lemma 2). This proves the intuition behind the surrogate serving as a proxy to the main problem.
>
> 3. This reply is complementary to Q.3. We demonstrate how this trade-off is better understood in terms of the $l_1-$dim of $\Theta$, and a new notion called $\lambda$, the value diameter. Using posterior consistency tools, we achieve sublinear dim scaling $d_{l_1}^{1/2}$ (vs linear in previous tabular works). We also make a conceptual contribution by showing that, unlike previous studies, the number of time-steps $H$ is not the right quantity to bound the information ratio. Instead, it is the value diameter.
>
> 4. On limitations/assumptions, as noted in Section 5, the posterior consistency assumption is needed to show $d_{l_1}^{1/2}$ scaling for the information ratio bound.

---

> > ### Comment · Reviewer_DrLb · 2023-08-13
> >
> > I have read the other reviews and the rebuttal. I am satisfied with the answer provided by the authors. I will not change my score.

---

### Official Review · Reviewer_z7jd · 2023-07-05

**Soundness:** 3 good
**Presentation:** 3 good
**Contribution:** 3 good
**Rating:** 7
**Confidence:** 2

**Summary:**

The paper shows a refined analysis of Thompson Sampling in RL. The analysis leverages the notion of Kolmogorov dimension, and results in an improved rate of the regret. The authors further presented the bounds in terms of several specific settings, which match the state-of-the-art results.

**Strengths:**

The writing of the paper is clear and easy to follow. The paper studies an important problem in RL. The notion of Kolmogorov dimension as well as the corresponding analysis is novel. The paper also presents a complete discussion for specific settings, and makes sufficient comparisons with previous works, which shows the significance of its results.

**Weaknesses:**

1. While the bound relies on a new notion $\lambda$, the term can still be as big as $H$ in many cases. It hard to to quantify how much improvement is made by this notion, and therefore, in the discussion of specific RL settings, the corresponds bounds only match the state-of-the-art results, but don't improve them.

2. The paper doesn't provide a lower bound in terms of the proposed notion, which weakens the significance of this notion.

**Questions:**

1. The authors mentioned the incorrectness of [12] in terms of surrogate loss. Can the authors make some discussion on it and how did they fix the issue?

**Limitations:**

The paper has addressed it limitations.

---

> ### Author Rebuttal · Authors · 2023-08-09
>
> We thank the reviewer for their comments, which we will include accordingly in our final revision as detailed below. We also invite the reviewer to read the author rebuttal beforehand. Below are our replies:
>
> **Weaknesses**:
>
> 1. Thank you for the comment. Given other reviewer's remarks, we have realized that we need to discuss a bit more the potential conceptual and practical improvement brought about by $\lambda$. A summary of this reply will appear in the revised version.\
> The value-diameter $\lambda$ is bounded by another notion of environment-diameter, which is roughly speaking the longest path among the shortest paths between any two states. If states can be reached from one another quickly in environments, then $\lambda$ can be far smaller than $H$ (as alluded to after Definition 10). Perhaps more importantly, the conceptual contribution is the realization that the information ratio is not bounded by the term $H$ and $d\_{l\_1}$, but the value-diameter $\lambda$ and $d\_{l\_1}$. While it is true that $\lambda$ can be always upper bounded by $H$, it is important to recognize the nature of what is bounding the information ratio, which is not $H$, but the value-diameter; this opens the door to further optimizations in specific cases, one of which we just mentioned. Regarding the discussion of specific RL settings and the significance of our results, we would like to note again that our results generalize to nonlinear settings, and further, we achieve the first correct and state-of-the-art regret bounds for the Bayesian linear and finite mixtures settings.
>
> 2. Thank you for the comment. We understand your statement as asking for a regret lower bound based on the notion $\lambda$. We thank you for reminding us, as we have mentioned this as a limitation of our work (after the section Conclusion). We will include this more explicitly as an open problem for future work in our revised version.
>
> **Questions**:
>
> 1. Thank you for your question. We assume that you are referring to surrogate "environments"/"regret", as there is not a notion of surrogate "loss" in our paper (or in the related work [12]). We will include a summary of our reply below in our revised paper to provide an outline of this matter.\
> We recall that the goal is to construct surrogate environments that have a surrogate regret which approximates that of every environment in the same $\varepsilon-$value partition. We claim that the construction of surrogate environments in our paper corrects the one in [12]; the formalized statement is our Lemma 2, with proof and details given in our Appendix B. The incorrect proof by [12] in [App. B.1, 12] contains the use of a technical inequality lemma that does not apply to the setting the authors claim (please refer to our Remark 6 in Appendix B, line 502).\
> What we show is that the desired property of a surrogate environment is achieved when the value function of TS is smaller than the average value over the partition. This informs us to take the surrogate environment as the average of the environments in the partition, and prove the statement.\
> In addition to surrogate environments construction, we corrected a fundamental issue with the surrogate information ratio bound.  We have quoted and explained the authors' argument in App. J.2.1, and shown with equations in details, and explained the intuition, for why their argument fails. We elaborate on the intuition here as well.\
> Note this ratio is used to bound the surrogate (and therefore, main) regret. In [12], the authors claim to bound this ratio by showing that the numerator (surrogate regret) is smaller than a particular surrogate mutual information: $\mathbb{I}^{\pi^*\_{\mathcal{E}}}\_\ell(\widetilde{\mathcal{E}}^*\_\ell;\mathcal{H}\_{\ell,H})$. Recall that the denominator in the (surrogate) information ratio is supposed to represent the information gain by the algorithm on the true (or surrogate) environment. What we observe is that the surrogate mutual information ratio that one should bound is:\
> \
> $
> \frac{\left(\mathbb{E}\_\ell\left[V\_{1,\pi^*\_{\mathcal{E}}}^{\widetilde{\mathcal{E}}\_\ell^*}(s\_1^\ell)-V\_{1,\pi}^{ \widetilde{\mathcal{E}}\_\ell^*}(s\_1^\ell)\right]\right)^2}{\mathbb{I}\_\ell^{\pi}\left(\widetilde{\mathcal{E}}\_\ell^*; \mathcal{H}\_{\ell, H}\right)},
> $\
> \
> where $\pi$ is the algorithm (so we select $\pi=\pi\_{\text{TS}}$). Indeed, clearly the algorithm can not know about the true environment $\mathcal{E}$, which makes it questionable to try to bound the surrogate regret
> $\mathbb{E}\_\ell\left[V\_{1,\pi^*\_\mathcal{E}}^{\widetilde{\mathcal{E}}\_\ell^*}(s\_1^\ell)-V\_{1,\pi}^{\widetilde{\mathcal{E}}\_\ell^*}(s\_1^\ell)\right]$ by a mutual information such as $\mathbb I\_\ell^{\pi^*\_{\mathcal{E}}}\left(\widetilde{\mathcal{E}}\_\ell^*; \mathcal{E}\_{\ell, H}\right):= \mathbb I\_\ell\left(\widetilde{\mathcal{E}}\_\ell^*; \mathcal{H}\_{\ell, H}|\pi^*\_\mathcal{E} \right)$, where there is **assumed** knowledge of the true environment in the conditional $\pi^*\_\mathcal{E}$, as opposed to conditioning on the algorithm itself like in the ratio above. Therefore, the mutual information used in [12] is not the right one.

---

> > ### Comment · Reviewer_z7jd · 2023-08-14
> >
> > I would like to thank the authors for their response. My score will remain the same.

---

### Official Review · Reviewer_4ohk · 2023-07-06

**Soundness:** 3 good
**Presentation:** 3 good
**Contribution:** 2 fair
**Rating:** 5
**Confidence:** 2

**Summary:**

The paper presents uniform Bayesian regret bounds for Thompson Sampling by utilizing a uniform bound of information ratio and specific bounds of the Kolmogorov dimension in different settings.


**Strengths:**

1. The paper presents a uniform Bayesian regret bound for Thompson Sampling which yields results in a variety of settings, improving upon previous approaches in some scenarios.
2. The authors incorporate a comprehensive discussion with previous works which helps to understand the contribution of the proposed bounds.


**Weaknesses:**

1. Potential overclaim: in the introduction, the authors state that they first define Bayesian RL with time inhomogeneous settings, which might be an overclaim since there are also previous works discussing this setting like [1].

[1]Variational Bayesian Reinforcement Learning with Regret Bounds. Brendan O'Donoghue

2. The presentation of the paper would benefit from a table that includes all the results discussed in the paper for a comprehensive comparison.

**Questions:**

Why is time inhomogeneous specially highlighted in this work? If we consider time also as a part of the state observation, it would be homogeneous (i.e., share the same model across all timesteps). With that being considered, is it possible just to extend the time inhomogeneous bounds to this setting, and is the proposed bound also better than those?

---

> ### Author Rebuttal · Authors · 2023-08-09
>
> We would like to thank the reviewer for their comments improving the paper, which we will include in our final revision. We also invite the reviewer to read the author rebuttal beforehand. Below are our replies:
>
> **Weaknesses**:
>
> 1. Thank you for pointing this out. The phrase "we first define" was intended to mean that we are *starting* our presentation with this notion, similar to "we first do X/we then do Y", but we realize how the phrasing causes confusion, and so we change it accordingly in our revision. We have previously referred to [12] and other works for where this notion was previously introduced, but we thank the reviewer for providing the additional reference (to be also included in our revision).
>
> 2. We would like to thank the reviewer for this suggestion. For the Bayesian RL, there are very few works, of which we can name [25] (for time-homogeneous Dirichlet-prior tabular) and [12] (for time-inhomogeneous Bayesian RL), and while the latter is general and makes comparable claims, part of our paper is essentially spent on correcting their claims and proofs. We will expand on the comparisons with these works using a table. We note that frequentist model-based bounds in the table would be misleading, and such bounds are also limited to tabular settings.
>
> **Questions**:
>
> We thank the reviewer for their question and observation. Indeed, we see that if one applies the proposed mapping, we get a time homogeneous environment as a result. Hence, we could apply the regret bounds in that setting. The regret bound in this setting would be, to our knowledge, the first Bayesian regret bound for these general time-homogeneous RLs. However, we are not sure if this mapping is surjective on the set of time-homogeneous RL problems. We appreciate this remark. We will include it in our revised paper after our main theorem and mention the reviewer's contribution in the acknowledgments.

---

> > ### Comment · Reviewer_4ohk · 2023-08-17
> >
> > Thank you for the reply and I'll keep my score.

---

> > > ### Author Response · Authors · 2023-08-18
> > >
> > > Dear Reviewer,
> > > Thank you for your reply and feedback. Please let us know if you have any further questions.

---

### Official Review · Reviewer_pUAk · 2023-07-13

**Soundness:** 2 fair
**Presentation:** 2 fair
**Contribution:** 2 fair
**Rating:** 6
**Confidence:** 3

**Summary:**

The authors propose the novel Bayesian regret analysis for posterior sampling for reinforcement learning algorithm. The proposed regret bounds are applicable in a large variety of different RL settings, such as tabular, linear and finite mixture MDPs.

**Strengths:**

- The novel analysis for posterior sampling algorithm in the setting of Bayesian regret;
- The presented result holds not only in the setting of tabular MDPs but also in linear and finite mixture MDPs.

**Weaknesses:**

- The weak notion of Bayesian regret is the main weakness of the presented result. Currently, there exists near-optimal results in posterior-sampling based algorithms in the frequentist setting (see section Questions for precise references).
- The computational side of the presented algorithm was not discusses.
- The upper bound in linear setup seems to be a contradiction with established lower bound in the setup of linear contextual bandits (see reference below for example). This effect requires additional explanations why is it possible in the presented setting.
- Lattimore, Tor, and Csaba Szepesvári. *Bandit algorithms*. Cambridge University Press, 2020.

**Questions:**

- Is there any results in literature where this type of dimension were called “Kolmogorov”? Seems that this definition is a just usual covering dimension (at least in the presented setup of $\ell_1$ distance).
- Is there examples where $\lambda$ is much smaller than $H$?
- What is the limitation to show this analysis for a more general class of MDPs such as MDPs with a finite Eluder dimension?
- Where is topological structure of $S$ and $A$ were used during the proofs? What is the topological structure of them?
- The missed references of frequentist regret bounds for TS-based exploration in tabular and linear MDPs:
    - Zanette, Andrea, et al. "Frequentist regret bounds for randomized least-squares value iteration." *International Conference on Artificial Intelligence and Statistics*. PMLR, 2020.
    - Agrawal, Shipra, and Randy Jia. "Optimistic posterior sampling for reinforcement learning: worst-case regret bounds." *Advances in Neural Information Processing Systems* 30 (2017).
    - Tiapkin, Daniil, et al. "Optimistic posterior sampling for reinforcement learning with few samples and tight guarantees." *Advances in Neural Information Processing Systems* 35 (2022): 10737-10751.
    - Agrawal, Priyank, Jinglin Chen, and Nan Jiang. "Improved worst-case regret bounds for randomized least-squares value iteration." *Proceedings of the AAAI Conference on Artificial Intelligence*. Vol. 35. No. 8. 2021.
- Line 83: In [32] they consider not episodic setup. The state-of-the-art result in the episodic setup were presented in [4].

**Limitations:**

The paper presented the theoretical research on Bayesian regret for posterior-sampling algorithms for reinforcement learning, thus it does not require discussion of ethical limitations.

---

> ### Author Rebuttal · Authors · 2023-08-09
>
> We thank the reviewer for their comments improving the paper, and invite them to read the author rebuttal beforehand. Below are our replies:
>
> **Strengths**:
>
> Thank you for the strength highlights. We also like to point out that the mentioned results are corollaries of our main contribution on general nonlinear RLs stated in Theorem 4.
>
> **Weaknesses**:
>
> 1. This suggests that the PSRL problem in the frequentist setting is (nearly-)solved. However, references considered, there exists results only for tabular Dirichlet-based RLs, a small subset of general RLs. Our work expands the scope of knowledge on the Bayesian bounds for PSRL further than what exists in the frequentist case. Regarding the references: Thank you, and we shall include them in the intro for more context. Two of those are for model-free RL (ours is model-based PS), and the other two's results are very limited in scope (tabular RL Dirichlet-based priors). To our knowledge, a frequentist bound for PS is still an open problem for general RLs and those tabular cases solved optimally are for a *variant* of PSRL (Optimistic PSRL). Our bound is an appropriate first step towards the much more general principled study for Bayesian RLs.
>
> 2. A discussion on this will appear in our revision. For general RL settings, we refer to the ICML23 [arXiv:2305.00477] where they show experiments regarding TS outperforming other methods in Deep RL. We will also point out PSRL papers with experiments (some cited in Related Works) on TS and its variants [29,20,17,13,14] and discussions on computational efficiency of PSRL.
>
> 3. Despite sharing "linear" in name, said bandit problem is not directly related to linear RLs. Though it could be cast into an RL problem, it will not be an episodic/Bayesian/inhomogeneous/and finally not a linear RL either, unless made to be so:(1) No "episode" in the definition of linear contextual bandits. (2) The inhomogeneity condition should be added too. (3) [Bandit Algorithms, 2020] mentions the frequentist case only. (4) Contexts $X_t$ are like states in an environment. According to the bandit's definition, these are pre-selected, regardless of the actions, implying an environment where transitions $P(s'|s,a)$ are independent of $a$. (5) A linear RL requires these transitions  to be linear-- expressible as $\langle \phi(s), \psi(s')\rangle $; there is no such parallel assumption in the bandit problem. So one needs to consider an "episodic inhomogeneous Bayesian linear contextual bandits with linear contextual transition functions"! It is unclear what the lower bound for the regret would be after imposing this many assumptions.
>
> **Questions**:
>
> 1. The term "Kolmogorov", as some limsup of covering *numbers*, has appeared before in e.g. [arXiv:1406.1853]. Regarding terminology usage: Our $l_1-$dimension is a type of upper box dimension, also referred to as "Kolmogorov" dimension [wikipedia, Minkowski-Bouligand dimension]. While most notions of dimensions match on well-behaved metric spaces, their definitions/scopes are not identical. In particular, *covering dimension* [wikipedia, "Lebesgue covering dimension"'] is a topological notion, while the upper box requires a metric. Further, we have (for metric spaces):\
> upper-box dim $\ge$ lower-box dim $\ge$ Hausdorff dim $\ge$ large inductive dim = covering dim;\
> The first two are in the Minkowski-Bouligand wiki. The last two are in [wikipedia, Inductive dimension]. Every single one of these inequalities can be strict, e.g. the metric space of rational numbers in $[0,1]$ with $l_1$ metric, has box dimension one, and Hausdorff dimension zero, so covering dimension is also zero [wikipedia, Hausdorff dimension]. All in all, this motivates the use of the exact and appropriate dimension terminology.
>
> 2. Thank you for the question. This discussion will be in our final version. Assume every state is reachable from another in at most $D$ steps (defined more precisely as "MDP diameter"). Using $0\le r(s,a)\le 1$, it can be shown: $\lambda \le D$. Therefore, if environments have small diameters, $\lambda$ can be much smaller than $H$. In addition, we note the conceptual contribution: the information ratio is bounded by (the $l_1-$dimension of $\Theta$) times (the diameter of the value function). The latter is always $\le H$, but its nature is different from $H$.
>
> 3. Thank you for the question. We will summarize what follows in our conclusion/future works.With regards to applicability to general MDPs, so long as our assumptions regarding posterior consistency hold, the analysis applies. We conjecture that (at least a weakened version of) our posterior consistency assumption should hold in general, and we leave that for future work. As for Eluder dimension (in)finiteness, especially used in frequentist bounds, it is of no relevance here. As Theorem 4 states, there is no assumption except for the posterior consistency. One could ask the relation between Eluder and $l_1$ dimensions, and which is easier to derive. We leave this for future work.
>
> 4. There is no assumption other than those in the preliminaries: topological spaces with probability measures. The Bayesian bound does not depend directly on some dimension of these spaces. However, given that transition functions are defined on $\mathcal{S},\mathcal{A}$, it is not surprising to see the $l_1-$dim being expressed in terms of $\mathcal{S},\mathcal{A}$, such as in the tabular case.
>
> 5. We addressed this under "Weaknesses 1.".
>
> 6. Thank you for the correction. Please note this is for tabular only. To our knowledge, there is no lower bound in the frequentist model-based for general RLs.

---

> > ### Comment · Reviewer_pUAk · 2023-08-18
> >
> > I would like to thank the authors for their response. In particular, my main concern regarding the potential contradiction with the lower bound in linear setting was properly addressed.
> >
> > Just as a small remark, under the covering number I meant the definition of [1905.00475]. However, I appreciate a deep discussion on the comparison between different types of the dimensions.
> >
> > Overall, I am happy to increase my score.

---

> > > ### Author Response · Authors · 2023-08-18
> > >
> > > Dear Reviewer,
> > > Thank you for the reply. We appreciate very much your valuable feedback and the score increase.

---

### Official Review · Reviewer_bSE3 · 2023-07-19

**Soundness:** 3 good
**Presentation:** 3 good
**Contribution:** 3 good
**Rating:** 7
**Confidence:** 2

**Summary:**

This paper considers the problem setting of Bayesian reinforcement learning, in which both the transition function and the reward function are sampled from a known prior distribution. The authors study Thompson sampling in this setting and prove a Bayesian regret bound of order O(\lambda \sqrt{ d T}) where \lambda is the average value diameter induced by the prior, d is the Kolmogorov dimension (with respect to l1 distance) of the environment, and T is the number of episodes the learner interacts with the environment.

The authors instantiate their general result to two previously studied settings.

- In the tabular setting, the authors show that their main result implies a regret bound that matches the previous state of the art by Osband and Van Roy (2017), but generalized to hold over any prior distribution rather than specific to the product of Dirichlet distribution prior considered by Osband and Van Roy.

- In the linear MDP setting, the authors give a regret bound of $O(\lambda \sqrt{d^f T})$, where $d^f$ is the Kolmogorov l1 dimension of the feature space. The authors also present a counterexample showing that the previously claimed state of the art due to Hao and Lattimore (2022) was in fact incorrect.


The paper also provides corollaries for specialized finite mixtures settings.


**Strengths:**

The paper's main contribution is a general treatment of Thompson sampling in MDPs. Specifically, the paper claims to provide the most general results to date on the Bayesian regret of Thompson sampling in RL. These results appear to generalize the results of Osband and Van Roy (2017) in the tabular case, and, when their counterexample to Hao and Lattimore (2022) is taken into account, provides the tightest upper bounds in the linear MDP case. The paper clearly signposts these contributions and provides proofs for their claims.

**Weaknesses:**

Some of the theorem statements/assumptions could be more explicit. For example, the strong consistency assumption (Assumption 1) is not rigorously defined until Appendix J. Also, in my reading of the proof of Theorem 4, it appears that there is another assumption needed in the statement (Assumption 2 from Appendix D). I think it is also worth pointing out that T_0 in Theorem 4 is doing a lot of heavy lifting, as it seems like it could actually be a very large constant, depending on the prior and the structure of the RL problem.


**Questions:**

On line 138, it is claimed that the law of Thompson sampling aligns with the true posterior distribution. Is this true with no other assumptions? For example, if two environments induce the same optimal policy, then it seems like this would not be true.

**Limitations:**

The paper does point out that they do not have lower bounds to substantiate the tightness of their upper bounds. Another limitation of the paper, not mentioned by the authors, is that the analysis is limited to the setting where the prior is specified with perfect accuracy.

---

> ### Author Rebuttal · Authors · 2023-08-09
>
> First, we would like to thank the reviewer for their thorough and careful summary of our paper and our results. We greatly appreciate the time and the comments made to improve the manuscript. We invite the reviewer to read the Author Rebuttal beforehand.
>
> **Weaknesses**:
>
> $\bullet$ Thank you for raising this issue. We will give more context and rigor to Assumption 1 within the main text, and we will bring forth the second assumption into the main text as well.
>
> $\bullet$ We will emphasize the limitation with respect to $T_0$, and indeed, while it may not change the asymptotics of the bound, its effect can be dominant for even large $T$s in practice.
>
>
> **Questions**:
>
> Thank you very much for your question. This raises a subtle mathematical question which could be addressed without any additional assumptions. Our definition of TS is exactly the one by [12], which itself follows the seminal work on IDS [29] (Van Roy \& Russo, 2014). We will include the following reply in our revised version and acknowledge the reviewer's contribution.
>
> The question is: If two or even a nonzero-measure set of environments give the same optimal policy, then how could one have
>  $\mathbb{P}(\mathcal{E}|\mathcal{D}\_\ell)=\mathbb{P}(\pi^\ell\_{\text{TS}}=\pi^*\_\ell|\mathcal{D}\_\ell),\forall\mathcal{E}$, while also having the latter as a probability measure on the set of optimal policies $\Pi^*$, i.e. $\int\_{\Pi^*} \mathbb{P}( \pi^*|\mathcal{D}\_\ell) \text{d} \rho\_{\Pi^*}= 1$?
>
> The answer is to define the measure $\rho\_{\Pi^*}$ on $\Pi^*$ to have appropriate measure on optimal policies $\pi^*$, based on the set environments of which $\pi^*$ is an optimal policy. Mathematically, this means that the map $star : \Theta \to \Pi^*$, where $star(\mathcal{E}) = \pi^*\_{\mathcal{E}}$, must be used to define $\rho\_{\Pi^*}$:
>
> $ \rho\_{\Pi^*}(\mathcal{O}) := \rho(star^{-1}(\mathcal{O})), \ \forall \mathcal{O} \subset \Pi^* $
>
> i.e., $\rho\_{\Pi^*}$ is defined as the push-forward of the prior measure $\rho$ on the set of environments under the map $star$. This ensures that even when a nonzero measure set of environments have the same optimal policy, it is possible to postulate the law of TS to be $\mathbb{P}( \mathcal{E}|\mathcal{D}\_\ell) = \mathbb{P}( \pi^\ell\_{\text{TS}} = \pi^*\_\ell |\mathcal{D}\_\ell), \forall \mathcal{E}$.
>
> **Limitations**:
>
> We thank the reviewer for their observation. Studying cases with prior misspecificity will be included as part of our limitations and future studies in our revised version.

---

> ### Comment · Reviewer_bSE3 · 2023-08-14
>
> Thank you to the authors for the thorough response and for answering my question. After reading the response and the other reviews, my view of this paper remains positive, and I will keep my score as accept.

---

### Author Rebuttal · Authors · 2023-08-09

We thank very much all reviewers for their questions and comments, which we will include in our revision. Please note that we had to address each reviewer's response within the character limit. We kindly invite all reviewers to ask any further questions they have in the discussion period.

Please also note that in our replies, references are cited with either of these two formats:

(1) With the arXiv code, e.g. "[1406.1853]" is meant to reference arXiv:1406.1853, Or,

(2) With the associated number in the reference section of the paper, e.g. [29] refers to "Daniel Russo and Benjamin Van Roy. Learning to optimize via information-directed sampling. Advances in Neural Information Processing Systems, 27, 2014."

---

> ### Author Response · Authors · 2023-08-18
>
> We would like to thank all reviewers for responding to our rebuttals. We appreciate the discussions and feedbacks. Please let us know if there are any further questions.

---

### Decision · Program_Chairs · 2023-09-21

**Decision:**

Accept (poster)

**Comment:**

Studies the performance of Thompson sampling on general RL problems when the true prior over rewards and transitions is provided to the learner. The Bayesian regret is given in terms of a quantity called the Kolmogorov l1 dimension—a measure connected to covering numbers of the MDP. For specific paradigms of MDPs including tabular, linear, and mixture, the authors bound this quantity providing results that match prior art. While the general treatment is appreciated, different reviewers were disappointed to not find a lower bound in terms of this Kolmogorov dimension, calling into question the tightness of the results. Nevertheless, such a general treatment is appreciated for such a challenging set of problems and spurs new directions to be explored.